# UPP1 promotes lung adenocarcinoma progression through the induction of an immunosuppressive microenvironment

Yin Li [1,6], Manling Jiang [2,6], Ling Aye [3,6], Li Luo [2,6], Yong Zhang [4], Fengkai Xu [1], Yongqi Wei [1], Dan Peng [2], Xiang He [2], Jie Gu [1], Xiaofang Yu [5], Guoping Li [2] ✉, Di Ge [1] ✉ & Chunlai Lu [1] ✉

The complexity of the tumor microenvironment (TME) is a crucial factor in lung adenocarcinoma (LUAD) progression. To gain deeper insights into molecular mechanisms of LUAD, we perform an integrative single-cell RNA sequencing (scRNA-seq) data analysis of 377,574 cells from 117 LUAD patient samples. By linking scRNA-seq data with bulk gene expression data, we identify a cluster of prognostic-related UPP1[high] tumor cells. These cells, primarily situated at the invasive front of tumors, display a stronger association with the immunosuppressive components in the TME. Our cytokine array analysis reveals that the upregulation of UPP1 in tumor cells leads to the increased release of various immunosuppressive cytokines, with TGF-β1 being particularly prominent. Furthermore, this UPP1 upregulation also elevates the expression of PD-L1 through the PI3K/AKT/mTOR pathway, which contributes to the suppression of CD8 + T cells. Cytometry by time-of-flight (CyTOF) analysis provides additional evidence of the role of UPP1 in shaping the immunosuppressive nature of the TME. Using patient-derived organoids (PDOs), we discover that UPP1[high] tumors exhibit relatively increased sensitivity to Bosutinib and Dasatinib. Collectively, our study highlights the immunosuppressive role of UPP1 in LUAD, and these findings may provide insights into the molecular features of LUAD and facilitate the development of personalized treatment strategies.

Lung cancer remains the primary cause of cancer-related deaths globally, with lung adenocarcinoma (LUAD) being its most dominant histological subtype[1,2]. Despite considerable progress in therapeutic approaches for LUAD, particularly with the introduction of immunotherapy, the prognosis remains less than favorable for a substantial number of patients[3]. Challenges such as the heterogeneity of the tumor microenvironment (TME), acquired resistance to treatments,

and discrepancies in patient responses further complicate the therapeutic landscape[4,5]. This highlights the need for personalized treatment strategies and a deeper understanding of the molecular mechanisms to enhance patient outcomes.

The LUAD TME is a complex and multifaceted cellular environment, composed of a diverse array of cell types, including tumor cells, immune cells like T cells, B cells, and macrophages, as well as

[1]Department of Thoracic Surgery, Zhongshan Hospital, Fudan University, Shanghai 200032, China. [2]Laboratory of Allergy and Precision Medicine, Chengdu Institute of Respiratory Health, Affiliated Hospital of Southwest Jiaotong University, The Third People's Hospital of Chengdu, Chengdu 610031 Sichuan, China. [3]Shanghai Medical College, Fudan University, Shanghai 200032, China. [4]Department of Pulmonary and Critical Care Medicine, Zhongshan Hospital, Fudan University, Shanghai 200032, China. [5]Department of Nephrology, Zhongshan Hospital, Fudan University, Shanghai 200032, China. [6]These authors contributed equally: Yin Li, Manling Jiang, Ling Aye, Li Luo. ✉e-mail: lzlgp@163.com; ge.di@zs-hospital.sh.cn; springcoming127@qq.com

endothelial cells and fibroblasts[6]. Within this environment, tumor cells engage in complex interactions with surrounding immune and stromal cells. These inter-cellular interactions are central to the initiation, development, and eventual prognosis of the tumor, fostering an immunosuppressive environment that inhibits the immune response against the tumor[7]. This dynamic within the TME has profound implications for therapeutic strategies, especially immunotherapies, emphasizing the importance of understanding these interactions to enhance the efficacy of cancer treatments[8,9].

Uridine phosphorylase 1 (UPP1) plays a vital role in uridine metabolism, maintaining homeostasis and aiding pyrimidine salvage[10]. It cleaves uridine to uracil and ribose-1-phosphate (R1P), supporting glycolysis, especially when nutrients are limited[11]. In tumors with low glucose, UPP1 utilizes ribose from uridine to sustain tumor cell metabolism, enhancing their growth and survival[12]. Recent research on UPP1's role in LUAD have shown its capacity to modulate tumor cell sensitivity to glycolysis inhibitors, thereby driving glycolytic pathways and enhancing tumor growth[13]. Interestingly, evidence also suggests that inhibiting UPP1 enhances the infiltration of anti-tumor T cells[12]. Furthermore, Wang et al. uncovered a close association between UPP1 and immune checkpoints within the TME, suggesting a potential link between UPP1 and the TME[14]. However, the exact role of UPP1 within the LUAD TME and its potential influence in shaping it remain to be elucidated.

In this work, we conduct an integrative single cell RNA sequencing (scRNA-seq) data analysis for LUAD, uncovering a cluster of prognostic-related UPP1[high] tumor cells. Using multiplex immunofluorescence staining on patient samples from LUAD, we show that UPP1[high] tumor cells predominantly localize at the invasive front of the tumor. In addition, we reveal that upregulation of UPP1 in tumor cells influences the release of immunosuppressive cytokines and the expression of PD-L1. Further cytometry by time-of-flight (CyTOF) analysis substantiates that upregulation of UPP1 in tumor cells contributes to the shaping of an immunosuppressive TME. These findings may provide more insights into the molecular features of LUAD and help facilitate the development of personalized treatment strategies.

## Results

### Integrative single-cell RNA sequencing data analysis identifying UPP1[high] tumor cells and the association of UPP1 with patient prognosis

To investigate the tumor microenvironment (TME) features of LUAD, we integrated five independent single-cell RNA sequencing (scRNA-seq) datasets of LUAD (Fig. 1a, Supplementary Fig. 1a, and Supplementary Data 1). After a series of data quality control measures, including data integration, normalization, and batch effect removal[15,16], a total of 117 patient samples and 377,574 cells were included for subsequent analysis (Fig. 1b, and Supplementary Fig. 1b). Unbiased dimensionality reduction and clustering of these cells led to the identification of eight major cell populations based on the expression of classical marker genes, including T cells, NK cells, B cells, myeloid cells, mast cells, fibroblasts, endothelial cells, and epithelial cells (Fig. 1c, and Supplementary Fig. 1c–g). Within the epithelial cell population, the inferCNV algorithm[17] was utilized to identify tumor cells, resulting in a total of 49,113 tumor cells for further analysis (Supplementary Fig. 1h).

Further dimensionality reduction and clustering were applied to the major cell populations, and this analysis revealed distinct subpopulations within the major cell types, including 9 sub-populations of CD4 + T cells, 6 sub-populations of CD8 + T cells, 5 sub-populations of NK cells, 3 sub-populations of dendritic cells, 9 sub-populations of macrophages, 2 sub-populations of mast cells, 9 sub-populations of B cells, 8 sub-populations of fibroblasts, 6 sub-populations of endothelial cells, and 20 sub-populations of tumor cells (Fig. 1d, e and Supplementary Fig. 2). Cell sub-populations from different cohorts were

evenly distributed, indicating the effectiveness of data integration and the biological representativeness of the identified cell sub-populations (Supplementary Fig. 3, and Supplementary Data 2).

Cell-type profiles derived from scRNA-seq data can be linked with bulk transcriptomes and can reflect cancer clinical outcomes[18]. To explore the association between the identified tumor cell subpopulations and the clinical outcomes of LUAD patients, we computed the relative cell abundances of these tumor cell sub-populations based on their specific marker genes (Log2FC > 1 and adjusted $p < 0.05$) using the single sample Gene Set Enrichment Analysis (ssGSEA) algorithm[19] in three independent bulk datasets (bCohort) encompassing 1529 patients (Supplementary Data 1). We then conducted a prognosis analysis correlating these tumor cell abundances with the overall survival (OS) of the LUAD patients (Supplementary Fig. 4a). This analysis revealed that seven cell populations exhibited a significant correlation with poorer patient outcomes, including tumor cell cluster 3 (highly expressing S100A2), cluster 8 (TK1), cluster 10 (PTTG1), cluster 12 (IGFBP5), cluster 13 (ISG15), cluster 16 (UPP1), cluster 17 (IL13RA2) (Fig. 1f and Supplementary Fig. 4b). Consistent with previous studies, the genes highly expressed in these prognostic-related cell populations (Supplementary Fig. 5a, b) have also been reported to be notably associated with the progression and prognosis of LUAD patients[20–24]. However, it's noteworthy that the role of UPP1 in LUAD is relatively less studied, and its involvement in tumor progression is not yet fully elucidated. Thus, we specifically focused on the UPP1[high] tumor cell population and the role of UPP1 in the context of LUAD.

Functional enrichment analysis of the UPP1[high] tumor cell population demonstrated these cells were related to cancer-related biological processes, including Tumor_Invasiveness, PI3K/AKT/mTOR_signaling_pathway, MYC_targets, MAPK_signaling_pathway, as well as TGFβ_signaling_pathway. Additionally, there is a notable connection with tumor immunity-related biological processes, such as Inflammatory_Response and Regulation_of_Immune_Responses (Supplementary Fig. 5c, and Supplementary Data 4). Subsequently, we calculated the average functional enrichment score for the significant biological processes enriched in this UPP1[high] tumor cell population. By correlating this with the top 10 highly expressed genes in this group of cells, we aimed to determine which gene most prominently defines the functional characteristics of this cell group. This analysis revealed that UPP1 exhibited the strongest correlation with the functional enrichment scores of these cells, highlighting its pivotal role in the functional attributes of this cell population (Fig. 1g).

Subsequently, utilizing our tissue microarray (TMA) cohort from Zhongshan Hospital, which encompassed 205 LUAD patients, we conducted immunohistochemistry staining for UPP1. The expression of UPP1 in tumor cells was then analyzed using the AI-based software, Aipathwell[25,26]. Our data indicated that elevated UPP1 expression was inversely correlated with both overall survival (OS) and recurrence-free survival (RFS) among LUAD patients (Fig. 1h, i, Supplementary Fig. 5d, e, and Supplementary Data 5). Additionally, based on the proteomic resources for LUAD patients provided by Xu et al.[27], we assessed the relationship between UPP1 expression and patient outcomes (Supplementary Fig. 5f). In this cohort, patients with higher UPP1 protein levels also exhibited comparatively poorer prognosis. Notably, the expression of UPP1 also exhibited a correlation with the histological differentiation grade of the malignancy. Taken together, these findings highlight the pivotal role of UPP1 as a prognostic marker in LUAD.

### UPP1[high] tumor cells are associated with the immunosuppressive tumor microenvironment

To assess the interactions of UPP1[high] tumor cells with other cells in the TME, we analyzed their co-enrichment patterns with other cell populations based on the correlation values between the frequency of

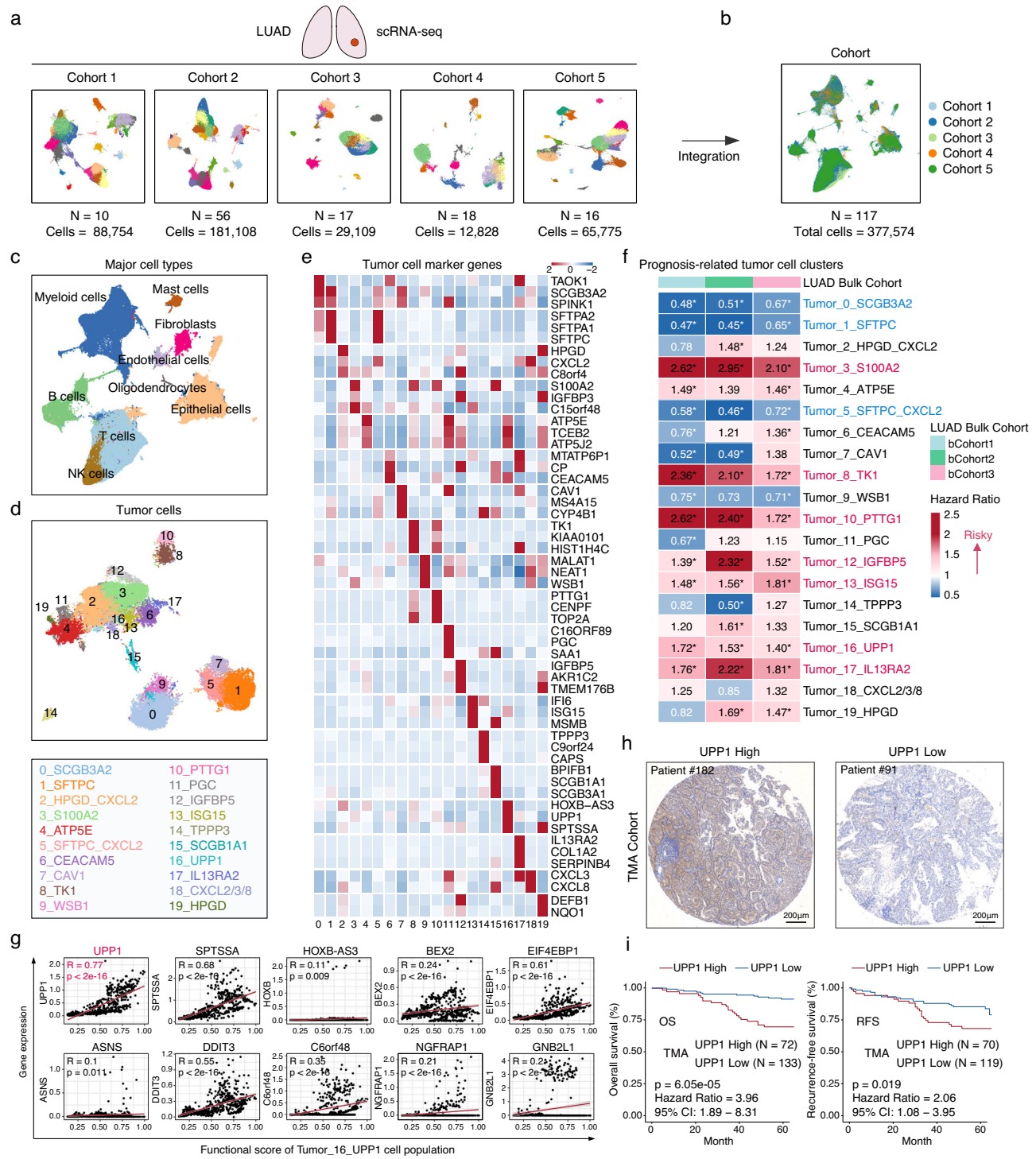

UPP1high tumor cells and other cell populations across tumor samples[28]. This analysis showed that the presence of UPP1high tumor cells was significantly positively associated with FOXP3+ regulatory T cells (Tregs), MMP11+ cancer-associated fibroblasts (CAFs), LAG3 + PDCD1+CD8+ exhausted T cells, and M2-like SPP1+ macrophages (Supplementary Fig. 6a–e). Notably, these four cell populations are recognized for their role in promoting tumor progression and their close association with the immunosuppressive nature of the TME[29–32]. A validation in the bulk datasets also revealed that UPP1high tumor cells exhibited a strong correlation with these four cell groups (Supplementary Fig. 6f, g). To gain a better understanding of the interactions between UPP1high tumor cells and these four cell populations within the

TME, we used the CellphoneDB analysis[33] for inter-cellular communication network evaluation. This analysis confirmed the mutual interaction relationships between UPP1high tumor cells and the aforementioned four cell populations (Supplementary Fig. 6h).

The CellphoneDB analysis of ligand-receptor pairs showed that the interaction networks between UPP1high tumor cells and FOXP3+ Tregs/LAG3 + PDCD1+CD8 + T cells were involved with numerous immune checkpoints, including CD274/PDCD1LG2_PDCD1, FGL1_LAG3, and NECTIN_TIGIT. Also, IL2_IL2_receptor, TGFB1_TGFbeta_receptor2/ TGFB1_TGFBR3 ligand-receptor interactions were enriched. The interactions with MMP11+CAFs were associated with FGFs_FGFRs, PDGFD_PDGFRB, VEGF_NRPs/KDR, and TGFB1_integrin_aVb6_complex.

**Fig. 1 | Integration of LUAD scRNA-seq data and identification of prognostic-related tumor cell populations. a** scRNA-seq data from five LUAD cohorts. Cohort 1 (10 samples, 88,754 cells), Cohort 2 (56 samples, 181,108 cells), Cohort 3 (17 samples, 29,109 cells), Cohort 4 (18 samples, 12,828 cells), Cohort 5 (16 samples, 65,775 cells). **b** 117 patient samples and 377,614 cells were included for subsequent analysis. The Uniform Manifold approximation and Projection (UMAP) plot showing the cell distribution. **c** The UMAP plot showing the major cell populations. **d** The UMAP plot displaying tumor cell clusters. 20 distinct tumor cell clusters were identified (top). The top marker gene for each of these clusters is presented (bottom). **e** The heatmap showing the mean expression of the top three marker genes for the 20 tumor cell clusters. **f** The prognostic (overall survival) association of each tumor cell cluster. The number in the heatmap representing the hazard ratio for each tumor cell cluster. A hazard ratio greater than 1 (shown in red) suggested that the cell cluster was associated with poor prognosis. Conversely, a hazard ratio less than 1 (shown in blue) suggested that patients with a relatively higher proportion of this cell cluster tended to have a better prognosis. The asterisk (*) indicated $p < 0.05$. If the cell clusters consistently exhibited either an association with poor prognosis or a trend towards a better prognosis in all three bulk cohorts, the names of these cell clusters were highlighted: red indicating an association with poor patient prognosis, blue suggesting a relatively better prognosis. Statistical analysis was conducted using log-rank tests. **g** In Tumor_16_UPP1 tumor cell cluster, the correlations between the expression levels of the top 10 marker genes and the functional score of the enriched biological processes. The correlation analysis was conducted using the two-tailed Pearson's correlation. **h** Representative IHC staining images of UPP1 in our tissue microarray (TMA) cohort from Zhongshan Hospital ($n = 205$). Scale bars = 200 μm. **i** Kaplan–Meier overall survival and recurrence-free survival curves of UPP1 expression in our TMA. OS analysis ($n = 205$). RFS analysis ($n = 189$). Statistical analysis was conducted using log-rank tests. Source data are provided as a Source Data file.

The interplay between UPP1high tumor cells and SPP1+ macrophages was linked with a variety of chemokines, including CCL2_CCR2, CCL3_CCR5, CXCL3_CXCR2, and CXCL8_CXCR1, as well as interactions such as IL1B_IL1_receptor and TGFB1_TGFbeta_receptor1 (Supplementary Fig. 7a). These diverse intercellular interactions suggested that UPP1high tumor cells, together with these cell groups, potentially contribute to an immune-suppressive TME, which could be associated with unfavorable prognosis of LUAD patients.

To further investigate the impact of UPP1high tumor cells and their associated cell groups on the prognosis of LUAD patients, we constructed a UPP1-related TME module based on the marker genes of these five cell populations (Supplementary Fig. 7b). The UPP1-related TME module score for each patient within the bulk datasets was calculated using the GSVA algorithm[34,35], and its correlation with patients' OS was analyzed. Our findings indicated a strong association between the UPP1-related TME module and patient prognosis (Supplementary Fig. 7b). A closer examination of the TME characteristics between high and low module scores revealed that patients with a high module score exhibited pronounced immune-suppressive features. This was evident from the elevated expression of immune checkpoints, enrichment of immune-suppressive signals, and increased CAFs and angiogenesis activities (Supplementary Fig. 7c).

We next sought to confirm that UPP1high tumor cells were indeed spatially related to these four cell populations mentioned through multiplex immunofluorescence staining on LUAD patient samples (Supplementary Fig. 7d, and Supplementary Data 5). Unbiased phenotypic identification and quantification using AI-based Visiopharm software (https://visiopharm.com) demonstrated that UPP1high tumor cells predominantly located at the invasive margins where the tumor interfaces with immune cells and the stroma (Fig. 2). Further spatial distance analysis revealed that UPP1high tumor cells were in closer proximity to FOXP3+ Tregs, MMP11+ fibroblasts, LAG3 + PDCD1 + CD8+ exhausted T cells, and SPP1+ macrophages as compared to UPP1low tumor cells (Supplementary Fig. 8), suggesting a stronger crosstalk between UPP1high tumor cells and these immunosuppressive components in the TME. These findings provided substantial evidence supporting the association of UPP1high tumor cells with the immuno-suppressive TME.

## High expression of UPP1 in tumor cells drives immunosuppression in a TGF-β1-dependent manner

To further determine whether the high expression of UPP1 in tumor cells contributes to immunosuppression through specific ligands such as cytokines or chemokines, we conducted an assessment of alterations in cytokine expressions in the LUAD cell line overexpressing UPP1 using the high-throughput 80-cytokine array (Fig. 3a). Our findings unveiled a significant upregulation in a variety of cytokines in UPP1-overexpressing tumor cells, including TGF-β1, GM-CSF, IL-1β, IL-6, IGFBP-3, CXCL5, CCL20, and VEGF, with TGF-β1 being the most prominently elevated (Fig. 3b, Supplementary Fig. 9a, b, and Supplementary Data 6). Subsequent ELISA experiments confirmed that the secretion level of TGF-β1 from the tumor cells also significantly increased following UPP1 upregulation (Fig. 3c). Notably, TGF-β1 is the predominant form of TGF-β in TME that contributes greatly to the formation of an immune-suppressive microenvironment[36–38]. It could promote the differentiation of CD4+ Treg cells and inhibit the cytotoxic activity of CD8 + T cells[39,40]. Also, it could direct macrophages towards an immune-suppressive M2 phenotype[41,42], and facilitate the accumulation of CAFs and deposition of extracellular matrix (ECM)[43,44].

Considering the TGF-β signaling was broadly engaged in the intercellular interactions between UPP1high tumor cells and the four types of immune-suppressive-related cell populations mentioned above (Supplementary Fig. 7a), we hypothesized that TGF-β1 may serve as one of the key signaling molecules secreted by UPP1high tumor cells and promote the conversion of these cells towards an immune-suppressive phenotype. To validate the role of TGF-β1 in the immunosuppressive effects mediated by UPP1high tumor cells, we conducted indirect co-culturing experiments. Specifically, CD4 + T cells, CD8 + T cells, THP-1-derived macrophages, and HFL1 fibroblasts were co-cultured with UPP1-overexpressing tumor cells (Fig. 3d).

Our observations from the co-culture of UPP1-overexpressing tumor cells with CD4 + T cells highlighted that UPP1 upregulation in tumor cells significantly drove the differentiation of CD4 + T cells into CD25 + FOXP3+Tregs. However, introducing TGF-β1 neutralizing antibodies considerably curtailed this differentiation (Fig. 3e and Supplementary Fig. 9c).

Similarly, the co-culturing of UPP1-overexpressing tumor cells with CD8 + T cells facilitated the transformation of CD8 + T cells towards a LAG3 + PDCD1+ exhausted T cell phenotype. By blocking the release of TGF-β1 from UPP1-overexpressing tumor cells using TGF-β1 neutralizing antibodies, we observed a marked decrease in the LAG3 + PDCD1+ exhausted T cell population (Fig. 3f and Supplementary Fig. 9d).

Furthermore, when UPP1-overexpressing tumor cells were co-cultured with macrophages, there was an evident upregulation of SPP1 expression in the macrophages (Fig. 3g, h). This was accompanied by a shift towards an M2-like macrophage phenotype, characterized by elevated expressions of markers like PD-L1, CD163, and CCL20 (Fig. 3i). Subsequently, this M2 polarization was mitigated upon treatment with TGF-β1 neutralizing antibodies.

Lastly, scRNA-seq analyses of fibroblasts revealed that fibroblasts co-cultured with either UPP1-overexpressing tumor cells or control cells exhibited distinct gene expression profiles (Fig. 3j). Fibroblasts in the presence of UPP1-overexpressing tumor cells displayed a higher CAF score (Fig. 3k) and increased expression of CAF markers such as FAP and MMP11 (Fig. 3l). Based on the marker genes of previously mentioned MMP11+ CAFs (Supplementary Fig. 7a), which we identified

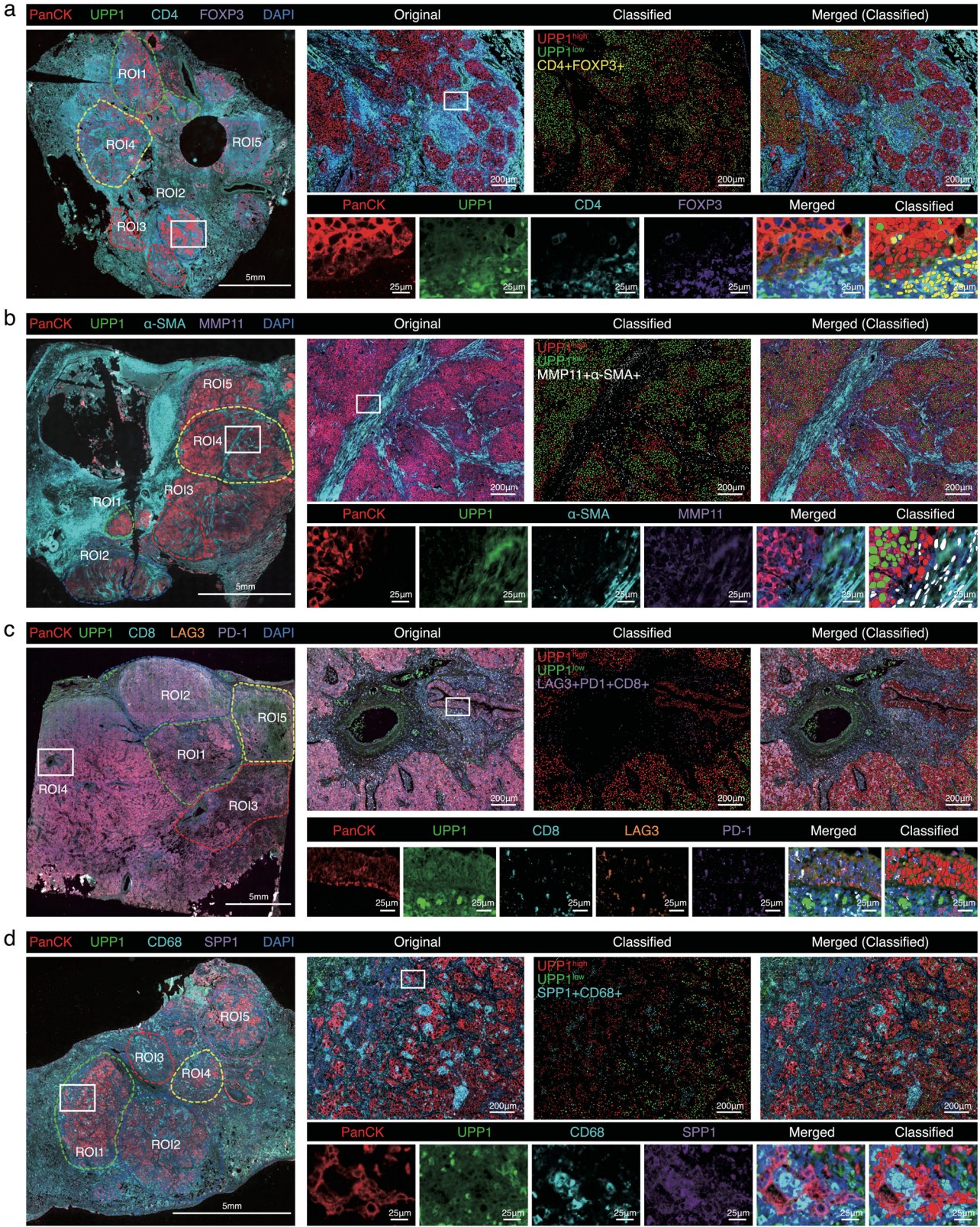

as having a co-enrichment pattern with UPP1^high tumor cells in LUAD patient samples, we calculated the proportion of MMP11+ CAFs-like fibroblasts in the co-cultured groups. The results showed that the proportion of MMP11+ CAFs-like fibroblasts was significantly higher when co-cultured with UPP1-overexpressing tumor cells than with control cells (Fig. 3m, n). These fibroblasts displayed stronger pro-

tumor effects, such as enhanced collagen formation, epithelial-mesenchymal transition (EMT), ECM remodeling, and activation of the TGF-β signaling pathway (Fig. 3o). While the treatment with TGF-β1 neutralizing antibodies could suppress the expression levels of FAP and MMP11 in fibroblasts when co-cultured with UPP1-overexpressing tumor (Fig. 3p).

**Fig. 2 | Multiplex immunofluorescence staining (mIF) on LUAD patient samples assesses the spatial correlations between UPP1 tumor cells and related cell populations. a** mIF of PanCK/UPP1 (UPP1$^{high/low}$ tumor cells), and CD4/FOXP3 (FOXP3 + CD4 + T cells) (*n* = 15), representative images shown. **b** mIF of PanCK/UPP1 (UPP1$^{high/low}$ tumor cells), and α-SMA/MMP11 (MMP11+ fibroblasts) (*n* = 15), representative images shown. **c** mIF of PanCK/UPP1 (UPP1$^{high/low}$ tumor cells), and CD8/LAG3/PD-1 (LAG3 + PD-1 + CD8 + T cells) (*n* = 15), representative images shown. **d** mIF of PanCK/UPP1 (UPP1$^{high/low}$ tumor cells), and CD68/SPP1 (SPP1 + CD68+

macrophages) (*n* = 15), representative images shown. The image on the far left represents a panoramic scan, with the ROIs indicating the selected areas. Scale bars = 5 mm. The image on the upper right represents a magnified portion of the white square in the panoramic scan on the left, which is displayed as an original image, classified image, and merged image. Scale bars = 200 μm. The images below represent further magnified views of each antibody staining channel from the selected area. Scale bars = 25 μm. For the ROIs of each section, the spatial distances between the tumor cells and the immune/fibroblast cells were calculated.

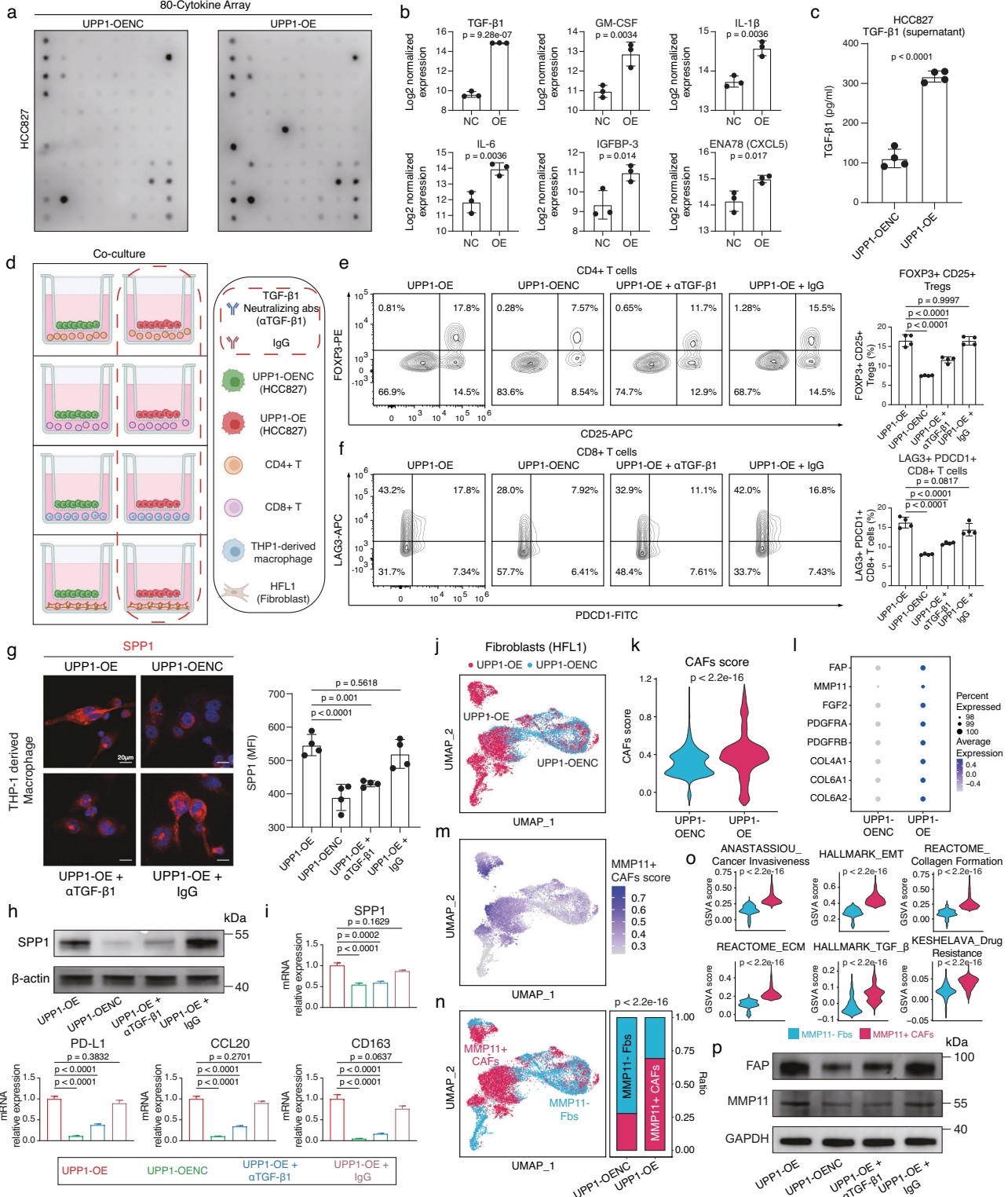

**Fig. 3 | Elevated UPP1 expression in tumor cells promotes immunosuppression through a TGF-β1-dependent manner. a** 80-cytokine array analysis between UPP1-overexpressing HCC827 tumor cells (UPP1-OE) and control cells (UPP1-OENC) (*n* = 3). **b** Top six upregulated cytokines in HCC827 UPP1-OE tumor cells (*n* = 3). **c** ELISA comparison of TGF-β1 expression in culture supernatants of HCC827 UPP1-OE and UPP1-OENC tumor cells (*n* = 4). **d** Co-culture workflow, created with BioRender.com. αTGF-β1, TGF-β1 neutralizing antibodies; IgG, isotype control IgG. **e** Flow cytometry analysis of CD25 + FOXP3+ Tregs proportion (*n* = 4). **f** Flow cytometry analysis of LAG3 + PDCD1 + CD8 + T cells proportion (*n* = 4). **g** Representative immunofluorescence (IF) staining of SPP1 in THP-1-derived macrophages (*n* = 4). MFIs represent mean fluorescence intensities. Scale bars = 20 μm, **h** Western blot analysis of SPP1 expression in THP-1-derived macrophages after co-culture (*n* = 3). **i** RT-qPCR measurement of M2 macrophage markers in THP-1-derived macrophages post co-culture (*n* = 3). **j** scRNA-seq analysis of fibroblasts after co-culturing with HCC827 UPP1-OE tumor cells/UPP1-OENC tumor cells. UPP1-OE tumor cells = 7707; UPP1-OENC tumor cells = 8200. **k** Comparison of CAFs

scores in fibroblasts co-cultured with either UPP1-OE tumor cells or UPP1-OENC tumor cells. **l** Dot plot showing the mean expression of CAFs marker genes. **m** The UMAP plot displaying the distribution of the MMP11+ CAFs scores. Fibroblasts with an MMP11+ CAFs score higher than the median value were categorized as MMP11+ CAFs-like fibroblasts, while the others were defined as MMP11- fibroblasts (Fbs). **n** Comparison of the proportion of MMP11+ CAFs-like fibroblasts between the UPP1-OE tumor cells group and the UPP1-OENC tumor cells group. **o** Comparison of the CAFs-related biological processes between the UPP1-OE tumor cells group and the UPP1-OENC tumor cells group. **p** Western blot analysis of CAFs markers after the treatment of TGF-β1 neutralizing antibodies (*n* = 3). *n* denotes biologically independent samples. Data were represented as mean ± SD in **b**, **c**, **e**–**g**. Data were represented as mean ± SEM in **i**. Two-tailed student's *t*-test was used in **b**, **c**; one-way ANOVA with multiple comparisons in **e**–**g**, **i**; two-tailed Wilcoxon rank-sum test in **k** and **o**. Two-tailed Fisher's exact test in **n**. Source data are provided as a Source Data file.

## UPP1 regulates PD-L1 expression via PI3K/AKT/mTOR pathway

Immune checkpoint expression serves as a key strategy for tumor cells to evade immune defenses[45]. Considering the enrichment of immune checkpoints in the cell-cell interaction networks between UPP1[high] tumor cells and T cells (Supplementary Fig. 7a), we next focused on whether high UPP1 expression in tumor cells could lead to the upregulation of immune checkpoints, thereby promoting immune evasion. Using scRNA-seq datasets, bulk datasets, and CCLE mRNA and protein expression datasets[46,47], correlations between UPP1 and various tumor immune checkpoints were assessed. Intriguingly, a significant correlation between UPP1 and PD-L1 expression was observed across these datasets (Fig. 4a). To validate this association, PD-L1 immunohistochemical staining was also performed on our TMA cohort, revealing a consistent and significant linkage between UPP1 and PD-L1 (Fig. 4b, c, Supplementary Fig. 10a, and Supplementary Data 5). In addition, the western blotting and flow cytometry analysis showed that the expression level of UPP1 positively related with PD-L1 in tumor cells (Fig. 4d, e).

To explore the underlying mechanism by which UPP1 modulates PD-L1 expression in tumor cells, we conducted an enrichment analysis on six classical signaling pathways known to regulate PD-L1 in UPP1[high] tumor cells[45]. The results revealed that the PI3K/AKT/mTOR pathway was the most active in UPP1[high] tumor cells (Fig. 4f). Previous studies have shown that the phosphorylation of PI3K/AKT/mTOR could regulate PD-L1 expression[48]. Therefore, we hypothesized that UPP1 may regulate PD-L1 expression through the PI3K/AKT/mTOR signaling pathway. Our findings confirmed that upregulation of UPP1 led to the activation of the PI3K/AKT/mTOR pathway (Fig. 4g). Subsequently, upon UPP1 inhibition and subsequent treatment with SC79, an AKT activator, the downregulation of PD-L1 induced by UPP1 suppression was reversed via the activation of AKT/mTOR (Fig. 4h). This suggested that high UPP1 expression in tumor cells contributed to the upregulation of PD-L1 through the PI3K/AKT/mTOR pathway.

Next, we focused on whether the upregulation of PD-L1 in UPP1-overexpressing tumor cells could affect the direct killing ability of CD8 + T cells. UPP1 was overexpressed in LLC-OVA tumor cells, consistently, the upregulation of UPP1 led to an increase in PD-L1 levels (Supplementary Fig. 10b–d). Subsequently, UPP1-overexpressing LLC-OVA tumor cells and control LLC-OVA cells were directly co-cultured with OT-1 CD8 + T cells and the results showed that UPP1-overexpressing tumor cells demonstrated a reduced susceptibility to T cell-mediated elimination (Supplementary Fig. 10e). However, the treatment of PD-L1 antibodies could partially alleviate the impaired ability of OT-1 CD8 + T cells to eliminate UPP1-overexpressing LLC-OVA tumor cells (Supplementary Fig. 10f, g). Similarly, flow cytometry counting of alive tumor cells after co-culture revealed that PD-L1 blockade could reduce the proportion of UPP1-overexpressing LLC-OVA tumor cells that survived the attack of OT-1 CD8 + T cells

(Supplementary Fig. 10h). Besides, the assessment of cytotoxic markers, Perforin and Granzyme B, in OT-1 CD8 + T cells showed that the upregulation of UPP1 in tumor cells significantly reduced the cytotoxicity of OT-1 CD8 + T cells, while this killing ability could be partially restored by blocking PD-L1 (Fig. 4i and Supplementary Fig. 10i). These results supported that UPP1-induced PD-L1 expression could affect the killing ability of CD8 + T cells.

## CyTOF analysis confirming the immunosuppressive role of UPP1 in vivo

We next conducted a comparative study using both immunocompetent (C57BL/6) and immunodeficient (nude) mice to confirm the association between UPP1 and tumor immunity in vivo. UPP1-overexpressing tumor cells, UPP1-downregulated tumor cells, and their respective controls were subcutaneously implanted into both C57BL/6 mice and nude mice, respectively (Supplementary Fig. 11a, b). Remarkably, UPP1-overexpressing tumors in C57BL/6 mice exhibited a growth pattern similar to that in nude mice (Supplementary Fig. 11c, d). However, tumors without UPP1 upregulation exhibited suppressed growth in C57BL/6 mice compared to nude mice, suggesting that without the high expression level of UPP1, the tumor cells are unable to sustain their growth in an active immune system, indicating a potential relation between UPP1 and immune evasion. On the other hand, when UPP1 expression in tumor cells was inhibited, although the downregulation of UPP1 led to a certain degree of tumor growth suppression in the nude mice, this inhibitory effect was significantly more pronounced in the immunocompetent C57BL/6 mice (Supplementary Fig. 11e, f). These results indicated that the high expression of UPP1 played a significant role in helping tumors evade the immune system.

To substantiate that UPP1 upregulation in tumor cells can promote the shaping of an immunosuppressive microenvironment in vivo, we performed the CyTOF analysis, covering 21 TME-related protein markers (Fig. 5a, Supplementary Fig. 12a, and Supplementary Table 1). Based on marker genes of immune cells, six major categories of immune cells were identified in the tumor microenvironment, including B cells (CD19+), CD4 + T cells (CD3e + CD4 + ), CD8 + T cells (CD3e + CD8a + ), myeloid cells (CD11b + ), NK cells (NK1.1 + ), and a small group of double-negative CD3 + T cells (CD3e + CD4-CD8a-) (Fig. 5b, and Supplementary Fig. 12b, c).

Regarding CD4 + T cells, the proportion of immunosuppressive CD25 + FOXP3+ Tregs significantly increased in the UPP1-overexpressing group (Fig. 5c, d). Meanwhile, the CD25 + FOXP3+ Tregs in the UPP1-overexpressing group demonstrated a higher expression level of the immunosuppressive cytokine IL-10. For CD8 + T cells, the overexpression of UPP1 caused a marked exhaustion of CD8 + T cells, characterized by the upregulation of various immune checkpoint molecules, including PD-1, LAG3, TIM3, and CTLA4 (Fig. 5e, f). Besides, the proportion of TNFα + /IFNγ+ effector CD8 + T cells was

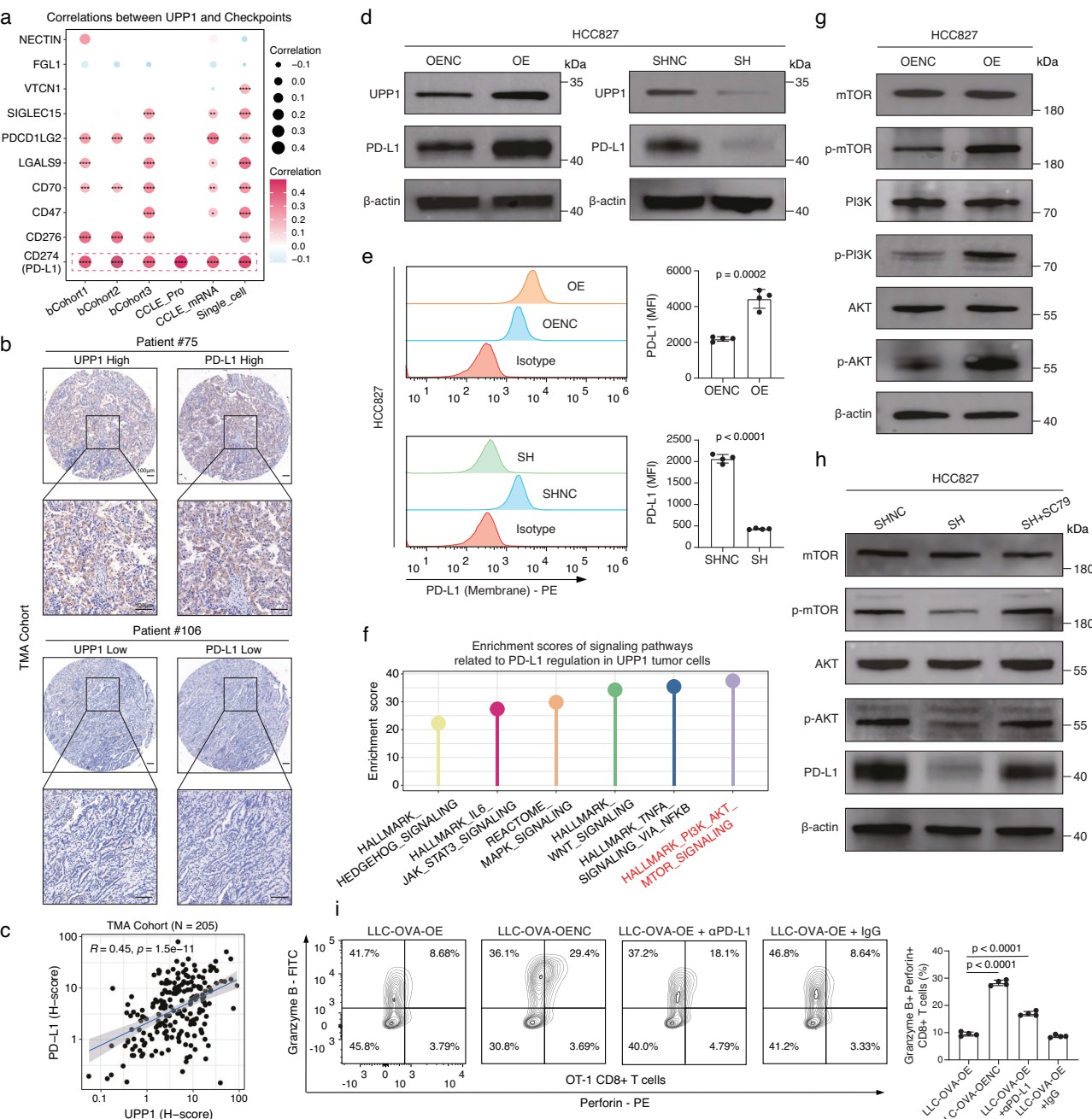

**Fig. 4 | UPP1 regulates PD-L1 expression via PI3K/AKT/mTOR pathway. a** Two-tailed Pearson correlation analysis between UPP1 and immune checkpoints across different datasets, including the scRNA-seq dataset (cells = 49,113), bulk datasets (bCohort1, $n = 559$; bCohort2, $n = 398$; bCohort3, $n = 572$), and CCLE mRNA ($n = 440$) and protein expression ($n = 37$) datasets. *$p < 0.05$, **$p < 0.01$, ***$p < 0.001$, ****$p < 0.0001$. **b** Representative images of IHC staining of UPP1 and PD-L1 in our TMA cohort from Zhongshan Hospital ($n = 205$). Scale bars = 100 μm. **c** Two-tailed Pearson correlation between UPP1 and PD-L1 in our TMA cohort ($n = 205$). Data are presented as mean ± SEM. **d** Western blot analysis of PD-L1 expression in HCC827 tumor cells with UPP1 overexpression (OE) or UPP1 knockdown (SH). Representative images shown ($n = 3$). **e** Flow cytometry analysis of PD-L1 expression on the cell surface of HCC827 tumor cells with UPP1 overexpression or UPP1 knockdown. Representative flow cytometry plots are shown ($n = 4$). Data are presented as mean ± SD. Statistical analysis was conducted using the two-tailed student's *t*-test. MFI, mean fluorescence intensity. **f** Functional enrichment analysis of six classical

signaling pathways known to regulate PD-L1 in UPP1[high] tumor cells. Pathways were ranked based on ssGSEA enrichment scores. **g** Western blot analysis of the levels of PI3K, AKT, and mTOR and phosphorylated PI3K, AKT, and mTOR in HCC827 tumor cells with UPP1 overexpression. Representative images shown ($n = 3$). **h** Western blot analysis of the levels of phosphorylated AKT and mTOR in HCC827 tumor cells with UPP1 knockdown and after the treatment of SC79 for 48 h (20 μM). Representative images shown ($n = 3$). **i** Flow cytometry analysis of immune effector molecules, Perforin and Granzyme B, in OT-1 CD8 + T cells after co-cultured with LLC-OVA UPP1-OE tumor cells, LLC-OVA UPP1-OENC tumor cells, and LLC-OVA UPP1-OE tumor cells with the treatment of PD-L1 antibodies (αPD-L1) or isotype control IgG. Representative flow cytometry plots are shown ($n = 4$). Data are presented as mean ± SD. Statistical analysis was conducted using one-way ANOVA with multiple comparisons. n denotes biologically independent samples. Source data are provided as a Source Data file.

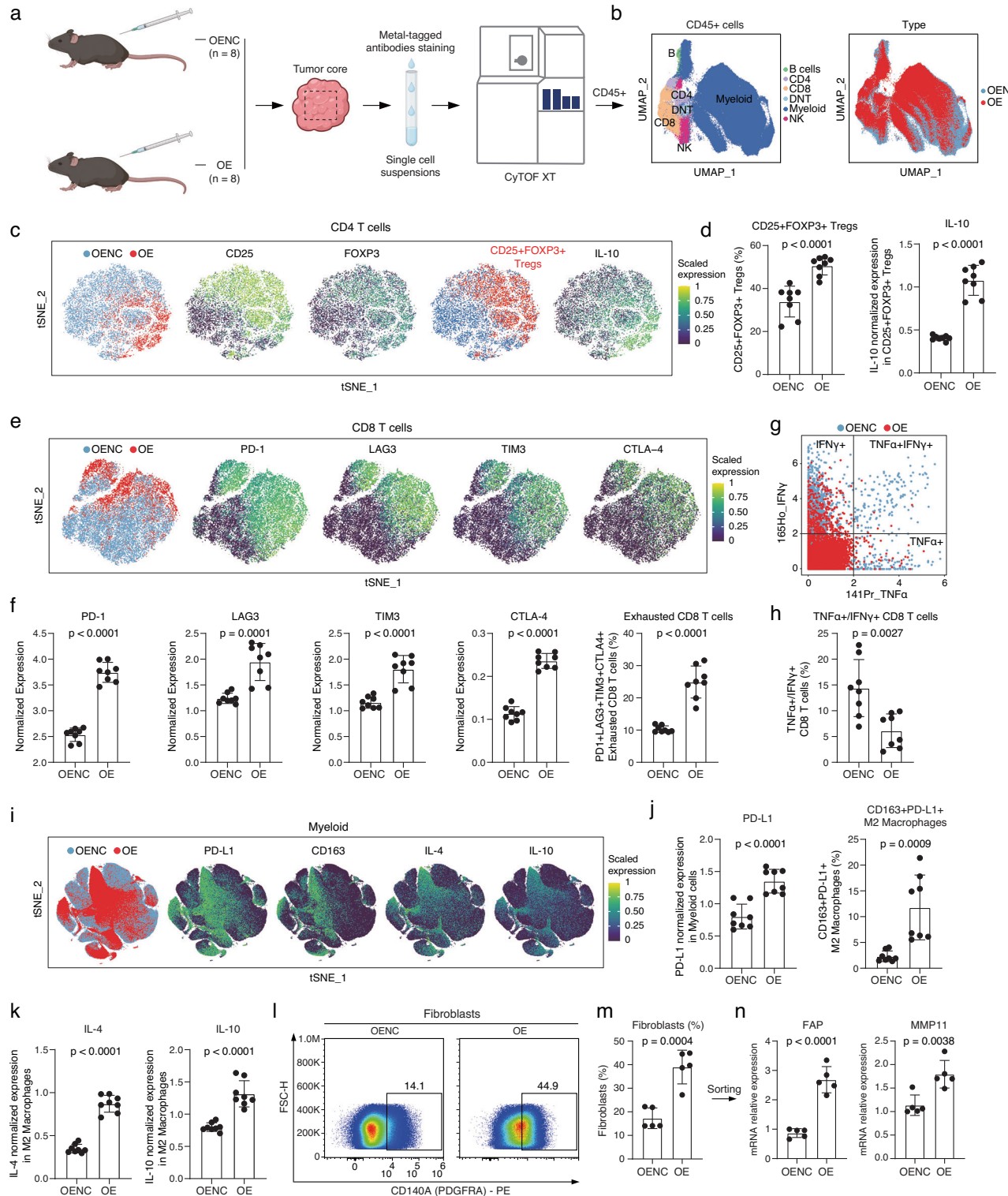

noticeably declined (Fig. 5g, h). In terms of myeloid cells, earlier studies have pointed out that the broad expression of PD-L1 on myeloid cells is also an important factor contributing to the immunosuppression of the TME and closely associated with unfavorable responses to immunotherapy[49]. Here, our CyTOF analysis showed a marked increase in the global PD-L1 expression on myeloid cells (Fig. 5i, j). Additionally, there was a significant enrichment of CD163 + PD-L1 + M2 macrophages within the UPP1-overexpressing group (Fig. 5j). These M2 macrophages concurrently exhibited elevated levels of immunosuppressive cytokines, IL-4 and IL-10 (Fig. 5k). Moreover, in the UPP1-

overexpressing group, there was a noticeable decline in the proportion of TNFα + /IFNγ+ effector NK cells and an increase in the proportion of PD-L1+ neutrophils (Supplementary Fig. 13a, b).

In addition, flow cytometry analysis was conducted to analyzed the fibroblasts within the TME. The UPP1-overexpressing group displayed a significant increase in the proportion of infiltrating fibroblasts (Fig. 5l-m and Supplementary Fig. 14). Notably, these fibroblasts exhibited high expression levels of FAP and MMP11 (Fig. 5n).

In summary, the CyTOF analysis substantiated that the high expression of UPP1 in tumor cells promoted the shaping an

**Fig. 5 | CyTOF analysis of the alterations in the characteristics of TME in UPP1-overexpressing tumors. a** Experimental flowchart, created with BioRender.com. **b** UMAP plot showing major cell populations. **c** tSNE plots of CD4 + T cells. From left to right: The distribution of CD4 + T cells from the UPP1-OE and UPP1-OENC groups; Expression of CD25 and FOXP3 in CD4 + T cells; The CD4 + T cells expressing both CD25 and FOXP3 are labeled as CD25 + FOXP3+ Tregs; Expression of IL-10 in CD4 + T cells. **d** The proportion of CD25 + FOXP3+ Tregs between the UPP1-OE and UPP1-OENC groups (left). The expression level of IL-10 in CD25 + FOXP3+ Tregs (right) (*n* = 8). **e** tSNE plots of CD8 + T cells. From left to right: The distribution of CD8 + T cells from the UPP1-OE and UPP1-OENC groups; Expression of PD-1, LAG3, TIM3, and CTLA4 in CD8 + T cells. **f** Immune checkpoint expression (left) and proportion of PD1 + LAG3 + TIM3 + CTLA4+ exhausted CD8 + T cells (right) in UPP1-OE vs. UPP1-OENC groups (*n* = 8). **g** TNFα + /IFNγ + CD8 + T cells distribution in UPP1-OE vs. UPP1-OENC groups. **h** Proportion of TNFα + /IFNγ + CD8 + T cells in UPP1-OE vs. UPP1-OENC groups (*n* = 8). **i** tSNE plots of the myeloid cells. From left to right: The distribution of myeloid cells from the UPP1-OE group and UPP1-OENC group; Expression levels of PD-L1, CD163, IL-4, and IL-6 in myeloid cells. **j** PD-L1 expression in all CD11b+ myeloid cells between the UPP1-OE and UPP1-OENC groups (left). The proportion of CD163 + PD-L1 + M2 macrophages between the UPP1-OE and UPP1-OENC groups (right) (*n* = 8). **k** IL-4 and IL-10 levels in CD163 + PD-L1 + M2 macrophages between the UPP1-OE and UPP1-OENC groups (*n* = 8). **l** LLC tumor cells (UPP1-OE or UPP1-OENC) were subcutaneously injected into C57BL/6 mice. Tumors were then collected for flow cytometry analysis to compare changes in the proportion of fibroblasts (*n* = 5). **m** Fibroblast infiltration proportion between the UPP1-OE and UPP1-OENC groups (*n* = 5). **n** Fibroblast sorted from (**m**) for RT-qPCR detection of FAP and MMP11 expression (*n* = 5). *n* denotes biologically independent samples. Data were represented as mean ± SD in **d, f, h, j, k,** and **m**; mean ± SEM in **n**. All Statistical analysis were conducted using the two-tailed student's *t*-test. Source data are provided as a Source Data file.

immunosuppressive TME, which was characterized by increased proportions of immunosuppressive Tregs and exhausted CD8 + T cells, along with elevated PD-L1 expression in myeloid cells, enrichment of M2 macrophages, and fibroblast infiltration.

### UPP1 inhibition improved the cytotoxicity of CD8 + T cells and sensitized anti-PD-L1 immunotherapy

Cytotoxic CD8 + T cells are essential for the elimination of cancer cells, and their activity and functionality can significantly impact the effectiveness of immunotherapies[50]. As the upregulation of UPP1 in tumor cells can significantly alter the functional state of CD8 + T cells, we next sought to determine whether inhibiting UPP1 expression in tumor cells could restore the cytotoxicity of CD8 + T cells and enhance the sensitivity to immunotherapy.

The results showed that the control LLC cells undergoing the anti-PD-L1 immunotherapy did not result in a significant inhibition of tumor growth, indicating that the tumor cells were relatively resistant to anti-PD-L1 immunotherapy. However, upon inhibiting UPP1 expression, tumor proliferation was suppressed (Fig. 6a–d). Notably, the administration of anti-PD-L1 immunotherapy to mice with UPP1-inhibited tumors showed a further increase in the inhibitory effects on tumor growth, suggesting that inhibiting UPP1 expression increased the sensitivity of tumors to anti-PD-L1 immunotherapy.

Subsequent flow cytometry analyses of the infiltrating CD8 + T cells in tumors observed that inhibiting UPP1 expression in tumor cells increased the proportion of tumor-infiltrating CD8 + T cells. In addition, the treatment of anti-PD-L1 immunotherapy to mice on this basis could further enhance the infiltration of CD8 + T cells (Fig. 6e). Functional assessment of these CD8 + T cells revealed that UPP1 inhibition enhanced their cytotoxic capabilities. Remarkably, combining UPP1 inhibition with anti-PD-L1 immunotherapy could further significantly enhance the cytotoxic capabilities of CD8 + T cells (Fig. 6f and Supplementary Fig. 15a). On the other hand, we also transplanted LLC-OVA tumor cells into OT-1 mice. Similarly, inhibiting UPP1 expression on tumor cells significantly increased the ability of CD8 + T cells to eliminate tumors (Supplementary Fig. 15b–d). These findings suggested that suppressing UPP1 in tumor cells not only improved the cytotoxicity of CD8 + T cells but also enhanced the sensitivity to anti-PD-L1 immunotherapy.

### Bioinformatics-based screening for potential therapeutic agents suitable for tumors with high UPP1 expression

Drug repurposing based on the molecular characteristics of patients has been reported in numerous studies and has demonstrated its reliability[51–53]. To explore if tumors with relatively higher UPP1 levels might be more sensitive to certain existing drugs, we conducted a bioinformatics-based screening (Supplementary Fig. 16a). Gene expression data and associated drug response data of hundreds of cancer cell lines from CTRP, GDSC, and PRISM pharmacogenomic

datasets were collected (Fig. 7a). These data were then used to infer the drug sensitivity data for LUAD patients based on the previously reported methods[54,55]. Next, LUAD patients were categorized into UPP1 high expression and UPP1 low expression groups based on the UPP1 expression levels (Fig. 7b) and the drug sensitivity of the UPP1 high expression group and the UPP1 low expression group was compared (Fig. 7c). By integrating the results from the CTRP, GDSC, and PRISM datasets, 11 potential drugs were identified that might be more effective for tumors with high UPP1 levels (Fig. 7d). On the other hand, we compared the drug sensitivity differences between tumor cells with high and low UPP1 expression across the three datasets (Fig. 7e). After integrating the identified 11 potential drugs with the drugs that UPP1 high tumor cells were relatively sensitive to, 3 potential drugs were selected, including Bosutinib, Dasatinib, and Erlotinib. It was worth mentioning that a recent study by Gonçalves et al. that analyzed the Drug-Protein associations also highlighted the correlation between Dasatinib and UPP1[56].

Next, we conducted relative sensitivity experiments on these three drugs using tumor cells with overexpressed UPP1 and their control group cells. The results showed that tumor cells with high UPP1 expression were relatively more sensitive to both Bosutinib and Dasatinib (Fig. 7f). Subsequently, we established patient-derived organoids (PDOs) from 6 LUAD patients to further assess the efficacy of Bosutinib and Dasatinib (Fig. 7g, Supplementary Fig. 17, and Supplementary Data 5). The results showed that LUAD patients with higher UPP1 expression levels were more responsive to Bosutinib and Dasatinib (Fig. 7h). On top of this, mice experiments were further conducted. We observed that both Bosutinib and Dasatinib could inhibit tumor growth in both the UPP1-overexpression and control groups (Supplementary Fig. 16b–f). Notably, the degree of tumor growth inhibition was more prominent in the UPP1-overexpression group when treated with Bosutinib and Dasatinib (Fig. 7i). In summary, both in vitro PDO models and in vivo murine models consistently demonstrated that tumors with high UPP1 expression were relatively more responsive to Bosutinib and Dasatinib.

## Discussion

In this study, an integrative approach that combined scRNA-seq with bulk data analysis in LUAD led to the identification of a specific tumor cell subpopulation characterized by elevated UPP1 expression. Notably, this subpopulation was associated with immunosuppressive TME and poorer patient outcomes, underscoring the potential prognostic value of UPP1 expression in LUAD.

The spatial distribution of these cells, as determined through mIF staining and AI-driven image analysis, revealed their predominant localization at the tumor margins. Interestingly, their relative proximities to the four types of immunosuppressive-related cell components in the TME indicated a complex interplay with various immune and stromal components. FOXP3+ Tregs, a subset of CD4 + T cells, are

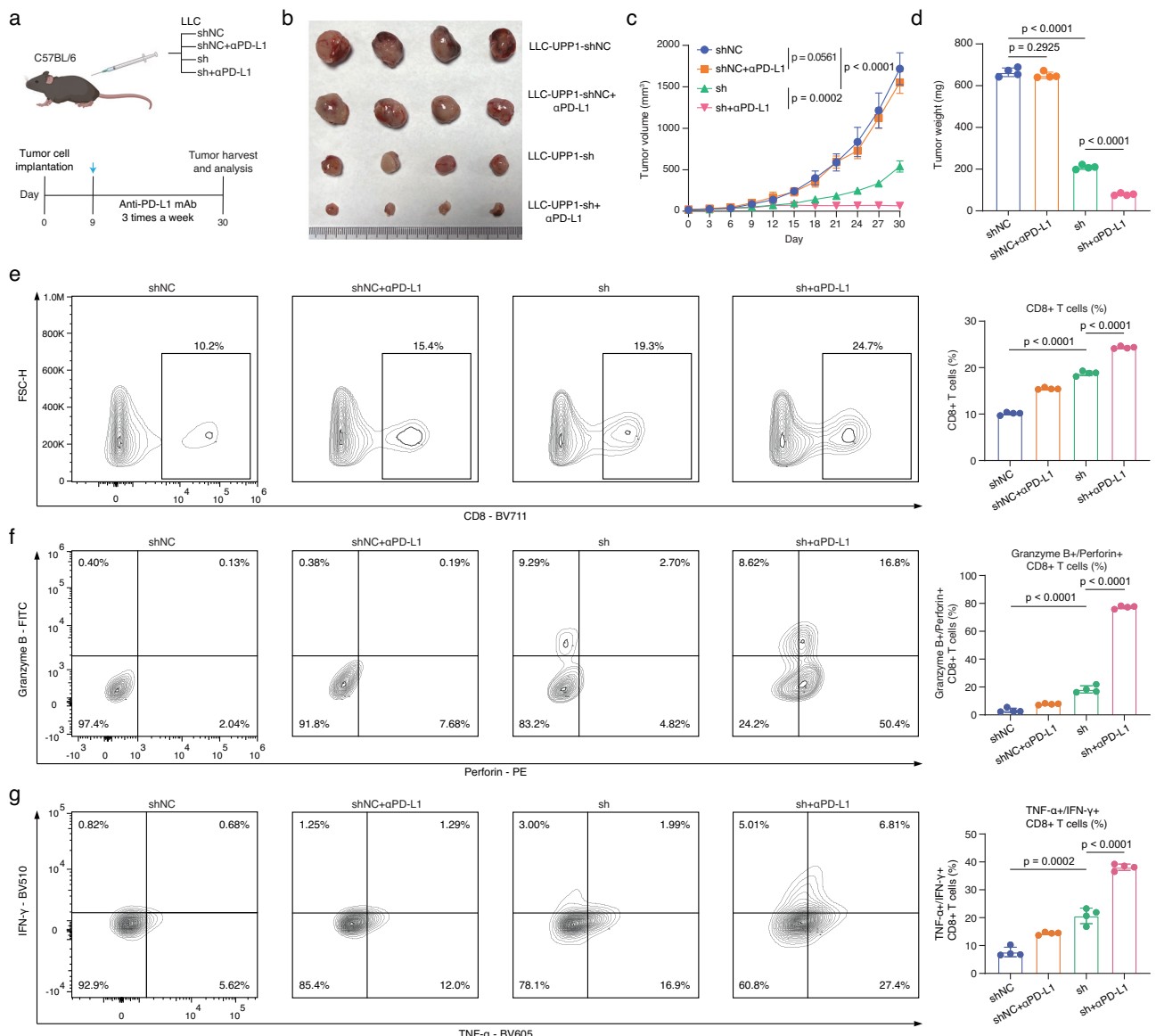

**Fig. 6 | UPP1 inhibition improved the cytotoxicity of CD8 + T cells and sensitized anti-PD-L1 immunotherapy. a** Experimental workflow, created with BioRender.com. UPP1-knockdown LLC tumor cells (sh) and control cells (shNC) were subcutaneously implanted into C57BL/6 mice with/without the treatment of with anti-PD-L1 antibodies (αPD-L1). **b** Tumors harvested from mice bearing LLC-UPP1-shNC tumor cells, LLC-UPP1-sh tumor cells, LLC-UPP1-shNC tumor cells treated with anti-PD-L1 antibody, and LLC-UPP1-sh tumor cells treated with anti-PD-L1 antibody (*n* = 4). **c** Tumor growth curves (*n* = 4). Statistical analysis was performed using two-way ANOVA with multiple comparisons. **d** Tumor weight at day 30 (*n* = 4). Statistical analysis was performed using one-way ANOVA with multiple comparisons. **e** Flow cytometry analysis of the proportion of CD8 + T cells infiltrated in the tumors (*n* = 4). Statistical analysis was performed using one-way ANOVA with multiple comparisons. **f** Flow cytometry analysis of immune effector molecules, Perforin and Granzyme B, in CD8 + T cells (*n* = 4). Statistical analysis was performed using one-way ANOVA with multiple comparisons. **g** Flow cytometry analysis of immune effector molecules, TNFα+ and IFNγ + , in CD8 + T cells (*n* = 4). Statistical analysis was performed using one-way ANOVA with multiple comparisons. n denotes biologically independent samples. Data are presented as mean ± SD in **c**–**g**. Source data are provided as a Source Data file.

known for their immunosuppressive roles in the TME, often inhibiting the activity of effector T cells and promoting tumor growth[57]. Their close association with the UPP1[high] tumor cells suggested a potential immunosuppressive niche, which might be conducive to tumor survival and evasion from immune surveillance; Similarly, the observed proximity of these cells to MMP11+ fibroblasts suggested a dynamic interplay, with fibroblasts expressing high-level MMP11 playing a pivotal role in extracellular matrix degradation[58]; The association of UPP1[high] tumor cells with SPP1+ macrophages further added to the complexity of the TME. SPP1 is often associated with pro-tumorigenic roles, with macrophages expressing SPP1 promoting tumor growth, angiogenesis, and metastasis[59]; Lastly the presence of exhausted

CD8 + T cells underscored a compromised anti-tumor immune response[60]. These associations between UPP1[high] tumor cells and various immunosuppressive components in the TME highlighted the potential role of UPP1 in shaping an immunosuppressive microenvironment in LUAD.

UPP1 has been implicated in the salvage pathway of pyrimidine metabolism, catalyzing the reversible phosphorylation of uridine to uracil[11]. However, its role in tumorigenesis and tumor progression has only recently come to light. In the context of LUAD, the elevated expression of UPP1 could bolster glycolytic metabolism in cancer cells, potentially enhancing tumor growth and reducing their susceptibility to glycolysis inhibitors[13]. Moreover, a recent study provided further

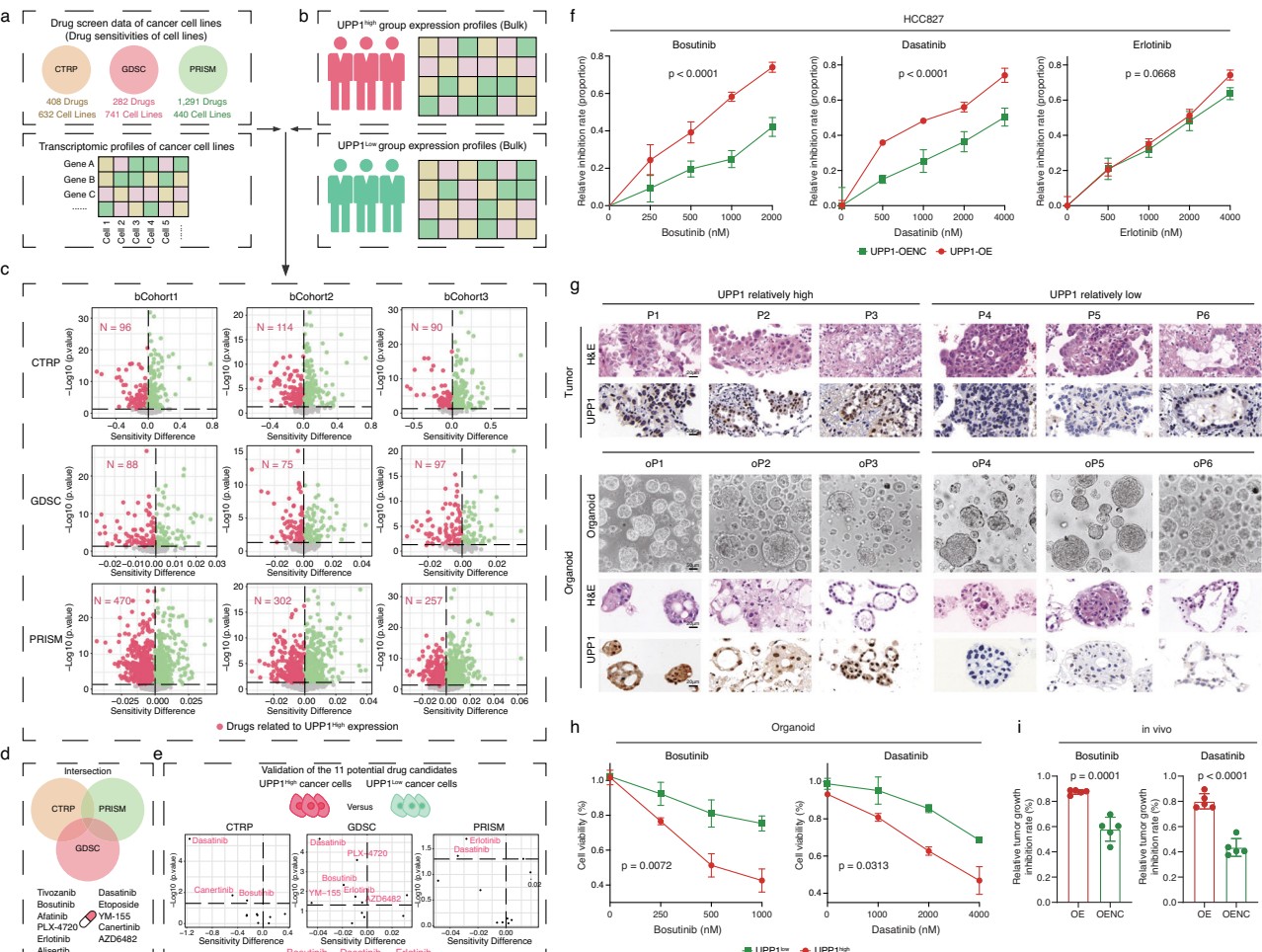

**Fig. 7 | Bioinformatics-guided drug screening and validation. a** Drug screen data and associated transcriptomic data from three datasets (CTRP, GDSC, and PRISM) were included. **b** LUAD patients were classified into UPP1 high and UPP1 low expression groups based on the median value of UPP1. **c** Based on cancer cell line data, the LUAD patients' sensitivities to various drugs were estimated. The x-axis represents the differences in IC50 values. The y-axis represents p-values. The red numbers indicate the number of drugs that were identified as relatively more sensitive in the UPP1 high-expression group. **d** Integration of the drugs identified from (**c**). **e** Cancer cell lines were categorized into UPP1 high and UPP1 low expression groups based on the median value of UPP1, differential drug analysis was conducted. The x-axis represents the differences in IC50 values. The y-axis represents p-values. Drugs showing increased sensitivity in the UPP1 high-expression group are marked in red. The identified drugs were combined with 11 drugs found in (**d**). **f** HCC827 UPP1-OE and control tumor cells were separately treated with Bosutinib, Dasatinib, and Erlotinib. Cell viability was assessed using the CCK8 assay (*n* = 5). **g** Establishment of PDOs. LUAD organoids were categorized into UPP1 high (*n* = 3) and UPP1 low (*n* = 3) groups. Images of organoids in bright field, Scale bars = 50 μm; for others, Scale bars = 20 μm. Representative images shown. **h** PDOs were separately treated with Bosutinib and Dasatinib. Cell viability was assessed using the CCK8 assay (*n* = 3). **i** LLC tumor cells (UPP1-OE or UPP1-OENC) were subcutaneously injected into C57BL/6 mice. Subsequently, the mice were treated with Bosutinib or Dasatinib, and the degree of tumor inhibition was compared (*n* = 5). The relative tumor growth inhibition rate was calculated based on the reduction in tumor size after drug treatment compared to the tumor size in the untreated UPP1-OE or UPP1-OENC groups. n denotes biologically independent samples. Data were represented as mean ± SD in **f**, **h**, and **i**. Two-tailed Wilcoxon rank-sum test was used in **c** and **e**. two-way ANOVA in **f** and **h**. Two-tailed student's *t*-test in **i**. Source data are provided as a Source Data file.

insights into the role of UPP1 in cancer, especially under glucose-restricted conditions. The study found that pancreatic ductal adeno-carcinoma (PDA) cells, when deprived of glucose, leveraged uridine-derived ribose as an alternative fuel source[12]. Notably, in vivo studies revealed that UPP1 knockout (KO) tumors showed reduced vessel density and increased anti-tumor T cell (CD8 T cells) infiltration[12]. These phenotype changes suggested that UPP1 may play a multi-faceted role in tumor biology, not only influencing metabolic pathways but also modulating the TME and immune response. Insights from Wang et al. highlighted the positive association of UPP1 with immune and inflammatory responses in glioma[14]. Specifically, UPP1 exhibited a strong correlation with MHC-II and LCK, signifying its relations with antigen-presenting cells and T cells. Additionally, UPP1 was found positively correlated with various immune checkpoint members,

underscoring its potential oncogenic role in glioma by modulating the tumor-related immune response.

Here, we approached from a different angle to explore the relationship between UPP1 and tumor immunity. Using the high-throughput cytokine detection array, we observed that upregulation of UPP1 in tumor cells resulted in a significant rise in various cytokines. Particularly, there was a notable upregulation of TGF-β1, GM-CSF, IL-1β, IL-6, IGFBP-3, CXCL5, CCL20, and VEGF. TGF-β1, known for its immu-nosuppressive properties, saw the most significant increase, hinting at a potential pathway for tumors with high UPP1 expression to influence immune surveillance[36]. Subsequently, our co-culture experiments revealed a pivotal role for TGF-β1 in the context of UPP1-overexpressing tumor cells. Elevated UPP1 expression drove immune cells towards suppressive phenotypes, with CD4 + T cells

differentiating into Tregs and CD8 + T cells adopting an exhausted state. Additionally, macrophages shifted towards an M2 phenotype, and fibroblasts exhibited pro-tumor characteristics in the presence of UPP1-overexpressing cells. While these phenotype changes were notably curtailed by TGF-β1 neutralizing. These findings highlighted the UPP1-TGF-β1 axis may as a central modulator of the TME, presenting potential therapeutic implications.

In addition to the influence of TGF-β1, the upregulation of other cytokines in UPP1-overexpressing tumor cells hints at a broader role for UPP1 in modulating tumor immunity. For instance, GM-CSF, while essential for granulocyte and macrophage differentiation and proliferation, can foster an immunosuppressive environment in the context of tumors by promoting the development of myeloid-derived suppressor cells (MDSCs)[61]; Further, IL-1β, a potent pro-inflammatory cytokine, may indirectly subvert anti-tumor immunity by fostering a chronic inflammatory environment, leading to the recruitment and activation of immunosuppressive cells, such as regulatory T cells and MDSCs[62]; Elevated levels of CXCL5 can enhance tumor growth, metastasis, and angiogenesis, often indicating a poor prognosis[63]. Meanwhile, CCL20 can draw regulatory T cells to the tumor site, aiding in immune evasion[64]; Finally, beyond its well-known role in angiogenesis, VEGF may also exert direct immunosuppressive effects[65]. It can impede the maturation and function of dendritic cells, crucial for initiating T-cell responses, and promote the recruitment and expansion of immune-suppressive regulatory T cells and MDSCs[66]. However, it is important to note that a more thorough mechanistic exploration is required to understand how UPP1 influences the expression of these cytokines.

On the other hand, our findings also highlighted a regulatory mechanism wherein UPP1 modulated the expression of the immune checkpoint PD-L1 via the PI3K/AKT/mTOR pathway. And increased PD-L1 levels impacted the functional activity of CD8 + T cells. This is consistent with prior studies that have emphasized the immunosuppressive role of PD-L1, especially in dampening T cell-mediated anti-tumor responses[67]. The ability of UPP1-overexpressing tumor cells to resist T cell-mediated elimination, which can be partially reversed by PD-L1 blockade in vitro, further underscored the clinical relevance of the UPP1-PD-L1 axis. Given the current interest in immune checkpoint blockade therapies, understanding such regulatory mechanisms may offer potential avenues for therapeutic interventions, enhancing the efficacy of treatments targeting the PD-L1 pathway.

Our results indicated that inhibiting UPP1 expression significantly reduces PD-L1 expression. Intriguingly, after suppressing UPP1 expression, there is an observed increase in tumor sensitivity to PD-L1 blockade in vivo. This phenomenon might be partly attributed to the changes in the tumor microenvironment following the alteration in UPP1 expression. For instance, we found that the suppression of UPP1 led to an increased infiltration of CD8 + T cells. Previous research has indicated that a reduction in CD8 + T cell infiltration is one of the reasons for the poor efficacy of PD-L1 monoclonal antibodies[68,69]. Thus, the observed relative increase in CD8 + T cell infiltration upon UPP1 suppression may contribute to a heightened immunotherapeutic response. Additionally, it was also found that tumors with higher UPP1 expression showed a significant increase in PD-L1 expression in macrophages. Previous research has indicated that PD-L1 expression in other immune cells such as macrophages is also crucial for the effectiveness of immunotherapy[70,71]. The overall increase in PD-L1 expression due to elevated UPP1 may also be a contributing factor affecting the efficacy of PD-L1-targeted treatments. On the other hand, while our study primarily focused on the association between UPP1 and PD-L1, it is possible that UPP1 expression may also be related to the activation and upregulation of other immune checkpoints, such as CD276, CD70, CD47, and LGALS9. The upregulation of these checkpoints can affect the efficacy of PD-1/PD-L1 immunotherapy. For instance, previous studies have shown that combined application of CD47 with PD-1/PD-L1 blockade can result in stronger anti-tumor effects[72,73]. This suggests that single checkpoint pathway inhibition might not be sufficient. These insights indicate that UPP1 may also be implicated in additional critical immune escape mechanisms. Hence, our study only partially explores the mechanisms by which UPP1 participates in immune suppression. Further, more in-depth research is needed to systematically reveal the core and complete mechanisms of UPP1 in immune invasion.

With CyTOF analysis, we further provided a deeper insight into role of UPP1 within the TME in vivo. The UPP1-overexpressing group manifested a pronounced immunosuppressive signature, with a notable rise in CD25 + FOXP3+ Tregs and their associated immunosuppressive cytokine, IL-10. This was complemented by the evident exhaustion of CD8 + T cells, marked by the upregulation of various immune checkpoints. The myeloid landscape also revealed enhanced PD-L1 expression and a rise in CD163 + PD-L1 + M2 macrophages, both accompanied by elevated levels of immunosuppressive cytokines. Additionally, the increased fibroblast infiltration in the UPP1-overexpressing group hinted potential role for UPP1 in stromal remodeling. Collectively, these findings from CyTOF analysis provided substantial evidence supporting UPP1's pivotal role in shaping an immunosuppressive TME.

Consistent with the findings of Nwosu et al., our study also observed that inhibiting UPP1 expression led to an increased infiltration of CD8 + T cells within the tumor[12]. This enhanced presence of effector T cells suggested a more active and potentially effective anti-tumor immune response. Building on this, the administration of anti-PD-L1 treatment further augmented the infiltration of CD8 + T cells and bolstered the expression of cytotoxic molecules. This synergy between UPP1 inhibition and PD-L1 blockade underscored the significance of UPP1 as a modulator of the tumor immune microenvironment. The heightened responsiveness to immunotherapy in the absence of UPP1 further emphasized its role in immune evasion mechanisms employed by tumors. Thus, our findings highlight the potential therapeutic value of targeting UPP1 in combination with immune checkpoint blockade, offering a potential avenue for enhancing the efficacy of cancer immunotherapies.

Lastly, we sought to identify whether there were certain drugs that might be more suitable for UPP1 high tumors. Based on the bioinformatics-based screening and validations using both in vitro and in vivo models, we revealed a notable association between high UPP1 expression and increased sensitivity to Bosutinib and Dasatinib. Bosutinib is a selective inhibitor designed to target the Src family kinases and has been approved for the treatment of chronic myeloid leukemia (CML) in adults[74,75]. Dasatinib, on the other hand, has a broader range, inhibiting not just the Src family kinases but also several other kinases[76,77]. However, their application in solid tumors, such as LUAD, has been limited. This limitation can be attributed to the unique molecular characteristics of solid tumors compared to hematological malignancies and concerns about potential off-target effects and associated toxicities of these TKIs in solid tumors[76]. The Src family kinases are involved in various cellular signaling pathways, including the PI3K/AKT/mTOR pathway[78,79]. When activated, they can enhance the signaling through this pathway. Given UPP1's role in activating the PI3K/AKT/mTOR pathway, it's possible that inhibiting Src family kinases with drugs like Bosutinib and Dasatinib could interfere with the signaling initiated by UPP1. This might provide some explanations into the observed sensitivity of UPP1 high tumors to these drugs. However, whether these drugs can be effective clinically for patients with high UPP1 expression requires further investigation. This analysis, while preliminary, might offer insights for considering UPP1 in future treatment strategies and drug development.

## Methods

### Ethics

This study was approved by the ethics committee of Zhongshan Hospital, Fudan University (B2021-128), and conducted in accordance with the principles of the Declaration of Helsinki. All participants from Zhongshan hospital donating surgical tissues provided written informed consent. All diagnoses were confirmed by histological reviews by qualified pathologists.

### Single-cell RNA sequencing data collection

We conducted a PubMed search for articles published up to 2021 using the keywords "lung adenocarcinoma" and "single-cell RNA sequencing." Each identified article was rigorously reviewed, and only those offering public access to either raw or processed single-cell RNA sequencing data were selected. From this search, we incorporated five scRNA-seq LUAD datasets (Supplementary Data 1) into our study: He et al. (CRA001963)[80], Kim et al. (GSE131907)[81], Laughney et al. (GSE123904)[82], Wu et al. (GSE148071)[83], and Xing et al. (HRA000154)[84]. The datasets from He et al. and Xing et al. were presented as raw FASTQ files, while the datasets from the other three studies were in the form of pre-processed cell matrix files. The Cell Ranger toolkit (version 6.1.2), provided by 10x Genomics, was used to align the reads from raw files to the human reference genome (GRCh38) to obtain the cell matrix files.

### Single-cell RNA sequencing data processing

The cell matrix file of each sample in each single-cell dataset was separately imported into R using the CreateSeuratObject function in Seurat (version 4.0.5). Cells that had more than 200 genes detected and a total transcript count exceeding 1000, but not surpassing 30,000 were chosen for further analysis[28,80–82,85]. Additionally, cells with more than 20% of their transcripts coming from mitochondrial genes were considered potentially apoptotic and thus excluded. Then, doublets were removed using DoubletFinder with default settings for each sample[86]. Moreover, a total of 1514 genes associated with mitochondria (50 genes), heat-shock protein (178 genes), ribosome (1253 genes), and dissociation (33 genes) were excluded to avoid unexpected noise and expression artifacts caused by dissociation[28]. After filtering out the low-quality cells, the first round of integration and batch-effect correction was conducted by integrating the samples within each single-cell dataset using Harmony[15]. Then, all five single-cell datasets were further integrated and batch effects were adjusted again using Harmony. Finally, a meta single cell cohort including 117 patients and 377,614 cells was used for subsequent analysis.

### Unsupervised dimensional reduction and clustering of major cell populations

For the identification of major cell populations, the meta dataset was firstly normalized using the NormalizeData function in Seurat (version 4.0.5)[16]. Then, the top 2000 variable features were selected based on the variance stabilizing transformation (VST). Following this, data was scaled to center each gene's expression measurements with a mean of zero and a standard deviation of one. This was followed by a dimensionality reduction using principal component analysis (PCA). Based on the harmonized dimensions, nearest neighbors were identified, and cell clusters were then identified at a resolution of 0.8. The dimensionality of was reduced using UMAP.

We then utilized the FindAllMarkers function in Seurat to identify markers for each cell cluster. In parallel, we also employed singleR for cell cluster identification[87]. We then made a comprehensive determination of the major cell populations by integrating the well-known marker genes, the marker genes for each cell cluster obtained through FindAllMarkers, and the cell clusters recognized by singleR. Well-known marker genes include: CD3D, CD8A, CD4, FOXP3, PDCD1, GZMB, NK7G and KLFR1 for CD8 + T, CD4 + T, and NK cells; CD79A, CD79B, IGHG1, and JCHAIN for B cells and plasma cells; CD14, FCGR3A, CD68, CD163, MARCO, CD1C, LAMP3, TPSAB1 for myeloid lineage (macrophages, dendritic cells, and mast cells); ACTA2 and PECAM1 for fibroblasts and endothelial cells; OLIG1 for oligodendrocytes; and EPCAM for epithelial cells. Among these epithelial cells, malignant tumor cells were further distinguished from non-malignant cells by inferring large-scale copy-number variations (CNVs) of each cell using inferCNV[17,81].

### Clustering and annotation of cell sub-populations

Next, we performed a second round of clustering of major cell populations to further characterize sub-populations. Each major cell population was extracted and subjected to another round of normalization, variable feature selection, data scaling, dimensionality reduction. For the clustering of cells, we experimented with multiple resolution parameters ranging from 0.4 to 1.5[28]. For each of these parameters, we analyzed the number of cells and the UMAP distribution of cells in each cluster, as well as the differentially expressed genes of each cell cluster, aiming to select a resolution parameter that was relatively stable, minimized over-clustering, and preserved the biological significance of identified cells for further analysis. For the clustering of T and NK cells, we used a parameter of 0.9. Myeloid cells were clustered using a parameter of 0.5, B cells with a parameter of 0.4, and stromal cells with a parameter of 0.8. Finally, for tumor cells, we opted for a resolution of 0.6.

As a result, we identified 9 sub-populations of CD4 + T cells, 6 sub-populations of CD8 + T cells, 5 sub-populations of NK cells, 3 sub-populations of dendritic cells, 9 sub-populations of macrophages, 2 sub-populations of mast cells, 9 sub-populations of B cells, 8 sub-populations of fibroblasts, 6 sub-populations of endothelial cells, and 20 sub-populations of tumor cells. Next, we used the FindAllMarkers function to identify differentially expressed marker genes of these cell sub-populations (Supplementary Data 2). For the naming of these cell sub-populations, we adopted the "cluster_celltype_marker" format described in previously published studies[28,88,89]. For cell populations that distinctly expressed known markers, we used these markers to represent these cells. For instance, a CD4T cell population that highly expressed FOXP3 (a defining marker for Tregs) was named as 8_CD4T_FOXP3 (8 being the cluster number) (Refer to Supplementary Fig. 2). Conversely, for other cell populations where the highly differentially expressed genes did not correspond to traditional cell-defining markers, we named them based on the top differentially expressed gene. Cell populations presented in more than 10 tumor samples were included for subsequent analysis[28].

### LUAD bulk transcriptomic data collection and processing

Bulk transcriptomic data of LUAD patients with complete overall survival information were collected from Gene Expression Omnibus (GEO) and TGCA databases. These data were uniformly processed based on our previously described pipelines[51,90]. Specifically, datasets from GSE19188, GSE30219, GSE31210, GSE37745, and GSE50081, all derived from the Affymetrix Human Genome U133 Plus 2.0 Array platform, were integrated as the bulk cohort 1 (bCohort1), consisting of 559 LUAD patients. This integration involved processing the raw CEL files of these samples in a standardized manner using the RMA algorithm for normalization and background adjustment[91]. Following this, probes from the Affymetrix HG-U133 Plus 2.0 Array were linked to their corresponding genes via the GPL570 platform. The signals from related probe sets were then averaged to obtain a unique expression value for every gene, compiling expression data for a total of 21,755 genes. To ensure consistent data interpretation across datasets, batch correction was conducted using ComBat function from the sva R package[92]. GSE72094, consisting of 398 patients, was designated as bulk cohort 2 (bCohort2), and the 572 TCGA patients were defined as bulk cohort 3 (bCohort3).

## Linking single-cell RNA sequencing data to bulk transcriptomic data for the identification of prognostic-related tumor cell populations

The bulk gene expression data was first subjected to a correction for tumor purity. ESTIMATE method was used to calculate the tumor purity of bulk data[93]. Then, we adjusted the gene expression data for tumor purity using linear regression[94,95]. For each gene, a linear model was fitted with gene expression as the dependent variable and tumor purity as the independent variable. The residuals of the linear models, which represent the differences between the observed expression values and the fitted values predicted by the models, were calculated for each gene. This process adjusts the gene expression data for the confounding effects of tumor purity, allowing for more accurate downstream analyses.

To evaluate the prognostic associations of the tumor cell sub-populations, we used the single sample Gene Set Enrichment Analysis (ssGSEA) method[19] to quantify the relative cell abundances of these tumor cell sub-populations based on their differentially expressed marker genes (Log2FC > 1 and adjusted $p < 0.05$) in each of the aforementioned purity-adjusted bulk cohorts (bCohort1/2/3). The relative abundances of these cell populations were categorized into high and low groups using the survminer R package[28,51,52,96]. Subsequently, we performed the log-rank test to compare the overall survival differences between the high and low groups. Tumor cell sub-populations that showed an association a worse prognosis across all three bulk cohorts were given priority attention.

### Functional enrichment of UPP1[high] tumor cells

To characterize the functional features of UPP1[high] tumor cells, we conducted a functional enrichment analysis on this group of cells. The gene sets (Supplementary Data 3) used for the functional enrichment analysis were retrieved from MSigDB database (https://www.gsea-msigdb.org/gsea/msigdb), including the HALLMARK, KEGG, REACTOME, and Chemical and Genetic Perturbations (CGP) categories[97]. The AUCell method was used to calculate the (Area Under the Curve) AUC scores of the gene sets[98,99]. Subsequently, a differential gene-set analysis based on the AUC scores was conducted comparing the AUC score differences between UPP1[high] tumor cells and other tumor cell sub-populations. Multiple hypothesis testing was performed using Bonferroni correction, and terms of interests with adjusted $p$-values less than 0.05 were considered significant. The gene sets were then ranked based on their differential Log2AUC values (Supplementary Data 4).

To validate the expression of the UPP1 is a significant characteristic of this group of cells, we calculated the average score of the top five gene sets that were enriched in the HALLMARK, KEGG, REACTOME, and CGP categories for this group of cells. This average score served as a representation of the general level of enrichment of the biological processes within this UPP1[high] tumor cell population. Next, we conducted the correlation analyses between the average score and the expression of each of the top 10 most highly expressed genes in this group of cells, aiming to determine which gene's expression level most closely aligns with the functional characteristics of this UPP1[high] tumor cell population.

### Cell-cell interaction network analysis

To investigate the potential co-enrichment patterns of UPP1[high] tumor cells with other cells in the TME, we calculated the pairwise correlation values between the frequency of UPP1[high] tumor cells and other cell populations across tumor samples[28]. Correlation values were then ranked, with values over 0.3 deemed as significant. This analysis revealed that UPP1[high] tumor cells had significant correlations with CD4T_FOXP3, CD8T_LAG3_PDCD1, Mph_SPP1, and Fb_MMP11 cell clusters. The markers (Supplementary Fig. 6b) used for analyze the features of CD4T_FOXP3 and CD8T_LAG3_PDCD1 were collected from

ref. 89. The markers (Supplementary Fig. 6c, d) used for analyze the features of Mph_SPP1were collected from Cheng et al. and Bagaev et al.[88,100]. To determine the CAFs features of Fb_MMP11, a functional enrichment using GSVA method was conducted[35]. The gene sets (Supplementary Fig. 6e) used to characterize the features of CAFs[101] were collected from MSigDB database.

The relative abundances of CD4T_FOXP3, CD8T_LAG3_PDCD1, Mph_SPP1, and Fb_MMP11 in the bulk cohorts were also calculated based on their marker genes (adjusted $p < 0.05$) using the ssGSEA method, and their correlations with UPP1[high] tumor cell in the bulk cohorts were also analyzed. Meanwhile, the overall survival associations of CD4T_FOXP3, CD8T_LAG3_PDCD1, Mph_SPP1, and Fb_MMP11 cell clusters in the bulk cohorts were also conducted.

The cell-cell communication ligand-receptor pairs between UPP1[high] tumor cells and CD4T_FOXP3, CD8T_LAG3_PDCD1, Mph_SPP1, and Fb_MMP11 cell clusters were assessed using the CellPhoneDB analysis[33]. The gene expression matrix and metadata file, which includes cell type assignments, were prepared based on the single-cell RNA sequencing data. The cell types of interest (UPP1[high] tumor cells, CD4T_FOXP3, CD8T_LAG3_PDCD1, Mph_SPP1, and Fb_MMP11 cell clusters) were labeled in the metadata file. Then, the CellPhoneDB method was executed, utilizing the gene expression matrix and metadata file as input. This method conducted a statistical analysis to identify significant receptor-ligand pairs between the cell types. The output files generated by CellPhoneDB were used for analysis ($p < 0.05$).

### UPP1-related TME module score

The marker genes (Log2FC > 1 and adjusted $p < 0.05$) of UPP1[high] tumor cells, CD4T_FOXP3, CD8T_LAG3_PDCD1, Mph_SPP1, and Fb_MMP11 cell clusters were combined into a gene set. We then calculated the GSVA scores on the three bulk cohorts using this gene set, and labeled the resulting GSVA scores as the UPP1-related TME module score. Subsequently, the GSVA scores were categorized into high and low groups using the survminer R package. We then performed the log-rank test to compare the overall survival differences between the high and low groups. In addition, we compared the TME features of the high and low groups. These features were calculated using the GSVA method with the gene sets listed in Supplementary Data 3.

### LUAD patient samples from Zhongshan Hospital

Three different batches LUAD samples were collected from Zhongshan Hospital in the current study (Supplementary Data 5). The TMA consists of 205 LUAD patients who underwent surgical resection of pulmonary carcinoma between January 2013 and August 2013. The tumor stages were determined based on the latest edition of the American Joint Committee on Cancer (AJCC) TNM classification. The overall survival and recurrence-free survival times were tracked from the day of the resection surgery up to the day of death, relapse/metastasis, or the last follow-up in August 2018. Tissues for this cohort were preserved in FFPE blocks and were sliced and fixed on microscope slides for immunohistochemistry in this study; LUAD samples used for multiplex immunofluorescence (mIF) analysis comprises 15 LUAD samples collected from Zhongshan Hospital between 2021 and 2022. The fresh samples were fixed using 4% paraformaldehyde and later processed into paraffin blocks. Similar to the TMA cohort, tissues for this batch were preserved in FFPE blocks. These samples were then sliced and mounted onto microscope slides for the subsequent immunofluorescence analysis in this study; six LUAD samples used for the establishment of patient-derived organoids (PDOs) were collected from Zhongshan Hospital in 2022.

### Immunohistochemistry

The sections were heated at 65 °C for an hour to dissolve the embedded paraffin. This was followed by a dewaxing process, which involved

two 10-min immersions in xylene. Following dewaxing, the sections underwent rehydration via a series of ethanol washes with declining concentrations. For antigen retrieval, sections were immersed in sodium citrate buffer (Servicebio, G1206) and microwaved for two intervals of 8 min each. Following antigen retrieval, the sections were allowed to cool naturally until they reached room temperature. Subsequently, the sections were treated with a 0.2% Triton X-100 solution for 15 min for permeabilization. They were then blocked using Quick-Block™ Blocking Buffer (Beyotime, P0260) for 30 min to prevent non-specific antibody binding. The sections were incubated with primary antibodies at 4 °C overnight. After washing three times with PBS for 5 min each time, the sections were incubated with HRP-conjugated secondary antibodies for 1 h at room temperature. Target proteins were visualized using a diaminobenzidine (DAB) chromogen kit (Servicebio, G1212). Slides were counterstained with diluted hematoxylin for 3–5 min. Slides were scanned using Pannoramic MIDI (3DHISTECH, Hungary). For the quantification of IHC images, we used the AI-based IHC scoring software, Aipathwell, developed by Servicebio (Wuhan, China)[25,26]. This software generated four metrics: Positive Area (%), Mean Density, Area Density, and H-score (Supplementary Data 5). We here used the H-score for subsequent analysis. The following primary antibodies were used for IHC in this study: anti-UPP1 (Abcam, ab185680, 1:100) and anti-PD-L1 (Cell Signaling Technology, 13684, 1:100). The following secondary antibody was used: Anti-rabbit IgG HRP-linked-antibody (Servicebio, GB23303, 1:200).

## Cell lines

The cell lines HCC827 (CL-0094), LLC (CL-0140), HFL1 (CL-0106), and THP-1 (CL-0233) were purchased from Wuhan Procell Life Science and Technology Co., Ltd. (Wuhan, China). LLC-OVA cells (labeled with mCherry) were generously provided by Professor Guangchuan Wang (State Key Laboratory of Molecular Biology, Shanghai Institute of Biochemistry and Cell Biology, Center for Excellence in Molecular Cell Science, Chinese Academy of Sciences). STR authentication of cell lines was performed by vendors, HCC827, authenticated in October 2021; LLC, authenticated in October 2023; HFL1, authenticated in February 2023; THP-1, authenticated in February 2023, and cell lines were regularly tested for mycoplasma. Cancer cell lines were maintained in RPMI 1640 medium (Gibco, C11875500BT) supplemented with 10% fetal bovine serum (FBS) (BIOIND, 04-001-1ACS) and 1% penicillin/streptomycin. HFL1 cells were cultured in Ham's F-12K medium supplemented with 10% FBS (Wuhan Procell, CM-0106) and 1% penicillin/streptomycin. THP-1 cells were cultured in RPMI 1640 medium with 10% FBS, 0.05 mM 2-mercaptoethanol, and 1% penicillin/streptomycin.

## Multiplex immunofluorescence

To investigate the distribution of UPP1$^{high}$ tumor cells in TME and their spatial relationship with CD4T_FOXP3, CD8T_LAG3_PDCD1, Mph_SPP1, and Fb_MMP11 cells, we performed the multiplex immunofluorescence staining (mIF) for these five cell types. For each tumor sample, we prepared four serial sections. Each section was processed as described above for immunohistochemistry, which included paraffin dissolution, dewaxing in xylene, rehydration through ethanol washes, antigen retrieval, cooling, permeabilizatio, and blocking. The sections were incubated with primary antibodies at 4 °C overnight. After washing three times with PBS, the sections were incubated with secondary antibodies for 1 h at room temperature. Next, a second round of primary antibodies was applied and incubated for 1 h at room temperature, followed by a second round of secondary antibodies incubated for 1 h at room temperature. Following this, the sections were incubated with FlexAble labeled primary antibodies for 2 h at room temperature. The following antibodies were used for each staining:

For FOXP3 + CD4 + T cells and UPP1 tumor cells, we used anti-CD4 (Abcam, ab133616, 1:100) detected with anti-rabbit AF555 (Abcam,

ab150078, 1:1000), anti-FOXP3 (Abcam, ab191416, 1:100) detected with anti-rabbit AF647 (Beyotime, A0468, 1:500), and anti-UPP1 (Proteintech, 14186-1-AP, 1:100) combined with FlexAble 488 (Proteintech, KFA001, working solution). PanCK was detected using anti-PanCK (Proteintech, 26411-1-AP, 1:200) combined with FlexAble 750 (Proteintech, KFA004, working solution).

For MMP11+ Fibroblasts and UPP1 tumor cells, we used anti-α-SMA (Bioss, bsm-33187m, 1:500) detected with anti-mouse AF647 (Abcam, ab150115, 1:1000), anti-MMP11 (Abcam, ab119284, 1:100) detected with anti-rabbit AF555 (Abcam, ab150078, 1:1000), and anti-UPP1 (Proteintech, 14186-1-AP, 1:100) combined with FlexAble 488 (Proteintech, KFA001, working solution). PanCK was detected using anti-PanCK (Proteintech, 26411-1-AP, 1:200) combined with FlexAble 750 (Proteintech, KFA004, working solution).

For LAG3 + PD-1 + CD8 + T cells and UPP1 tumor cells, we used anti-PD-1 (CST, 86163 S, 1:100) detected with anti-rabbit AF647 (Beyotime, A0468, 1:500), anti-LAG3 (CST, 15372 S, 1:100) detected with anti-rabbit mCherry (Absin, abs50028, working solution), anti-CD8 (Proteintech, 66868-1-Ig, 1:200) combined with FlexAble 555 (Proteintech, KFA022, working solution), and anti-UPP1 (Proteintech, 14186-1-AP, 1:100) combined with FlexAble 488 (Proteintech, KFA001, working solution). PanCK was detected using anti-PanCK (Proteintech, 26411-1-AP, 1:200) combined with FlexAble 750 (Proteintech, KFA004, working solution).

For SPP1+ Macrophages and UPP1 tumor cells, we used anti-SPP1 (Proteintech, 22952-1-AP, 1:100) detected with anti-rabbit AF647 (Beyotime, A0468, 1:500), anti-CD68 (Abcam, ab213363, 1:100) detected with anti-rabbit AF555 (Abcam, ab150078, 1:1000), and anti-UPP1 (Proteintech, 14186-1-AP, 1:100) combined with FlexAble 488 (Proteintech, KFA001, working solution). PanCK was detected using anti-PanCK (Proteintech, 26411-1-AP, 1:200) combined with FlexAble 750 (Proteintech, KFA004, working solution).

After sequential reactions, slides were stained with DAPI (Servicebio, G1012) and scanned using Olympus VS200 (Olympus, Japan).

## Visiopharm anslysis

We used the Visiopharm software (https://visiopharm.com), which is a leading AI-driven precision pathology software, for mIF analysis. After uploading the images into the software interface, the regions of interest (ROIs) within the tissue sections were manually selected. For each individual image, we reviewed the distribution patterns of tumor cells and the corresponding stained cells and selected 3–5 areas enriched in tumor cells and their associated stained cells. Then, the software's AI-powered recognition algorithm was used to segment individual cells based on the nuclear staining of DAPI within the ROIs.

Following the segmentation process, we used the phenotype module within Visiopharm to automatically classify cells into distinct phenotypes. These phenotypes included UPP1$^{high}$, UPP1$^{low}$ tumor cells, and the associated stained cell populations, determined by the unique expression profile of markers within each cell.

Post-classification, we analyzed the spatial distribution of these phenotypes and calculated the spatial distances between UPP1$^{high}$ tumor cells/UPP1$^{low}$ tumor cells and associated cell populations within a 200μm radius within the selected ROIs. For enhanced visualization, color-coded overlays were applied to the original images, highlighting the location and distribution of the identified phenotypes.

## Generation of stable UPP1 overexpression and knockdown cancer cell lines

The full-length coding sequences for human UPP1 (NM_001287426) were cloned into the vector PLVX-IRES-Puro to generate stable overexpression cell lines. For the creation of mouse overexpression cell lines, mouse UPP1 (NM_001159401) was cloned into the vector pLKO.1-puro.

Short hairpin RNAs (shRNAs) were used for the knockdown process. Three different shRNAs for the human (human-shUPP1-1: GCTGAAAGTCACAATGATTGC; human-shUPP1-2: GACACAATTT CCCAGCCTTGT; human-shUPP1-3: GTGCTCCAACGTCACTATCAT) were cloned separately into the PLVX-IRES-Puro vector. For the mouse sequence, three different shRNAs (mice-shUpp1-1: GCTTCATC-CAACTTTCAAATC; mice-shUpp1-2: GCACTAGCACACACGATTTCC; mice-shUpp1-3: GGAAGGAATATCCCAACATCT) were individually cloned into the pLKO.1-puro vector.

All plasmids were verified by DNA sequencing. The lentiviruses containing the packaged plasmids were purchased from Tongji biotechnology company (Shanghai, China) and used to infect HCC827 (human) and LLC or LLC-OVA (mouse) cells for 48 h in the presence of 10 µg/ml polybrene (MCE, HY-112735). The stably transduced cell lines were then selected using puromycin (Beyotime, ST551). The efficiency of overexpression and knockdown was validated using Western blot and RT-qPCR.

## Western blotting

Cells were lysed using RIPA lysis buffer (Beyotime, P0013) containing protease and phosphatase inhibitors (Beyotime, P1050) on ice for 30 min. After this, the mixture was centrifuged at 12,000 g for 15 min at 4 °C, and the supernatant containing the proteins was carefully collected. Protein concentrations were measured using a bicinchoninic acid (BCA) assay (Beyotime, P0012), following the manufacturer's instructions. Equal amounts of proteins were then denatured by boiling at 100 °C in SDS-PAGE sample loading buffer (Beyotime, P0015) for 5 min.

The denatured proteins were then separated using YoungPAGE gels (GenScript, M00930) in MOPS running buffer. Following electrophoresis, the separated proteins were transferred onto PVDF membranes (Millipore, IPVH00010) using a wet transfer system. The membranes were then blocked using QuickBlock Blocking Buffer (Beyotime, P0252) to prevent non-specific binding. After blocking, the membranes were incubated overnight at 4 °C with primary antibodies. The next day, the membranes were washed to remove unbound primary antibodies and incubated with appropriate secondary antibodies for 1 h at room temperature. The protein bands were then visualized using the e-BLOT imaging system (Touch Imager, China).

The following primary antibodies were used for western blotting in this study: anti-human-UPP1 (Abcam, ab128854, 1:1000), anti-PD-L1 (Cell Signaling Technology, 13684, 1:1000), anti-β-Actin (Proteintech, HRP-60008, 1:2000), anti-SPP1 (Proteintech, 22952-1-AP, 1:1000), anti-GAPDH (Proteintech, HRP-60004, 1:5000), anti-MMP11 (Abcam, ab119284, 1:1000), anti-FAP (Abcam, ab207178, 1:1000), anti-PI3K (Cell Signaling Technology, 4257 S, 1:1000), anti-AKT (Cell Signaling Technology, 9272 S, 1:1000), anti-mTOR (Cell Signaling Technology, 2972 S, 1:1000), anti-PI3K p85 (phospho Y458) + PI3 Kinase p55 (phospho Y199) (Abcam, ab278545, 1:1000), anti-p-mTOR (phospho S2448) (Abcam, ab109268, 1:1000), anti-p-AKT (Santa Cruz, sc-293125, 1:200). The following secondary antibody was used: anti-rabbit-HRP (Cell Signaling, 7074 S, 1:2000) and anti-mouse-HRP (Jackson Immunoresearch, 115-035-003E, 1:10000).

## Quantitative real-time PCR (RT-qPCR)

The procedure for extracting total RNA from cells was conducted using the total RNA isolation kit (Vazyme, RC112-01), per the manufacturer's instructions. NanoDrop was utilized to assess the quality and concentrations of RNA in the samples. Next, each sample's total RNA, amounting to 1 ng, was converted into cDNA.

The removal of genomic DNA was carried out with a 4 × gDNA wiper Mix (Vazyme, R233-01-AB). The mixture, consisting of 12 µl of RNA and DNase/RNase-free ddH2O, along with 4 µl of gDNA wiper Mix, was incubated at 42 °C for 2 min. Subsequently, 4 µl of 5 × qRT SuperMix (Vazyme, R233-01-AC) was added to the mixture and

incubated at 50 °C for 15 min. The reaction was terminated by heating at 85 °C for 5 s.

The RT-qPCR mixture, totaling 10 µl, included 5 µl of 2 × Taq SYBR Green qPCR Mix (Vazyme, SQ101), 1 µl of primers, and 4 µl of diluted cDNA. The quantification of gene expression was executed using a BIO-RAD Real-time PCR System. The data was computed using the $2^{-\Delta\Delta CT}$ method, with GAPDH serving as the internal reference gene. The primers were synthesized by Tsingke Biotechnology Co., Ltd (Beijing, China). The primer sequences are provided in Supplementary Table 2.

## PCR

DNA was extracted from LLC-OVA cell lines utilizing the FastPure Cell/Tissue DNA Isolation Mini Kit (Vazyme, DC102-01). PCR amplifications were then conducted using specific primers, ensuring equivalent DNA quantities in each PCR reaction.

A 10 µl PCR mixture was prepared, comprising 1 µl of genomic DNA, 0.5 µl of each primer, 5 µl of 2 × Taq Master Mix (Vazyme, P112-01), and 3.5 µl of DNase/RNase-free ddH2O. The PCR parameters were set as follows: initial denaturation at 94 °C for 3 min, followed by 35 cycles of denaturation at 94 °C for 30 s, annealing at 60 °C for 35 s, and extension at 72 °C for 35 s. The reaction was concluded with a final extension at 72 °C for 5 min.

Following amplification, the PCR products were subjected to electrophoresis on a 1.0% agarose gel stained with Ultra GelRed (Vazyme, GR501-01) and visualized under ultraviolet light. The DL5000 DNA marker (Vazyme, MD102-01) was used to determine the molecular weight of the PCR products. The primers of OVA257-264[102] were synthesized by Tsingke Biotechnology Co., Ltd (Beijing, China). The sequences of the primers are listed in Supplementary Table 2.

## Cytokine-array analysis

The RayBio C-Series Human Cytokine Antibody Array C5 (RayBiotech, AAH-CYT-5-8) was used to assess the changes of 80 cytokines between UPP1 overexpressing tumor cells and control cells. A total of $1\times10^7$ cells were collected and subsequently lysed using the lysis buffer containing protease and phosphatase inhibitors on ice for 30 min. Following this, the samples were centrifuged at 14,000 g for 20 min, after which the supernatant was collected for further analysis.

The Antibody Arrays were removed from the plastic packaging and each membrane was placed into a well of the incubation tray. Each well was filled with 2 ml of Blocking Buffer and incubated for 30 min at room temperature, after which the blocking buffer was aspirated from each well. Each well was then filled with 1 ml of the sample and incubated overnight at 4 °C, then aspirated. The wells were then filled with 2 ml of 1X Wash Buffer I and incubated for 5 min at room temperature, a process repeated three times. This was followed by the addition of 2 ml of 1X Wash Buffer II into each well and incubated for 5 min at room temperature.

1 ml of the prepared Biotinylated Antibody Cocktail was then added to each well and incubated for 2 h at room temperature, after which the Biotinylated Antibody Cocktail was aspirated from each well. The membranes were then washed again. After that, 2 ml of 1X HRP-Streptavidin was added to each well and incubated for 2 h at room temperature. The membranes were then washed again.

The membranes were then transferred onto a sheet of chromatography paper. After that, equal volumes of Detection Buffer C and Detection Buffer D were mixed, and 500 µl of the Detection Buffer mixture was gently pipetted onto each membrane and incubated for 2 min at room temperature. Finally, the membranes were scanned using the e-BLOT imaging system (Touch Imager, China) and analyzed using the ImageQuant software (GE, USA).

## ELISA

Supernatants from UPP1-overexpressing tumor cells and control cells were analyzed using the human TGF-β1 ELISA kit (Proteintech,

KE00002), per the manufacturer's instructions. The TGF-β1 levels were detected using the SpectraMax ABS Plus microplate reader (MD, USA).

## Co-culture of UPP1-overexpressing tumors with CD4 + T cells, CD8 + T cells, THP-1-derived macrophages, and HFL1 fibroblasts

Peripheral Blood Mononuclear Cells (PBMCs) were isolated from blood samples of healthy volunteers using a density gradient centrifugation method with Ficoll-Paque Plus cell separation media (Cytiva, 17144003). CD4 + T cells were isolated from PBMCs using the CD4 + T Cell Isolation Kit (Miltenyi Biotec, 130-094-131) according to the manufacturer's instructions. The purity of CD4 + T cells was validated using flow cytometry; Untouched CD8 + T cells were purchased from iXCells Biotechnologies (10HU-024N); CD4 + T cells or CD8 + T cells were cultured in ImmunoCult T Cell Expansion Medium (STEMCELL, 10981) supplemented with 50 U/ml IL-2 (Peprotech, 200-02-50) and ImmunoCult CD3/CD28/CD2 T cell Activator (STEMCELL, 10970). The cells were maintained in the culture medium for 2 days before co-culture; THP-1 cells were differentiated into macrophages by treatment with PMA (100 nM) for 48 h.

A Transwell chamber with a 0.4 μm pore size permeable membrane (Corning, 3450) was utilized for the co-culture assays. $2.5 \times 10^5$ HCC827 tumor cells (UPP1-OE or UPP1-OENC) were placed in the top insert of the Transwell and co-cultured with CD4 + T cells, CD8 + T cells, THP-1-derived macrophages, and HFL1 fibroblasts, respectively. The number of cells in the lower chamber were as follows: $5 \times 10^5$ CD4 + T cells, $5 \times 10^5$ CD8 + T cells, $5 \times 10^5$ THP-1-derived macrophages, and $2.5 \times 10^5$ HFL1 fibroblasts. On the second day of co-culture, 10 μg/ml of TGF-β neutralizing antibody (BioXcell, BE0057) or isotype IgG was added to the upper chamber. The culture medium was replaced once at the midpoint of the co-culture period. For CD4 + T cells and CD8 + T cells, the co-culture continued for a total period of seven days. For THP-1-derived macrophages and HFL1 fibroblasts, the co-culture period was five days. CD4 + T cells and CD8 + T cells were analyzed using flow cytometry, THP-1-derived macrophages were analyzed using immunofluorescence, western blotting and RT-qPCR; fibroblasts were examined using both single-cell analysis and western blotting tests.

For the immunofluorescence of THP-1-derived macrophages, the cells were first fixed with 4% paraformaldehyde for 30 min. They were then permeabilized with PBS containing 0.2% Triton X-100 for 15 min at room temperature. The slides were blocked using QuickBlock™ Blocking Buffer (Beyotime, P0260) for 30 min at room temperature, followed by incubation with anti-SPP1 (Proteintech, 22952-1-AP) at 4 °C overnight. Goat Anti-Rabbit IgG H&L (Alexa Fluor 555) (Abcam, ab150078) was used to bind the primary antibodies at room temperature for 1 h. Following the staining of primary and secondary antibodies, DAPI (Servicebio, G1012) was used to stain the nuclei. The slides were then observed using CSU-W1 confocal microscopy (OLYMPUS, Tokyo, Japan) and analyzed with the Cell-Sens Application Suite Software.

## Flow cytometry

Flow cytometry was conducted using MA900 Multi-Application Cell Sorter (SONY, Japan). For flow cytometry analysis of cell samples, single-cell suspensions were prepared in PBS and blocked using Human TruStain Fc (Fc Receptor Blocking Solution) (Biolegend, 422302, working solution) or Mouse TruStain Fc PLUS (anti-mouse CD16/32) Antibody (Biolegend, 156604, working solution). For membrane antibody staining, the antibody mixture was added into the cell suspension and incubated at room temperature for 30 min. The cells were then washed with cold PBS and 5 μl of 7-AAD (Biolegend, 420403, working solution) was added. These prepared samples were then subjected to flow cytometry analysis.

For intracellular staining, surface marker staining was first performed at room temperature for 30 min. Subsequently, the cells were

fixed and permeabilized using the FIX & PERM Cell Permeabilization Kit (Invitrogen, GAS003). Then, intracellular antibody staining was conducted at room temperature for 2 h. The cells were then washed with cold PBS and subjected to flow cytometry analysis. For Foxp3 staining of Tregs, we used the eBioscience Foxp3 / Transcription Factor Staining Buffer Set (Invitrogen, 00-5523-00).

For flow cytometry analysis of mouse tumor tissue samples, the harvested tumors were first minced. Next, they were digested using RPMI 1640 medium supplemented with 1 mg/ml collagenase IV (Worthington, LS004188, for immune cell analysis) or collagenase II (Worthington, LS004176, for fibroblast analysis), and 10 ug/ml DNase I (Roche, 11284932001), at 37 °C on a shaking platform operating at 120 rpm. Then, the cell suspensions were passed through 70 μm cell strainers to prepare single-cell suspensions in PBS. They were then blocked using Mouse TruStain Fc PLUS (anti-mouse CD16/32) Antibody (Biolegend, 156604, working solution), and the staining process was conducted as per the aforementioned procedure. The antibodies used are as follows:

For CD4 + CD25 + FOXP3+ Tregs analysis, Anti-CD4-FITC (Invitrogen, 11-0048-42, 1:100), Anti-FOXP3-PE (Invitrogen, 12-4776-42, 1:20), and Anti-CD25-APC (Invitrogen, 17-0257-42, 1:20) antibodies were used.

For the analysis of exhausted CD8 + T cells, the antibodies used were Anti-CD8-PE (Biolegend, 344705, 1:100), Anti-LAG3-APC (Biolegend, 369211, 1:20), Anti-PD-1-FITC (Invitrogen, 11-9969-42, 1:20), and 7AAD (Biolegend, 420403, working solution).

For analyzing PD-L1 expression, the PE CD274 (PD-L1, B7-H1) (Invitrogen, 12-5983-42, 1:200) was used.

For evaluating the immune effector molecules of OT-1 CD8 + T cells against LLC-OVA tumor cells, the antibodies used were Anti-CD8-APC (Biolegend, 140410, 1:100), Anti-Perforin-PE (Invitrogen, 12-9392-80, 1:20), and Anti-Granzyme B-FITC (Biolegend, 372206, 1:20).

For the study of fibroblasts in tumors, the antibodies used were Anti-CD45-FITC (Invitrogen, 11-0451-81, 1:100), Anti-CD140-PE (Biolegend, 135905, 1:100), Anti-CD31-APC (Biolegend, 102410, 1:100), and 7AAD (Biolegend, 420403, working solution)[103].

For assessing the immune effector molecules of CD8 + T cells in tumors, the antibodies used were Anti-CD45-BV421 (Biolegend, 103134, 1:100), Anti-CD3-APC (Biolegend, 100326, 1:100), Anti-CD8-BV711 (Invitrogen, 407-0081-82, 1:100), Anti-IFN-γ-BV510 (Biolegend, 505841, 1:20), Anti-TNF-α-BV605 (Biolegend, 506329, 1:50), Anti-Perforin-PE (Invitrogen, 12-9392-80, 1:20), and Anti-Granzyme B-FITC (Biolegend, 372206, 1:20).

## Single-cell RNA sequencing of fibroblasts co-cultured with tumor cells

HFL1 fibroblasts co-cultured with HCC827 tumor cells (UPP1-OE or UPP1-OENC) were collected for single-cell RNA sequencing. The fibroblasts were digested using trypsin (Gibco, 25200072) and then passed through a 70-um strainer (Falcon, 352350) to obtain the single-cell suspensions. The cell suspensions were subjected to Chromium Next GEM Single Cell 5' Reagent Kits v2 with a cell recovery target of 10,000, following the manufacturer's instructions (10X Genomics). Libraries were sequenced on an Illumina NextSeq 2000 platform. The single-cell RNA sequencing raw data were processed based on the pipelines mentioned above.

To analyze the transformation of fibroblasts co-cultured with UPP1-OE into CAFs, we calculated the CAFs score for fibroblasts co-cultured with either UPP1-OE or UPP1-OENC using the AddModuleScore function in Seurat based on the canonical genes of CAFs (Supplementary Data 3). Subsequently, based on the marker genes of MMP11+ CAFs identified in Supplementary Fig. 6a, we calculated the MMP11+ CAFs scores for the co-cultured fibroblasts using also the AddModuleScore function in Seurat. Fibroblasts with an MMP11+ CAFs

score higher than the median value were categorized as MMP11+ CAFs-like fibroblasts, while those scoring below the median were defined as MMP11- fibroblasts (Fbs).

## Mice

Six-week-old female C57BL/6 and BALB/c nude mice were purchased from SPF Biotechnology Co., Ltd. (Beijing, China). Six-week-old female OT-1 mice were purchased from Cyagen Biosciences (Guangzhou, China). The mice were maintained in a specific-pathogen-free (SPF) environment. Previous research has indicated that there are no significant differences between male and female mice in the LLC cell tumor model[104]. During our preliminary experiments, we observed that male mice tended to engage in aggressive behavior post-transplantation, leading to the disruption of tumor growth due to fighting-related injuries. Female mice generally exhibit reduced fighting behaviors, which facilitates group housing and randomization[105], and helps in generating more consistent results, so we used female mice. In accordance with the guidelines and approvals granted by our ethics committee, the maximum allowable tumor size and burden for this study were set at a length and width not exceeding 2 cm and a volume limit of 2000 mm³. Tumor growth in the mice was monitored every three days. In accordance with the Institutional Animal Care and Use Committee (IACUC) guidelines, we employed the carbon dioxide (CO₂) inhalation method for the euthanasia of mice used in our study. Following the loss of consciousness, we confirmed the euthanasia with a secondary physical method, specifically cervical dislocation, to ensure a humane and ethical treatment of the animals as per the IACUC standards. All experimental procedures involving animals were approved by the Ethics Committee of Zhongshan Hospital, Fudan University.

## OT-1 CD8 + T cells killing assay

OT-1 splenic lymphocytes were isolated using the splenic lymphocytes isolation kit (Solarbio, P8860), following the manufacturer's instructions. After that, splenic lymphocytes were stained with Anti-CD3-APC (Biolegend, 100326) and Anti-CD8-BV711 (Invitrogen, 407-0081-82). Then, OT-1 CD8 + T cells were sorted and collected using MA900 Multi-Application Cell Sorter (SONY, Japan). The sorted cells were then expanded and activated for five days in RPMI 1640 containing 10% FBS, 5ug/ml IL-2 (Peprotech, 200-02-50), and 1ug/ml OVA257-264 peptide (Genscript, RP10611)[106]. After activation, the purity of CD8 + T cells was tested. Then, OT-1 CD8 + T cells and LLC-OVA target cells were co-culture at a 2:1 ratio overnight.

After the co-culture, the suspended CD8 + T cells and apoptotic unattached cells were gently washed away using PBS. Afterward, the surviving adherent tumor cells were fixed and stained with crystal violet solution (Beyotime, C0121). Images were captured and quantification was performed using Image J software. In addition, a flow cytometric analysis was carried out on the cells after the co-culture to determine the number and proportion of surviving tumor cells. DAPI was used to identify dead cells, and the surviving tumor cells were counted by identifying cells with negative DAPI staining and positive mCherry expression.

## Mice procedures

In all animal implantation experiments, a consistent quantity of $5 \times 10^5$ tumor cells was used. The cells were suspended in 50 μl of PBS and thoroughly mixed with 50 μl of Matrigel (Beyotime, C0383).

To evaluate the influence of UPP1 expression changes on the host's tumor immune regulation, a comparative experiment was first conducted. Four distinct cell groups - UPP1-overexpressing LLC tumor cells, UPP1-downregulated LLC tumor cells, and their respective controls - were prepared. These cells were subcutaneously implanted into C57BL/6 mice and nude mice ($n = 5$ per group).

For the CyTOF and fibroblast flow cytometry animal experiments, UPP1-overexpressing LLC tumor cells and control cells were subcutaneously implanted into C57BL/6 mice ($n = 8$ per group for CyTOF; $n = 5$ per group for fibroblasts).

In the anti-PD-L1 treatment animal experiment, UPP1-downregulated LLC tumor cells and control cells were implanted into C57BL/6 mice ($n = 4$ per group). Once tumors reached a volume of 100 mm³, InVivoMab Anti-mouse PD-L1 antibodies (BioXcell, BE0101, 200 μg/mouse) were administered via intraperitoneal injection three times per week for three weeks.

For the OT-1 mouse experiment, UPP1-overexpressing LLC-OVA tumor cells, UPP1-downregulated LLC-OVA tumor cells, and their respective controls were subcutaneously implanted into adult OT-1 mice ($n = 5$ per group).

For the drug treatment experiment, UPP1-overexpressing LLC tumor cells and their control cells were subcutaneously injected into C57BL/6 mice. When the tumor volumes reached 100 mm³, mice were administered either Bosutinib (MCE, HY-10158, at a dosage of 100 mg/kg)[107], Dasatinib (MCE, HY-10181, at a dosage of 15 mg/kg)[108], or DMSO as a control. These treatments were delivered via oral gavage using feeding needles, directly into the stomach.

## CyTOF sample preparation and data acquisition

Antibodies for mass cytometry were obtained in two ways: either directly acquired from Fluidigm or created in-house. For the in-house production, commercially available purified antibodies were conjugated with the suitable metal isotope using the MaxPar X8 Polymer kits from Fluidigm, as detailed in Supplementary Table 1. The conjugation process was carried out according to the manufacturer's instructions.

The harvested tumor samples were prepared into a single-cell suspension as described above. Subsequently, $3 \times 10^6$ cells were transferred to the Cell Staining Buffer. Staining was performed sequentially, starting with membrane antibodies (Fluidigm, 400276), followed by intracellular staining (Fluidigm, 400279), and finally nuclear staining (Fluidigm, 400277), all in accordance with the manufacturer's instructions. Upon completion of the staining process, the cells were fixed using 1.6% formaldehyde at room temperature for 10 min. Subsequently, they were incubated with 125 nM of Intercalator-Iridium (Fluidigm) and incubated overnight at 4°.

Samples were washed twice with Cell Staining Buffer (Fluidigm) and Cell Acquisition Solution Plus (Fluidigm) and then kept on the Carousel while awaiting acquisition. Acquisition was performed on a CyTOF XT system, using the pelleted mode for sample loading. Calibration was carried out using EQ 6 element calibration beads (Fluidigm). The samples were run at a rate that did not exceed 400 events per second, continuing until we obtained 250,000 events. Normalization of the data was accomplished using the CyTOF software, which generated FCS files that we utilized in the subsequent data analysis phase.

## CyTOF data analysis

CyTOF data was first analyzed using Cytobank (http://www.cytobank.cn) for quality control[109], involving the removal of beads (140_beads), dead cells (cisplatin), doublets (191Ir), as well as adjustments for residual, width, center, offset, and event length. Then, the normalized data obtained from Cytobank were used for subsequent analysis following the CyTOF workflow[110].

## In silico drug screening

Human cancer cell lines' drug screening data were obtained from three pharmacogenomic datasets, including Cancer Therapeutics Response Portal (CTRPv.2.0, https://portals.broadinstitute.org/ctrp), Genomics of Drug Sensitivity in Cancer (GDSC1&2, https://www.cancerrxgene.org/),

and PRISM (19Q4, https://depmap.org/portal/prism/). The data were pre-processed based on our previously described pipelines[52]. 632 cancer cell lines and 408 drugs from CTRP, 741 cancer cell lines and 282 drugs from GDSC, and 440 cancer cell lines and 1291 drugs from PRISM were enrolled in this study.

The ridge regression model in pRRophetic R package was applied to estimate the drug sensitivity of LUAD patients[55]. This model was firstly trained on the expression data and drug response data of cancer cell lines, then, based on the expression data of cancer cell lines and the expression data of patients, the predicted drug response data of LUAD patients were estimated. After obtaining the predicted drug response data, differential analysis was conducted between the UPP1 high and UPP1 low groups. Meanwhile, we classified cancer cell lines into UPP1 high or UPP1 low cell groups and conducted differential analysis, we then integrated the predicted drug response data of LUAD patients with the cancer cell lines' drug sensitivity data and obtained three agents that were correlated with UPP1 expression.

## Organoids

The LUAD clinical samples were minced into small pieces and were subjected to enzymatic digestion in 5 ml of Advanced DMEM/F-12 (Gibco, 12634010) supplemented with 5 mg/ml collagenase II (Gibco, 17101015), 100 μg/ml DNase I (Roche, 11284932001), and 10uM Y-27632 (Abmole, M1817) for 1 h at 37 °C on a shaking platform (120 rpm). Then, samples were centrifuged at 400 × g for 5 min. After removing the supernatant, the tissues were incubated in 5 ml of TrypLE Express (Gibco, 12605010) + 10 μM Y-27632 for 10 min at 37 °C. This step was followed by adding Advanced DMEM/F-12 supplemented with 10% FBS to neutralize the TrypLE Express. Cells were pipetted up and down to dissociate tissue fragments. Cell suspensions were passed through 70μm cell strainers, mixed with Matrigel (Corning, 356231), and 40,000 cells in 50 μl droplets were dispensed into pre-warmed 24-well plates. Plates were put upside down in the incubator for 10 min, and then the organoid culture medium was added (Jiayuan Bio-technology, WM-H-10)[111].

After 2-3 weeks of culture, organoids were harvested from the culture medium for passaging. The Matrigel domes containing the organoids were carefully scraped using a pre-wetted P1000 pipette tip. These organoids were then transferred to a 15 mL tube and centrifuged at 200 × g for 5 min at 4 °C. The supernatant was discarded, and the organoids within the Matrigel were digested using 2 mL of TrypLE Express supplemented with 10uM Y-27632. This digestion was conducted at 37 °C on a shaking platform set to 60 rpm for approximately 5 min. Then, 4 mL of cold Advanced DMEM/F-12 supplemented with 20% FBS was added. Following another centrifugation step, the supernatant was discarded, and the cell pellet was washed with 10 mL of cold Advanced DMEM/F-12. The organoids were then dissociated into smaller clusters using a pre-wetted 10 mL pipet in cold Advanced DMEM/F-12. Finally, the dissociated cells were re-seeded in fresh organoid culture medium. After three passages, these organoids are suitable for subsequent analysis.

For histological processing of organoids, organoids were harvested by scraping off the Matrigel domes. After centrifugation at 200 × g for 5 min, they were washed in cold Advanced DMEM/F-12 and shaken to dissolve any residual Matrigel. The organoids were then fixed in 10% neutral-buffered formalin for 2 h. After fixation, they were resuspended in PBS. To embed the organoids, a 2% agarose solution was prepared and cooled to about 50 °C. The organoids were gently mixed with this agarose solution and then pipetted onto parafilm, allowing the agarose to form a dome structure as it solidified. This agarose-embedded organoid structure was processed similarly to regular tissue samples for subsequent histological analysis[111,112]. The tumor purity of organoids was validated using TTF-1 (Maixin biotech, MAB-0677, 1:100), CK5 (Proteintech, 66727-1-Ig, 1:100), and p63 IHC kit (Proteintech, KHC0086, working solution).

## Validating the drugs using cell lines and organoids

The established UPP1-OE and UPP1-OENC tumor cells were used to validate whether the difference in UPP1 expression was correlated with different drug sensitivities in vitro. The cells were cultured with the presence of Bosutinib (MCE, HY-10158), Dasatinib (MCE, HY-10181), or Erlotinib (MCE, HY-50896) for 48 h. Then, the cell viability was assessed using the CCK8 assay (Beyotime, C0038).

For organoid validation, the drug sensitivity assay was conducted by seeding 2000 cells/well in 96-well plates in a complete organoid culture medium containing 5% Matrigel[113,114]. The organoid medium containing different concentrations of each drug was added. The cell viability was measured using the CCK8 assay.

## Statistical analysis

All results were repeated at least three independent times. RT-qPCR data are expressed as mean ± SEM, while other results are presented as mean ± SD. Survival analysis was performed using the log-rank test. Data were evaluated using a two-tailed Student's $t$-test or the Wilcoxon rank-sum test, and comparisons across multiple groups were made using one-way or two-way ANOVA. The Pearson correlation coefficient was employed for correlation analyses. A Chi-square test was used to identify differences in category proportions between two group variables. Statistical significance was determined at $p < 0.05$. All statistical analyses were conducted using R or GraphPad Prism Software.

## Reporting summary

Further information on research design is available in the Nature Portfolio Reporting Summary linked to this article.

## Data availability

The LUAD scRNA-seq publicly available data used in this study are available in the GSA database under accession code CRA001963[80] and HRA000154[84], and GEO database under accession code GSE131907[81], GSE123904[82], and GSE148071[83]. The LUAD bulk publicly available data used in this study are available in the GEO database under accession code GSE19188[115], GSE30219[116], GSE31210[117], GSE37745[118], GSE50081[119], and GSE72094[120]. The TCGA publicly available data used in this study are available in the Xena database (Batch effects normalized mRNA data, Pan-Cancer Atlas Hub) (https://xenabrowser.net)[121]. The scRNA-seq data generated in this study have been deposited in the GSA database under accession code HRA003967. The remaining data are available within the Article, Supplementary Information or Source Data file. Source data are provided with this paper.

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

## Acknowledgements

This work was supported by the Chengdu High-level Key Clinical Specialty Construction Project (ZX20201202020 to GPL) and the National Nature Science Foundation of China (82270772 to XFY).

## Author contributions

Conception and design: YL, MLJ, DG, GPL, and CLL; Data analysis: YL, MLJ, LA, YQW, and FKX; Experiments: YL, MLJ, LA, and LL; Manuscript writing: YL, MLJ, LA, YZ, JG, DP, XH, XFY, DG, GPL, and CLL. All authors read and approved the final manuscript.

## Competing interests

The authors declare no competing interests.
