## [Peer Review File · Nature Communications]

UPP1 promotes lung adenocarcinoma progression through the induction of an immunosuppressive microenvironmentReviewers' Comments:

Reviewer #1:

Remarks to the Author:

Firstly, I would like to commend the authors on a substantial body of work in this submission. The results presented are very interesting and could have meaningful implications on future trials and treatment options for patients suffering from LUAD. The results also open avenues for further research to understand the mechanisms of action of UPP1 and explore its clinical implications. Current research already shows the correlation of UPP1 with tumor progression and poor response in various cancers. However, the authors here present possible mechanisms by which UPP1 may effect this behavior. Recognizing that a singular mechanism may not be sufficient, they do well to explore a few different angles.

I would like to bring to attention the following points:

1. As a general point on methodology, I would like to see more technical detail in the methods and for the methods to be separated into sub-headings and not as one big paragraph containing multiple different steps
2. In the results section, please mention at least the name of a method used. For example, "We inferred the cell abundance of the aforementioned cell clusters"- when you make this statement, mention which method was used
3. Regarding sample integration, two of the datasets are metastatic tissues and from various different sites. Looking at the immune fraction of these sites can be tricky since they are influenced by the immune milieu of the site which may be vastly different from the primary. This should clearly be noted as a caveat and discussed with reasoning since you apply your findings broadly to all LUAD
4. I have some concerns about your scRNAseq clustering methods. Your cut-off of 200 genes is pretty low for good quality cells. We personally use 800-1000 genes detected as a cutoff for QC except for some speciality cell types. Having lower quality 'cells' can skew interpretation, especially in phenotype calling. On top of mitochondrial genes, were ribosomal genes considered for QC? Additionally, I believe your data is highly over clustered. 0.25 log fold change is insufficient to mark differences, with 1-1.5 being commonly used as a cutoff expression value. It is also highly problematic to pick just the top gene of one of these clusters as assume it to be a marker gene for that cluster as you do for most of the sub-clusters (Fig.1F is not meaningful).

You should also explain how sub clustering was done. To be accurate, you should extract all cells of a particular cluster and re-run the analysis from the beginning as with the initial dataset in order to perform sub-clustering. It cannot be performed by just adjusting values in initial clustering run. This is not clear from your methods.

5. Another drawback of over-clustering is your subsequent use of ssGSEA to de-convolute bulk transcriptome data which requires gene set input. A truncated gene set due to over-clustering may lead to false positive cell type abundance values as inferred from bulk expression
6. It would be essential and helpful to know what is the cell number (of a cluster) and which patient populations they hail from when discussing subclusters. For example when discussing ATP5E cluster abundance in metastatic sites, in how many samples is this subcluster seen in? Just one patient or many patients? If it is just from one or two patients, then the generalized claim of ATP5E being higher in metastatic epithelial cells is much weaker

7. Specific line items:

Line 199- how is UPP1 related to disease progression?
Line 203- there is not sufficient evidence to claim UPP1 is predictive of metastatic potential but only correlative.
Line 214- mention briefly what this means and what method was used to determine
Line 216-217- was the PD-1 expression normalized to TIL content of the tumors?
Line 222- What is survival 'probability' in Fig 3C?
Line 226- How were the responders separated by grade?
Line 238- How was this quantified? Immune exclusion cannot be proven conclusively by IHC
Line 239- How many samples was the IF performed on? What is the quantification of co-localization and crosstalk?
Line 394- what are the prominent features of CAFs?
Line 409- markers such as?
Line 431- aSMA is not a CAF marker?
Line 602- please explain these manual adjustments
Line 603- considering you are over-clustering, how are you defining canonical marker genes for subclusters which are largely unknown and tissue/sample/disease dependent?
Line 815- It would be good to see the drug response of the UPP1 over-expressing cell line and knock-down cell line compared to the wild type control of that same cell line for a baseline value

8. Regarding your IHC experiments: I noticed that the staining quality and intensity of the UPP1 antibody is not that high. How are you classifying as low/mid/high expressors for your IHC scoring?

A note to both the authors and the Editor:

As mentioned above, I have reservations about the methodology of clustering and the decision to overcluster using very low expression fold change thresholds. This does not lead to biologically discernable cell types with any confidence. Also, using these sub clustered gene sets to infer population abundance from bulk data is also inaccurate. That being said, the genes that the authors pick out, especially UPP1 does show correlation with patient response and the authors have performed several validation experiments to back up their hypotheses. I believe this works only because a single gene was picked as a candidate from the analysis and not a 'cell type', which would have been inaccurate. This is a case of process vs results, and in this case the result and validation ends up overriding sub-optimal process. I believe that isn't always a bad thing when validation is provided, but it would be remiss of me to not make note of this issue.
I suggest that the authors amend their language to avoid naming these sub-sub-clusters as 'cell types' and restrain to discussing the selected gene as important. This also applies to inferences from bulk transcriptome data.

My final suggestion would be to rethink what you are including in this manuscript. Although there is a lot of work presented here, the key findings are the role of UPP1 and it's potential significance in patient treatment, which should be presented as a cohesive story front and center, followed by a presentation of possible mechanism. The fibroblast and immune remodeling can be a part of the mechanistic hypothesis.

Overall, there are sufficient original and meaningful results presented here that merit publication.

Reviewer #2:

Remarks to the Author:

The authors set out to identify cancer subpopulations associated with poor prognosis, and started by integrating multiple scRNAseq datasets of lung adenocarcinoma (LUAD). Of 10 malignant cell subpopulations that were associated with poor prognosis and metastasis, the UPP1+ population was selected for further investigation. The authors show that UPP1 expression is associated with poor

prognosis, and that knockdown of UPP1 in cell lines impairs their proliferation and invasiveness, suggesting a tumour-intrinsic fitness effect of UPP1 levels.

The authors then investigate the association of UPP1 with anti-tumor immunity. However, the claim that UPP1 expression is associated with an immunosuppressive microenvironment has limited support (see major comments below). While its association with response to checkpoint inhibition (Fig. 3F) is interesting, this is seen in only one dataset. The characterization of the tumor microenvironment is based on difficult to interpret scores (e.g. Fig. 3B) and it remains rather speculative how the TME from UPP1^{high} and -low patients now really differs. It appears that modulating UPP1 levels can regulate PD-L1 expression, but the mechanism remains unclear and the association with PI3K/Akt signaling is correlative. All in all, it remains unclear whether UPP1 truly plays an important immunomodulatory role and is causal to this. Since this is the core message of the paper, this claim needs more solid support.

In the second half of the paper the authors explore the relationship of UPP1⁺ tumor cells with CXCL13⁺ T cells and fibroblasts. There are issues with the analyses (see major comments below) that limit the value of these data.

The authors should be complimented on the volume of the work, and their efforts to combine bioinformatic analyses with experimental validation. Yet there are several major issues that need to be addressed to convincingly present this story.

Major comments

- The figure legends provide very limited information on the analyses performed. It is frequently unclear how a metric has been defined or how the analysis was done. Graphs are poorly defined (e.g. number of replicates, statistical tests used). More detailed information is needed to allow interpretation of the data.

- In Fig. 2C associations between PMRT and various gene sets are reported but while significant the effect size is marginal. It is unclear why these gene sets were selected. Were these part of a larger collection and are only the significant results reported? Is this corrected for multiple hypothesis testing?

- The claim that UPP1 is associated with an immunosuppressive tumor microenvironment (TME) is insufficiently substantiated. Metrics for immunosuppression (e.g. in Fig. 3B) are poorly defined. The authors have a wealth of single cell sequencing data available yet choose to analyse the TME on bulk RNA sequencing datasets. Moreover, the data in Fig. 3E (T cell enriched subset) is not very different compared to the data in Fig 2D-E (overall cohort) and data showing that UPP1 has limited prognostic value in T cell depleted patients is lacking. Overall, this claim needs stronger support.

- The heading of the third section ("UPP1 contributed to the immune suppression of CD8⁺ T cells by regulating PD-L1 expression") is insufficiently substantiated. The point that UPP1 regulates PD-L1 expression is convincing but whether this is responsible for immunosuppression of CD8⁺ T cells is not proven. In fact in Fig. 4B the added effect of PD-L1 inhibition is minimal and most of the therapeutic effect comes from inhibition of UPP1 alone. This is in line with the data in Fig. 2G-H which suggests that there is a tumor-intrinsic effect of UPP1 inhibition which is unrelated to immune suppression. Would the effect of UPP1 knockdown on tumour growth be similar in immunodeficient mice or in mice in which T cells are depleted?

- The mechanistic link between UPP1 and PD-L1 is insufficiently clear.

- The relationship between UPP1^{high} tumor cells and CXCL13^{low} T cells is poorly supported. The data is mostly correlative (e.g. Fig. S8E), based on small and difficult to interpret differences between CXCL13^{high} and -low cells (Fig. 5G-H), without a comparison to UPP1^{low} cells (Fig. 5G-H) and based on inappropriate gating (Fig. 5K-L).

- Throughout the paper, the flow cytometry results are of low quality. Gates regularly cut through populations and appropriate controls to show the rationale of the placing of gates are missing. This significantly limits the interpretation of these data.

- There are major gaps in the fibroblast story. The authors identify a fibroblast subpopulation associated with poor prognosis, but its relationship to UPP1^{high} (as set apart from UPP1^{low}) tumor cells is unclear. It is unclear whether the populations enriched after co-culture with UPP1-oe tumor cells bear resemblance to the poorly prognostic population identified above. The relationship to

immunosuppression is based on correlative data (Fig. 6L) or data of poor quality (Fig. 6M).

Minor comments

- Fig. 1C: what do 'mutated' and 'wild type' refer to?
- Fig. 1E: the NK cell cluster also expresses CD3 markers that are shared with the T cell cluster. Has this cluster been correctly assigned?
- Fig. S1C/D: Can the authors explain why virtually no T cells and NK cells are represented from the Wu et al. cohort?
- Table S2 only contains marker genes for the CD8+ T cell cluster
- Line 186: "expression of UPP1 is higher in metastatic brain and lymph node" In Fig. S3A it does not look like the expression in brain is higher than in the primary tumor. No significance testing has been performed.
- Fig S3B: on what basis were these genes chosen? It seems a random subset of immune escape genes. Has this analysis been performed in an unbiased fashion, or were only significant immune escape genes reported?
- Fig. 2C: are the genes defining these pathways defined in the methods? How was this analysis carried out? AUC seems an unusual metric for this. Even though the results are statistically significant, the effect size is minimal. Can the median score be reported to allow a better interpretation of the effect size?
Also: how were these pathways selected? Were these part of a larger collection of gene sets, and has any multiple hypothesis correction been performed?
- Figure S3: the level of siRNA mediated UPP1 knockdown is minimal (Fig S3G), yet the effects on invasion and colony formation are quite strong. Are the authors confident that this is not due to a possible off target effect? Showing this with two independent siRNAs would significantly strengthen this observation
- Fig. 3B: why was this analysis performed on the bulk RNAseq datasets? The correlation is significant but weak. Is a similar trend seen in the single cell RNAseq datasets?
Also: what is the definition of the 'T cell inhibitory score'?
- Fig. 3C: is the association with OS only seen for hot tumors? What about cold tumors?
- Fig. 3D: the figure is presented in a confusing way. Why not make two separate panels for CTLtop and CTLbottom patients, and then show the effect of UPP1 levels instead? That would make it more clear that the effect is most pronounced in the CTLhigh patients.
- Line 230: "UPP1 was associated with the immune suppressive microenvironment": this claim is overstated based on the presented data. There is no real characterization of the microenvironment aside from PD1 levels on T cells. Also see the point above.
- Fig. S5F: this is anecdotal evidence – can the authors, with the support of a pathologist blinded to UPP1 status, score immune exclusion phenotypes and then perform this on their TMA cohort more broadly?
- Without a quantification of the proximity of UPP1+PD-L1+ tumor cells and PDCD1+CD8+ T cells the data in Fig. 3I is anecdotal and does not add to the story.
- Fig. S5G-H: the shift in MFI (particularly for the overexpression experiment) is limited – quantification on replicate experiments and significance testing is required. The absence of strong effect may be due to the low level of overexpression and knockdown as evident from the Western blots.
- Fig. S5K: the decrease in p-Akt signalling as a consequence of UPP1 knockdown is convincing but this does not prove that PD-L1 is regulated through an UPP1-PI3K-Akt axis as it is correlative data.
- Fig. 4F: the gating strategy for these plots is unclear. For most populations the gates do not clearly identify separated populations but rather cut through the main population. Was there a fluorescence minus one (FMO) control to set the gates?
- In Fig. S6C-D claims are made about the cytotoxicity of T cells but this is an indirect measure. Did the authors perform killing assays to directly measure whether these T cells are more potent in killing target cells? The same gating issues apply as in the comment above.
- Fig. 4J-K: quantification of the NucView signal is complicated by the fact that there is no clearly defined positive and negative population – the gates cut through the main body of the population. An

optimized or alternative method should be used to quantify apoptosis, ideally combined with a quantification of the number of live tumour cells left after co-culture.

- Fig. S7A-C: number of replicates not reported. UPP1 knockdown already has an effect on tumour cell survival in the absence of T cells – these data should be shown as well in the bar graphs.
- Fig. S7F: as for the other flow plots, the gating is challenging to interpret without a proper negative control.
- Fig. 5F: how is the T cell activation score defined?
- Fig. S8A: the Kaplan Meier curve is not clearly defined. When is a patient classified as CXCL13^{high} and when as CXCL13^{low}?
- Fig. S8D: the UPP1^{high} group should be included as well as reference. Moreover, a KM curve will aid the interpretation of this data.
- Sig. S9A/D: these data are difficult to interpret without a more detailed figure legend.
- Fig. S9H: the heat map is incompletely labeled
- Line 379 refers to Fig. S3J which should be Fig. S9J. The authors highlight IL-10 but other ligand/receptor interactions such as TGFB / TGFBR signalling are equally strong.
- The data on myeloid cells (Fig. S9) feels somewhat misplaced in the context of this story. The focus of this manuscript is on UPP1+ tumor cells. The authors describe myeloid cluster 8 but finish the section by stating that myeloid cluster 6 is actually more relevant in the interaction with UPP1+ tumor cells (although this is based on rather modest differences in interaction number and strength between groups).
- Fig. 6C: can the authors explain why the pseudotime trajectory seems to indicate an early \rightarrow late direction that is opposite to the arrow and opposite to the differentiation trajectory proposed by the authors?
- Fig. S10: in this and similar analyses, the authors only evaluate the interaction with UPP1^{high} but not UPP1^{low} tumor cells. These data therefore do not allow differentiating between an interaction with fibroblasts and tumor cells in general, or with UPP1+ tumor cells specifically.
- Fig. 6K: confidence intervals should be shown, as well as data on the other clusters (those not enriched in the UPP1^{high} group) to show that this is specific to the clusters of interest.
- Fig. S11C: it is unclear whether these fibroblasts were co-cultured with UPP1-oe or control cells, and in any case data on both experiments should be reported.
- Line 409: “using the CAFs-marker genes from the aforementioned CAFs” \diamond does this refer to the markers of the cluster 0 that was reported on in Fig. 6A-B?
- Fig. 7G: while statistically significant, the treatment effect size is limited and unlikely to having clinical significance.

Reviewer #3:

Remarks to the Author:

In their manuscript ‘Characterization of the immunosuppressive roles of UPP1^{high} tumor cells in the tumor microenvironment of lung adenocarcinoma and its prognostic associations’, Li et al integrate multiple datasets, including multiple lung single cell data sets, bulk RNA-seq data sets, ICI-treated datasets, and experimental data, to identify UPP1 as a possible regulator of the tumor immune interaction and mediator of outcomes. The authors assemble an impressive amount of analyses to implicate UPP1 in lung cancer biology; however, some of the conclusions seem overstated or not fully supported by the data, possible confounding features are not adequately addressed, and the presentation of the data and the associated analyses could be more clear.

1. In general, the cell subtyping within each compartment seems overclustered. Utilization of other metrics to assess for cluster stability and heterogeneity would help. The high degree of subclustering also doesn’t help the analysis/data presentation, because it is hard follow and to know which subclusters are biologically significant.

2. More detail regarding the clustering-outcome association is needed in the methods. The authors write that ssGSEA was used to quantify the population abundance of cell types; what gene lists were

used as the input genes? What was the outcome metric used in the bulk cohorts and are these uniformly calculated in these pooled datasets? Are the outcomes across cohorts relatively stable, or do they differ based on enrichment with EGFR mutant lung cancer, early stage, etc? How was the outcome analysis actually performed – Is the ssGSEA score used as a continuous variable?

3. How is purity accounted for in the bulk cluster analyses?

4. The malignant cell subclustering analysis needs much more explication. First, the malignant cell UMAP should be included in Figure 2 for ease of visualization along with top expressed genes (dot plot or heatmap) to help justify the assignment of identities such as AT2-like etc. Second, the mixing of malignant cell subclusters across samples needs to be confirmed. Are these subclusters driven by specific patients or do they represent broader biology? Third, as above, how stable/distinct are these subclusters? Fourth, are any of the outcome associations actually statistically significant? Only hazard ratios are reported.

5. The stage and disease site associations with malignant subclusters may be confounded by sample or disease specificity (i.e. if a subcluster predominantly occurs in an early-stage sample, it will appear associated with early stage disease). It is also important to indicate whether these clusters associate with specific genotypes or other clinical features that may drive prognosis.

6. Related to the previous two points, the association between UPP1 and outcome needs to be analysed for possible confounding by clinical and genomic features. For example, the fact that UPP1 associates with erlotinib sensitivity suggests it may simply associate with EGFR mutations, which would confound all of the subsequent analyses.

7. The text claims, based on figure S3D, that UPP1 associates with disease progression. These gene sets alone cannot support this claim, and furthermore the association is fairly weak. Would need to see the full range of gene sets that associate, and stronger clinical association. In general, the rationale for selecting UPP1 for further study is not totally clear.

8. Additionally, the claim that UPP1 associates with T cell inhibition seems poorly supported by the bulk RNA-seq data (3B) – these are very low correlation coefficients even if they are statistically significant.

9. The move to look at UPP1 expression within T cell enriched patients is also not supported – why would the expectation be that UPP1 outcome association would be context specific? How is T cell enriched defined? How is UPP1 high vs low defined? How does this differ from the overall outcome association shown in Fig 2? Did UPP1 associate with outcome in T cell low patients?

10. The ICI outcome cohorts are small, derived from distinct cancer types, not consistently presented, and do not represent the best published cohorts. Regarding presentation, why is the data from the TIDE cohort differently stratified than Fig 3C (CTL high vs low in UPP1 high vs low rather than looking at UPP1 high vs low in CTL high)? Similarly, why is PFS/OS for the Zhongshan IO cohort not presented, and why isn't response stratified by UPP1 low vs high to be consistent? If non-NSCLC cohorts are going to be used, would use other larger published cohorts, which are enumerated in several meta analyses including Litchfield et al, Cell 2020. The selected datasets are not the best IO validation datasets due to cancer type and small size, and even so not all of the included datasets are used.

11. The claim that UPP1 is associated with immunosuppression and IO resistance is not conclusively shown by the data in 3A-F. The previous figure demonstrated that UPP1 was potentially prognostic; there is little to suggest that the data is predictive to ICI over and above its prognostic role (related to the previous reviewer point re IO outcome cohorts). If the contention is that UPP1 associates with a more immunosuppressive immune microenvironment, why wasn't the single cell data used to explore this claim?

12. PD-L1 expression can be a positive predictive marker for ICI response in NSCLC. If UPP1 associates with PD-L1 expression, it could have positive or negative implications for ICI response. The mechanism behind UPP1 association with immune response is not explicated.

13. In Fig 3H-I and S5F, could UPP1 just be a tumor cell marker? Do UPP1 negative cells have a different T cell/PD-L1 configuration? Can this be demonstrated across more than a handful of samples?

14. The UPP1 knockdown and overexpression does not appear very effective in multiple cell lines.

15. Much of the T cell specific analyses identifies features that have previously been described, and are not convincingly associated with UPP1 status – most of this rests on Figs 5G-H which is very

difficult to interpret. The co-culture data are interesting but there is still a marginal effect on the proportion of CXCL13^{high} CD8 T cell expression.

16. The analysis of fibroblasts is interesting but adds a lot of data to an already dense manuscript that would benefit from more validation of the initial, central claims.

Minor comments:

- The paper is well written but would benefit from further English language editing for ease of reading (e.g. wildly used instead widely)
- There are multiple cohorts used at various points. It would be helpful to define upfront in the results the different cohorts, and to have them used consistently throughout, to avoid the appearance of data cherry-picking.
- Fig 3A derives from which cohort? Why only one?
- What does 'mutation' status refer to in the early figures? Better to be specific?
- Related to the comments above, would focus first on exploring UPP1 and the TME in the single cell datasets, then moving to the therapeutic implications. The initial rich, datasource is not fully taken advantage of.

Response to Reviewers' comments:

First and foremost, we would like to express our sincere gratitude to the reviewers for their thorough evaluation of our manuscript and for providing valuable feedback. We appreciate the time and effort invested in reviewing our work, and we have taken each comment into careful consideration.

In response to the comments and suggestions, we have made comprehensive revisions to our manuscript to address the concerns raised. We hope that these modifications have significantly improved the quality, clarity, and robustness of our research.

Below, we provide a point-by-point response to each of the reviewers' comments, detailing the changes we have made and clarifying any misunderstandings. In responding to the reviewer's comments, the black figure legends refer to figures presented in the response document, while the red figure legends denote figures and content in the manuscript.

We hope that the revised manuscript would meet the expectations of the reviewers. If there are any further issues, we sincerely hope you can point them out and offer us the opportunity to revise further.

Reviewers' comments:

Reviewer #1, expertise in scRNA-seq/TME/lung cancer (Remarks to the Author):

Firstly, I would like to commend the authors on a substantial body of work in this submission. The results presented are very interesting and could have meaningful implications on future trials and treatment options for patients suffering from LUAD. The results also open avenues for further research to understand the mechanisms of action of UPP1 and explore its clinical implications. Current research already shows the correlation of UPP1 with tumor progression and poor response in various cancers. However, the authors here present possible mechanisms by which UPP1 may effect this behavior. Recognizing that a singular mechanism may not be sufficient, they do well to explore a few different angles.

I would like to bring to attention the following points:

1. As a general point on methodology, I would like to see more technical detail in the methods and for the methods to be separated into sub-headings and not as one big paragraph containing multiple different steps

Response:

Thank you very much for this comment concerning the methods section of our manuscript. We agree that more technical detail would enhance understanding and that subdividing the section into subheadings would improve readability.

To address this, we have taken steps to improve the technical detail within the methods section. We have restructured this section, introducing subheadings to better organize and clarify the various steps of our methodology. Alongside each subheading, we have provided more comprehensive explanations of the specific techniques, materials, and software packages used in our study.

We hope that this new structure and additional information would enhance the understanding of our methods and increase the reproducibility of our results. We also hope that these revisions would meet your expectations and address your concerns.

2. In the results section, please mention at least the name of a method used. For example, "We inferred the cell abundance of the aforementioned cell clusters"- when you make this statement, mention which method was used

Response:

Thank you very much for this comment. We apologize for any lack of specificity in our original manuscript. We understand the importance of providing clear, concise details about the methods used in our research. We have updated the results section to include the names of the specific methods used, including the method used to infer cell abundance of the mentioned cell clusters.

We hope that these adjustments would improve the clarity of our results section, and we appreciate your guidance in helping us enhance the quality of our manuscript.

3. Regarding sample integration, two of the datasets are metastatic tissues and from various different sites. Looking at the immune fraction of these sites can be tricky since they are influenced by the immune milieu of the site which may be vastly different from the primary. This should clearly be noted as a caveat and discussed with reasoning since you apply your findings broadly to all LUAD

Response:

Thank you very much for this insightful comment and pointing out this crucial aspect. In these two datasets, a subset of samples is from metastatic tissues. We understand your concerns regarding the potential variability in the immune milieu of metastatic tissue samples, and its possible divergence from the primary tumor. In our initial methodology, we overlooked this and did include these metastatic samples in the analysis of cell-cell communications and characteristics of the microenvironment, which may introduce some bias into the results.

To address this concern, we have revised our analysis to exclude the metastatic samples when studying the microenvironment characteristics. Specifically, after the identification of cell populations, we only considered cell populations present in more than 10 tumor samples(1) and excluded these metastatic site samples from our subsequent analysis of cell-cell interactions and the associations between UPP1^{high} tumor cells and other immune microenvironment cells (Line 593-594; Line 651). As to the initial step of data integration and cell identification, we maintained the original methodology, that is, we integrated samples from all datasets and then conducted cell clustering and cell population identification within the overall sample. This approach is common in current single-cell studies, where data from metastatic site samples are typically integrated with other primary samples for initial cell population identification(1-3). We really appreciate your guidance in helping us refine our approach.

4. I have some concerns about your scRNAseq clustering methods. Your cut-off of 200 genes is pretty low for good quality cells. We personally use 800-1000 genes detected as a cutoff for QC except for some speciality cell types. Having lower quality 'cells' can skew interpretation, especially in phenotype calling. On top of mitochondrial genes, were ribosomal genes considered for QC?

Additionally, I believe your data is highly over clustered. 0.25 log fold change is insufficient to mark differences, with 1-1.5 being commonly used as a cutoff expression value. It is also highly

problematic to pick just the top gene of one of these clusters and assume it to be a marker gene for that cluster as you do for most of the sub-clusters (Fig. 1F is not meaningful).

Response:

Thank you very much for bringing up these important concerns. During our single-cell analysis, we referred to official guidelines and previously published papers. In many of these studies, the gene cutoff was set at 200(4-8). For example:

1. He D et al. Single-cell RNA sequencing reveals heterogeneous tumor and immune cell populations in early-stage lung adenocarcinomas harboring EGFR mutations. *Oncogene*. 2021;40(2):355-68. (**gene cutoff: 200**).
2. Laughney et al. Regenerative lineages and immune-mediated pruning in lung cancer metastasis. *Nat Med*. 2020;26(2):259-69. (**gene cutoff: 200**).
3. Wu F et al. Single-cell profiling of tumor heterogeneity and the microenvironment in advanced non-small cell lung cancer. *Nat Commun*. 2021;12(1):2540. (**gene cutoff: 200**).
4. Xing X et al. Decoding the multicellular ecosystem of lung adenocarcinoma manifested as pulmonary subsolid nodules by single-cell RNA sequencing. *Sci Adv*. 2021;7(5). (**gene cutoff: 200**).
5. Luo H et al. Pan-cancer single-cell analysis reveals the heterogeneity and plasticity of cancer-associated fibroblasts in the tumor microenvironment. *Nature Communications*. 2022;13(1):6619. (**gene cutoff: 200**).

Therefore, we adopted the same method in our analysis. We understand that using higher values, such as 800-1000, can yield higher quality cell populations. Our initial aim was to include as many cells with potential biological significance as possible, so we adopted a relatively relaxed threshold in this process. Aside from the gene cutoff parameter, other data processing steps in our study were conducted according to the official recommendations and referenced previously published literature. Thus, we believed the quality control process we implemented could yield high confidence cells. Upon inspecting the final results, we observed that each cell population we identified basically existed in every dataset. This indicated that the cells included in our final analysis were generally supported by multiple datasets, affirming the ubiquity and credibility of these cells (Figure S1-S3). On the other hand, in the revised manuscript, we have added more experiments to validate our claims. We believe that these additional steps should enhance the reliability of our conclusions.

For the consideration of ribosomal genes, we made adjustments based on the latest published literature(1). In this paper, the authors excluded a total of 1,514 genes associated with mitochondria (50 genes), heat-shock protein (178 genes), ribosome (1,253 genes), and dissociation (33 genes) to avoid unexpected noise and expression artifacts caused by dissociation. Therefore, we adopted the same method and excluded these genes for subsequent analysis (Line 543-545).

Regarding the clustering of cells, we have reconsidered our analysis process and results based on your valuable suggestions. Our initial aim was to identify as many cell groups as possible in order to find potential prognostic genes. Therefore, we set a relatively high "resolution" parameter during the clustering process in the original manuscript. But now, we realized that there might indeed be some instances of over-clustering. To address this, we revisited the data and reanalyzed it. In order to avoid over-clustering, we tested multiple resolution parameters for cell clustering. For each resolution parameter, we compared the number of cells and UMAP distribution of cells in each cluster, as well as the differentially expressed genes of each cell cluster. We then selected a relatively stable resolution parameter, balancing the need to minimize over-clustering while preserving the biological significance of identified cells, for further analysis.

We used tumor cells as an example, we tried resolution parameters from 0.4 to 1.5. We found that at 0.5, we could capture the UPP1^{high} tumor cell group. As the parameter increased, the number of UPP1^{high} tumor cells (cluster 16) remained stable, but other clusters, such as cluster 2 began to be gradually divided into multiple cell subgroups when the parameter was 0.7, with unclear boundaries between the subgroups, suggesting potential over-clustering (see below, Figure R1). Therefore, through this method, we finally chose 0.6 as the resolution parameter for tumor cell grouping in the revised manuscript (Figure 1D). Also, after consulting relevant literature sources, we realized that in the Seurat analysis, the resolution parameter isn't universally fixed. For instance, opting for lower resolution values, such as 0.1 or 0.4, tends to yield fewer clusters, potentially amalgamating distinct cell subpopulations. Conversely, higher resolutions like 1.5 or 2.0 might lead to a greater number of clusters, risking the over-segmentation of genuine cell subpopulations. In practical applications, a resolution range of 0.4 to 1.2 is often deemed reasonable, striking a balance between granularity and accuracy.

Furthermore, we investigated whether there were significant differences in the differentially expressed genes of UPP1^{high} tumor cells under different parameters. This analysis was conducted to verify whether our previous results, obtained at a higher resolution, may have been biased. However, the results consistently showed that high UPP1 expression was a defining characteristic of this group of cells across different parameters (Figure R2). These findings suggest that although there may be differences in other cell populations in our previous version of the paper, the characteristics of the UPP1^{high} tumor cell group remained stable under different parameters. This supports the validity of our decision to focus on the UPP1 for further research.

Figure R1. Clustering of tumor cells under different parameters.

Figure R2. Top differential genes in UPP1 tumor cell group under different parameters.

For other cell populations, we also used this method for re-clustering and identifying cell groups. we followed the recent literature and grouped T cells and NK cells together for clustering, and we used a parameter of 0.9(1). For myeloid cells, we selected a parameter of 0.5. Similarly, for B cells, we chose a parameter of 0.4. For stromal cells, we opted for a parameter of 0.8 (Figure R3-R6, Figure S2).

Figure R3. Clustering of T and NK cells under different parameters.

Figure R4. Clustering of myeloid cells under different parameters.

Figure R5. Clustering of B cells under different parameters.

Figure R6. Clustering of stromal cells under different parameters.

We have also provided descriptions of these methods in the revised manuscript (Line 570-580). In addition, we have also included featureplots and dotplots in the revised manuscript, aiming to illustrate the feature genes of these cell clusters (Figure S2). Moreover, after considering the comments from you and other reviewers, we have shifted the focus of our paper to the role of UPP1 and its related mechanisms. In the revised manuscript, we first conducted an analysis on the prognostic associations of tumor cell populations. Following this, we focused our attention on the UPP1 cell cluster. To substantiate its relation to an immunosuppressive microenvironment, we carried out a series of experiments. And during the analysis, we no longer used a 0.25 log fold change as the cutoff. Instead, we define marker genes for this group of cells as those with a LogFC > 1 and an adjusted p-value less than 0.05. These genes were then used for subsequent ssGSEA analysis.

As for picking the top genes for naming the cells. This method ("cluster_celltype_marker" format) for defining cell populations was reported in previously published literatures(1, 9, 10). For example, Xue et al. used CD4T_01_CCR7, CD4T_02_SELL, and CD4T_03_GPR183 format (Figure R7) to name cells(1).

Figure R7. Cell annotation in the paper published by Xue et al.

In our paper, we followed the same principle in naming cell populations. For cell populations that distinctly expressed known markers, we utilized these markers to represent these cells. For example, we named CD4T cell population that highly expressed FOXP3 as 8_CD4_FOXP3 (8 is the cluster number) (Line 586-592; Figure S2B). On the other hand, for other cell populations where the highly differentially expressed genes did not correspond to traditional known cell-defining markers, we elected to name them based on the top differential gene. Our purpose was not like the traditional single-cell analysis that depicted the characterization of the tumor microenvironment, such as what proportion Tregs cells or exhausted T cells constitute. Therefore, from the perspective of our research goals, we thought this naming method was reasonable. After naming the cells, we identified the cell populations we were interested in, and then proceeded to a detailed analysis of their characteristics. Thus, we thought this was a matter of analysis order and did not significantly impact the results.

Many single-cell research articles combined single-cell data with bulk data for prognosis analysis. The definition of marker genes in these articles varies, and there currently isn't a gold standard. Therefore, after considering your valuable suggestion, in the revised paper, we defined marker genes for this group of cells as those with a LogFC > 1 and an adjusted p-value less than 0.05. These genes were then used for subsequent ssGSEA analysis.

You should also explain how sub clustering was done. To be accurate, you should extract all cells of a particular cluster and re-run the analysis from the beginning as with the initial dataset in order to perform sub-clustering. It cannot be performed by just adjusting values in initial clustering run. This is not clear from your methods.

Response:

Thank you very much. We apologize for any confusion in our methods. The approach we adopted to perform sub-clustering is exactly as you described. First, we extract all the cells belonging to a specific cluster based on their barcodes. Then, we treat this extracted subpopulation as a separate dataset and re-run the entire analysis pipeline, which includes normalization, identification of variable genes, scaling, and running Harmony to remove batch effects. This process mirrors the

initial analysis performed on the entire dataset. After completing these steps, we define the clusters for the subpopulation (Line 570-573). The corresponding code for this process is as follows:

```
#extracting cell subpopulations
#sce2 is the integrated data
ce_tumor <- sce2[,sce2@meta.data$Index %in% tumor$Index]
ce_tumor <- NormalizeData(ce_tumor, normalization.method = "LogNormalize")
ce_tumor <- FindVariableFeatures(ce_tumor, selection.method = 'vst', nfeatures = 2000)
ce_tumor <- ScaleData(ce_tumor)
ce_tumor <- RunPCA(ce_tumor)
#run harmony on the new dataset
ce_tumor <- ce_tumor %>%
  RunHarmony("orig.ident",
             plot_convergence = TRUE,
             kmeans_init_nstart=20,
             kmeans_init_iter_max=10000,
             max.iter.harmony = 200,
             max.iter.cluster = 200)
ce_tumor <- FindNeighbors(ce_tumor, dims = 1:30,reduction = "harmony")
ce_tumor <- FindClusters(ce_tumor, resolution = c(0.4-1.5))#0.4-1.5
ce_tumor <- RunUMAP(ce_tumor, dims = 1:30,reduction = "harmony")
```

5. Another drawback of over-clustering is your subsequent use of ssGSEA to de-convolute bulk transcriptome data which requires gene set input. A truncated gene set due to over-clustering may lead to false positive cell type abundance values as inferred from bulk expression

Response:

We sincerely appreciate your insightful comment. Over-clustering does indeed pose a risk of truncating the gene set. In the revised paper, as mentioned in our response to the previous question, we have re-clustered the cells and adjusted the parameters to avoid over-clustering. Consequently, the gene set we now obtain is more reliable. Based on this, we have also adjusted the threshold, i.e., $\text{LogFC} > 1$ with an adjusted p-value < 0.05 . We hope these revisions could adequately address your concerns and enhance the robustness of our study.

6. It would be essential and helpful to know what is the cell number (of a cluster) and which patient populations they hail from when discussing subclusters. For example, when discussing ATP5E cluster abundance in metastatic sites, in how many samples is this subcluster seen in? Just one patient or many patients? If it is just from one or two patients, then the generalized claim of ATP5E being higher in metastatic epithelial cells is much weaker

Response:

Thank you very much for your comment. In response to your valuable comments, we have added Figure S3 to the revised manuscript to display the requested information. Firstly, we used UMAP plots to display cells colored by Cohort, Patient sample, Stage, and Type to assess the data integration results (Figure S3A-E). The UMAP results showed that cells from different cohorts were evenly distributed, indicating successful integration of our data from multiple cohorts.

In addition, in Figure S3F-J, we demonstrated the number of cells in each identified cell group, as well as the number of samples each cell group appears in (also in Table S2-Sheet6). These figures indicate that these cell populations are not exclusive to a single sample or restricted to metastatic samples, but rather they are represented across a diverse set of samples. Moreover, the ATP5E cell group that we previously identified can still be detected in our revised analysis, now labeled as 'Tumor_4_ATP5E' (Figure 1F). As shown in Figure S3, this cell group is derived from more than 50 different samples.

The cell clusters we identified are not significantly impacted by sample variations. The analysis of single-cell data in this manuscript serves as a tool to identify important prognosis-related genes. To further validate our claims, in the revised paper, we further added a sequence of validation experiments. We appreciate your valuable suggestions and hope that these address your concerns.

Specific line items:

7. Line 199- how is UPP1 related to disease progression?

Response:

Thank you very much for this comment. After carefully reading the comments from you and other reviewers, we have identified issues in our original manuscript and have removed the corresponding sentence and figures in the revised version. Previously, we attempted to express that our bioinformatics analysis suggested a relationship between UPP1 and biological processes associated with tumor progression, such as EMT, proliferation, and metastasis. However, we realized that the persuasive power of this statement might not be sufficient just based on what we thought was possible. In the revised article, we have modified this section and performed a new enrichment analysis on UPP1^{high} tumor cells (Figure S5C). Moreover, we have shifted our focus to mainly explore the role of UPP1 and related mechanisms and conducted additional experiments to validate our results.

8. Line 203- there is not sufficient evidence to claim UPP1 is predictive of metastatic potential but only correlative.

Response:

Thank you very much for this comment. Indeed, in the earlier version of our manuscript, the relationship between UPP1 and metastatic potential was only correlative. We acknowledge that there was an issue with the way this was described in the previous version of our manuscript. In the revised manuscript, we have removed this claim.

9. Line 214- mention briefly what this means and what method was used to determine

Response:

Thank you for your comment. We apologize for any confusion that may have arisen from the lack of detail in our original manuscript. We have updated the results section to include the names of the specific methods used.

10. Line 216-217- was the PD-1 expression normalized to TIL content of the tumors?

Response:

Thank you for this comment. In the original manuscript, we aimed to indicate a correlation between UPP1 and immune suppression, particularly relating to the exhaustion state of T cells. Therefore,

we conducted flow cytometry analysis on collected clinical samples focusing on T cells. However, comments from other reviewers highlighted that our depiction of the tumor microenvironment's features was insufficient. They pointed out that merely relying on PD-1 expression in T cells does not fully support our claim. Consequently, in the revised manuscript, we have removed this part. On the other hand, to further substantiate the relationship between UPP1 and the immunosuppressive microenvironment, we utilized multiplex immunofluorescence in lung adenocarcinoma patients, along with AI-based automatic image analysis (Visopharm) (Figure 2 and Figure S9). This approach helped demonstrate the relationship and spatial distribution of UPP1 within the immunosuppressive microenvironment. Additionally, we carried out both in vitro and in vivo experiments, along with CyTOF analysis (Figure 5), to provide evidence of T-cell exhaustion state, incorporating more exhaustion markers. We hope these modifications and additional experiments in the revised manuscript address your concerns and strengthen the conclusions of our study.

11. Line 222- What is survival 'probablility' in Fig 3C?

Response:

Thank you very much for this question, the term survival probability refers to the likelihood of a patient surviving or not experiencing a specific event (such as death, disease recurrence, etc.) at a given time point or over a certain period. The term survival probability came from the 'survival' R package we used for creating the survival curve. The generated image default label is "survival probability," which essentially refers to the survival rate. In the revised manuscript, we have replaced the term "survival probability" following the convention used in previously published literatures. We have changed it to Overall Survival (%) or Recurrence-Free Survival (%) respectively in the revised manuscript (Figure 1I, Figure S4, Figure S5F, Figure S7G, and Figure S8B), which may be more familiar to readers in our field. We appreciate your attention to detail and hope this revision makes the figure more understandable.

12. Line 226- How were the responders separated by grade?

Response:

Thank you for this comment. In the original manuscript, we included a group of 47 LUAD patients to assess the correlation between UPP1 and immunotherapy outcomes. For patients undergoing immunotherapy, we evaluated the response status according to the RECIST v1.1 criteria(11). Furthermore, we performed immunohistochemical staining on the treatment-naïve samples of these patients. Then, based on the immunohistochemical scoring, we categorized patients into high or low expression groups. The score for the proportion of positive cells was quantified as follows: (0, negative; 1, 1-10%; 2, 11-50%; 3, >50%). The staining intensity was calculated as follows: 0 (negative), 1 (weak), 2 (intermediate), or 3 (strong). The total score was calculated as the sum of the above two factors. Patients were classified into negative (0), weak (1-2), moderate (3-4), and strong (5-6) staining groups, among them, moderate and strong groups were defined as the high group, while the negative and weak groups were defined as the low group(12).

However, Reviewer 3 pointed out that our sample size of 47 immunotherapy specimens might be insufficient to substantiate a comprehensive correlation between UPP1 and immunotherapy. Considering this feedback and the other modifications we have made to the content of the manuscript, we decided not to include this part in the article.

13. Line 238- How was this quantified? Immune exclusion cannot be proven conclusively by IHC

Response:

Thank you very much for this comment. We agree with your point that the phenomenon of immune exclusion cannot be conclusively demonstrated solely through immunohistochemistry (IHC). We recognized that our previous manuscript may have overstated these results. Therefore, we have decided to remove the description of immune exclusion from our paper.

14. Line 239- How many samples was the IF performed on? What is the quantification of co-localization and crosstalk?

Response:

After considering your comments and those from other reviewers, we recognized that our previous immunofluorescence (IF) results may not have adequately represented the relationship between UPP1 and the tumor microenvironment. Therefore, in our revised manuscript, we re-examined this relationship. We performed multiplex immunofluorescence on the samples from 15 patients (Figure 2, Figure S8E and Figure S9). We also added pan-cytokeratin (PanCK) to define the tumor cells and then determined the high or low expression of UPP1 based on its expression on panCK+ tumor cells. For phenotypic identification and quantification, we used the Visiopharm software (<https://visiopharm.com>), a commercial provider of quantitative digital pathology solutions, for data analysis (Line 798-814). The cells were identified using the incorporated AI-based recognition algorithm. We then utilized the phenotype module provided by Visiopharm software to distinguish UPP1^{high} and UPP1^{low} tumor cells. We also conducted spatial distance analyses to verify the spatial distribution of UPP1^{high} tumor cells and their relationships with other cells in the tumor microenvironment. We believe that these modifications present a more accurate and comprehensive view of the role of UPP1 in the tumor microenvironment.

15. Line 394- what are the prominent features of CAFs?

Response:

Thank you very much for this comment. Cancer-Associated Fibroblasts (CAFs) are a significant component of the tumor microenvironment, known for their role in cancer progression. They are characterized by the expression of specific markers, including α -SMA, Fibroblast Activation Protein (FAP), and Platelet-Derived Growth Factor Receptor (PDGFR) α/β . Functionally, CAFs are involved in remodeling the extracellular matrix (ECM), promoting angiogenesis, and enabling tumor cell invasion and metastasis(13).

In the manuscript, we identified a subset of MMP11+ fibroblasts(14). Functional enrichment of this subset was positively associated with CAFs-related biological processes such as collagen/matrix formation and epithelial-mesenchymal-transition (EMT), while negatively correlated with immune-related processes (Figure S2K and Figure S7E). This subset of cells also highly expressed key CAF markers, such as COL1A1 and FAP. Our intention was to convey that these cells exhibit characteristics typical of CAFs. However, we realized that our choice of words might have been inappropriate, and we have made amendments in the manuscript accordingly.

Furthermore, we have restructured the presentation of CAFs within our paper. We no longer presented them in a separate figure, but instead focused the article on UPP1. We introduced this

type of cell in the context of the correlation between UPP1^{high} tumor cells and the tumor microenvironment, and then carried out subsequent analyses (Figure 2, Figure S7-S9).

16. Line 409- markers such as?

Response:

Thank you very much for pointing out this. We apologize for not providing specific explanations for the gene-sets used in our previous manuscript. In the revised version, we added a new Table S3 to include the specific markers used in our study. We also added the description of these gene-sets in the manuscript when we used them for functional analysis. Additionally, we included the sources of the gene sets in the figures (Figure S5C, S7E, Figure 3O, Figure 4F). We hope these modifications adequately address your concerns

17. Line 431- α SMA is not a CAF marker?

Response:

Thank you very much. According to previous literature reports, α SMA is often used as a marker for CAFs. It is expressed in myofibroblasts, a type of activated fibroblast that is commonly found within the tumor microenvironment, and is thus frequently associated with CAFs(1, 15). Of course, some studies also considered FAP as a CAF marker(16). After considering your suggestions, we have made modifications to our manuscript. In Figures 3P and 5N, we have used an increase in FAP as an indication of CAFs. Moreover, in the single-cell analysis, we are not limited to using α SMA; we also included multiple markers, including FAP, etc., to define CAFs (Figure S2L, Figure 3L). We hope these modifications adequately address your concerns.

18. Line 602- please explain these manual adjustments.

Response:

Thank you very much. Initially, during the clustering process, we noticed that some cell clusters we identified were relatively small, with approximately around 100 cells. Upon conducting the differential gene analysis, these cells demonstrated a high similarity of differential genes with other cell clusters. In a scenario with a lower 'resolution parameter, these two groups of cells were clustered together, so we manually merged these two groups of cells.

However, after carefully reading your valuable comment, we noticed that there may have been instances of over-clustering in our initial approach. As mentioned in the above responses, we have revised our clustering parameters and conducted a reanalysis to prevent potential over-clustering. In this revised version, no manual adjustments were performed.

19. Line 603- considering you are over-clustering, how are you defining canonical marker genes for subclusters which are largely unknown and tissue/sample/disease dependent?

Response:

Thank you very much for this comment. The term "canonical marker genes" referred to here were the marker genes published in previous literature or commonly recognized marker genes, such as CD45, CD3, FOXP3, PDCD1, or SPP1. In the previous version of the article, what we wanted to express was that some of the cell populations we obtained expressed classic known cell markers. For example, in the cluster analysis of CD8T cells, some of the cells we obtained expressed CCR7/SELL or PDCD1. We named these cells with these well-known markers. For some other cell

populations, their highly differentially expressed genes were not classic well-known cell marker genes, we used the top differential gene to name these cells.

In our revised version, we have reduced the resolution of our clustering to prevent over-clustering. In terms of cell naming, we followed the approach used in previously published papers, adopting the “cluster_cell_marker” format(1). For cell populations that distinctly expressed known markers, we utilized these markers to represent these cells. For example, we named CD4T cell population that highly expressed FOXP3 as 8_CD4_FOXP3 (8 is the cluster number) (Figure S2). On the other hand, for other cell populations where the highly differentially expressed genes did not correspond to traditional cell-defining markers, we elected to name them based on the top differential gene. Following your suggestion, we have adjusted the clustering parameters to avoid over-clustering. As such, the cell populations we have now identified should be representative and reliable. We hope these modifications meet your expectations.

20. Line 815- It would be good to see the drug response of the UPP1 over-expressing cell line and knock-down cell line compared to the wild type control of that same cell line for a baseline value

Response:

Thank you very much, and this is a very valuable suggestion. Our intent in this part was to explore whether LUAD patients with a higher baseline expression of UPP1 might be more sensitive to certain existing drugs. From a clinical perspective, this could mean that for patients with high UPP1 expression, if first-line treatments could not yield satisfactory results, there might be other drugs or treatment strategies available. Therefore, we performed analyses based on drug sensitivity data(12, 17-19). In the revised manuscript, we adjusted our experiments in line with your suggestion, we conducted drug sensitivity experiments using cells with over-expressed UPP1 and their corresponding control cells. We then also repeated the organoid experiments to verify if tumor cells with high UPP1 baseline expression were more sensitive to the drugs that our analyses identified. In addition to this, we added animal experiments to further validate our hypothesis (Figure 7 and Figure S17).

21. Regarding your IHC experiments: I noticed that the staining quality and intensity of the UPP1 antibody is not that high. How are you classifying as low/mid/high expressors for your IHC scoring?

Response:

Thank you very much for this comment. It is possible that the quality of our IHC staining images may be affected due to the reduction in resolution when the images were uploaded. On the other hand, to avoid potential errors and subjectivity in staining interpretation, we have employed a more objective approach in the revised paper. We used Aipathwell, an AI-based IHC scoring software developed by Servicebio (Wuhan, China)(20, 21). This software is trained on vast amounts of data and enables us to analyze each section and obtain specific H-scores, which we then used for subsequent analysis (Table S5). We used this software to analyze the all the IHC images in the revised version of the manuscript (Figure R8).

Figure R8. Aipathwell for IHC analysis.

And for classifying high and low expression patients, we used the “survminer” R package(1, 12, 22-24). This package uses the obtained H-scores and patient prognosis information to categorize patients into high and low expression groups (Line 624-625). Meanwhile, for the analysis of immunofluorescence, we also adopted an AI-based software, Visiopharm. We hope that the utilization of these AI-based tools can significantly reduce any bias or deviation that may have been present in our previous analyses.

22. A note to both the authors and the Editor:

As mentioned above, I have reservations about the methodology of clustering and the decision to overcluster using very low expression fold change thresholds. This does not lead to biologically discernable cell types with any confidence. Also, using these sub clustered gene sets to infer population abundance from bulk data is also inaccurate. That being said, the genes that the authors pick out, especially UPP1 does show correlation with patient response and the authors have performed several validation experiments to back up their hypotheses. I believe this works only because a single gene was picked as a candidate from the analysis and not a 'cell type', which would have been inaccurate. This is a case of process vs results, and in this case the result and validation ends up over-riding sub-optimal process. I believe that isn't always a bad thing when validation is provided, but it would be remiss of me to not make note of this issue.

I suggest that the authors amend their language to avoid naming these sub-sub-clusters as 'cell types' and restrain to discussing the selected gene as important. This also applies to inferences from bulk transcriptome data.

My final suggestion would be to rethink what you are including in this manuscript. Although there is a lot of work presented here, the key findings are the role of UPP1 and it's potential significance in patient treatment, which should be presented as a cohesive story front and center, followed by a presentation of possible mechanism. The fibroblast and immune remodeling can be a part of the mechanistic hypothesis.

Overall, there are sufficient original and meaningful results presented here that merit publication.

Response:

We deeply appreciate the time and effort you have spent in reviewing our manuscript, and we are grateful for the opportunity to improve our work based on your thoughtful and detailed comments. Regarding the concerns you raised about our single-cell analysis, we have taken your comments into careful consideration and made substantial modifications accordingly. We understand and acknowledge your concerns about the over-clustering of cell populations and the use of low expression fold change thresholds. We have reanalyzed our data with adjusted parameters to reduce

over-clustering and ensure a more accurate and biologically meaningful representation of cell populations. Additionally, we have increased the expression fold change thresholds to enhance the reliability and representativeness of the gene sets obtained.

Based on the newly obtained gene sets, we recalculated cell abundance in bulk data. As for inferring the cell abundance based on the gene sets obtained from single-cell data in bulk data, this method is common in current single-cell studies. Most of the gene sets used in previous single-cell studies are based on differentially expressed genes obtained from single cells, but the criteria varied(25-27). Some studies even simply calculate the mean or median expression of these genes to represent the abundance of this group of cells in bulk data(27).

From the revised results, we could still identify the association between UPP1^{high} tumor cell cluster and patients' prognosis. On this basis, we further evaluated the correlation between the top 10 differential genes in this group of cells and the functional characteristics of this group of cells to assess which genes contribute to and are important for the functional behaviors of this group of cells (Figure 1G). And UPP1 emerged as the top-ranked gene in this analysis, suggesting that this group of cells is characterized by high expression of UPP1, and UPP1 is also an important factor in the functional behavior of this group of cells. The subsequent experiments also confirmed this point, changes of the UPP1 expression in tumor cells have a significant impact on the biological behaviors of tumor cells, and also affect the microenvironment.

Under your comments and those of other reviewers, we reconsidered our research and made substantial changes to the article. In the revised article, we concentrated more on the role of UPP1. In the first part of the article, we identified a group of tumor cells with the higher expression of UPP1 based on the integrative single-cell analysis, then we verified the association between UPP1 and patient prognosis through patient samples. Further, we explored the link between UPP1 and the tumor microenvironment, validating our analysis results through cell-cell communication analysis and multiplex immunofluorescence. We also studied the impact of UPP1 expression changes on tumor cell function and their effects on other cells. These findings were further verified in vitro and in vivo through CyTOF analysis. Lastly, we investigated whether inhibiting UPP1 expression could enhance the efficacy of immunotherapy, and the potential of UPP1 as a new target for tumor treatment. We hope that the comprehensive modifications we have made to the article meet your expectations and address your concerns. If there are any further issues, we sincerely hope you can point them out and give us the opportunity to revise further.

Reviewer #2, expertise in LUAD models incl. organoids (Remarks to the Author):

The authors set out to identify cancer subpopulations associated with poor prognosis, and started by integrating multiple scRNAseq datasets of lung adenocarcinoma (LUAD). Of 10 malignant cell subpopulations that were associated with poor prognosis and metastasis, the UPP1+ population was selected for further investigation. The authors show that UPP1 expression is associated with poor prognosis, and that knockdown of UPP1 in cell lines impairs their proliferation and invasiveness, suggesting a tumour-intrinsic fitness effect of UPP1 levels.

The authors then investigate the association of UPP1 with anti-tumor immunity. However, the claim that UPP1 expression is associated with an immunosuppressive microenvironment has limited support (see major comments below). While its association with response to checkpoint inhibition (Fig. 3F) is interesting, this is seen in only one dataset. The characterization of the tumor microenvironment is based on difficult to interpret scores (e.g. Fig. 3B) and it remains rather speculative how the TME from UPP1^{high} and -low patients now really differs. It appears that modulating UPP1 levels can regulate PD-L1 expression, but the mechanism remains unclear and the association with PI3K/Akt signaling is correlative. All in all, it remains unclear whether UPP1 truly plays an important immunomodulatory role and is causal to this. Since this is the core message of the paper, this claim needs more solid support.

In the second half of the paper the authors explore the relationship of UPP1+ tumor cells with CXCL13+ T cells and fibroblasts. There are issues with the analyses (see major comments below) that limit the value of these data.

The authors should be complimented on the volume of the work, and their efforts to combine bioinformatic analyses with experimental validation. Yet there are several major issues that need to be addressed to convincingly present this story.

Response:

We deeply appreciate your thorough evaluation of our study and the constructive comments you have provided. Your expertise has given us valuable insights that undoubtedly enhanced the quality of our work. Following your comments, we have tried our best to make substantial revisions to our manuscript. We hope these changes adequately address your concerns and further strengthen our study. If there are any additional issues, we sincerely hope you can point them out and give us the opportunity to revise further.

Major comments

1. - The figure legends provide very limited information on the analyses performed. It is frequently unclear how a metric has been defined or how the analysis was done. Graphs are poorly defined (e.g. number of replicates, statistical tests used). More detailed information is needed to allow interpretation of the data.

Response:

Thank you very much for pointing out the lack of detail in our figure legends and we sincerely apologize for any confusion this may have caused. We revised our figure legends to include more specific details about the analyses that were performed, including clear definitions, thorough explanations of the analytical methods, and the number of replicates for each experiment. We also specified the statistical tests used for each analysis, providing a solid foundation for interpreting our

results. We hope these revisions would enhance the clarity and comprehensibility of our data presentations.

2. - In Fig. 2C associations between PMRT and various gene sets are reported but while significant the effect size is marginal. It is unclear why these gene sets were selected. Were these part of a larger collection and are only the significant results reported? Is this corrected for multiple hypothesis testing?

Response:

Thank you very much for your careful review and this valuable comment. After carefully reading your comments and those from other reviewers, we realized that our initial research focus may have been unclear, and there was a significant lack of substantiation for our core content. In the revised manuscript, we have concentrated our research focus on studying the role of UPP1 and its impact on the tumor microenvironment. The earlier inclusion of PMRT and related analysis may not have been appropriate, as it was not the main focus of our article. Therefore, we have removed this part from the revised manuscript.

Regarding the use of gene sets, we have added **Table S3** to specifically display which gene sets we selected, along with their references and where they were obtained. In the revised manuscript, we only conducted functional analyses in **Figure 3O**, **Figure 4F**, **Figure S5C**, **Figure S7E**, and **Figure S8C**.

For **Figure 3O**, **Figure 4F**, **Figure S5C**, and **Figure S7E**, we used the gene sets from MSigDB database (<https://www.gsea-msigdb.org/gsea/msigdb>). For **Figure S8C**, we used the immunosuppressive related gene sets, which was compiled by Xue et al. from the research conducted by Bagaev et al. and Xie et al.(1, 28, 29).

Figure 3O, **Figure 4F**, **Figure S7E**, and **Figure S8C** were conducted using GSVA method, which is a common gene signatures based enrichment method(30). The purpose of **Figure 3O**, and **Figure S8C** was to compare certain well-known biological processes between two groups, thus multiple hypothesis testing was not appropriate and not conducted in these cases. The aim of **Figure 4F** was to use GSVA method to rank the pathways related to PD-L1 expression in UPP1^{high} tumor cells(31), thus statistical analysis is not applicable. **Figure S7E** was conducted using GSVA analysis, the terms were included because these were the well-known biological processes related to CAFs(13). Multiple hypothesis testing in this case was conducted using Bonferroni correction, and terms with adjusted p-values less than 0.05 were considered significant. For **Figure S5C**, this was performed using AUCCell method(32), and the detailed response to this method was summarized in the response to the minor comment #15.

In the Methods section and figure legends of the revised manuscript, we have also added detailed descriptions of the functional enrichment analysis. Additionally, we also included the sources of the gene sets in the figures. We appreciate your comment and hope that these revisions have clarified and improved the presentation of our work.

3. - The claim that UPP1 is associated with an immunosuppressive tumor microenvironment (TME) is insufficiently substantiated. Metrics for immunosuppression (e.g. in Fig. 3B) are poorly defined. The authors have a wealth of single cell sequencing data available yet choose to analyse the TME on bulk RNA sequencing datasets. Moreover, the data in Fig. 3E (T cell enriched subset) is not very

different compared to the data in Fig 2D-E (overall cohort) and data showing that UPP1 has limited prognostic value in T cell depleted patients is lacking. Overall, this claim needs stronger support.

Response:

Thank you very much for this valuable comment and for pointing out our shortcomings. We agreed that we did not fully substantiate the relationship between UPP1 and the immunosuppressive microenvironment in the previous version of the manuscript. Previously, we only focused on the association between UPP1 and exhausted CD8 T cells.

Following your recommendation, we have now re-analyzed the relationship between UPP1^{high} tumor cells and the TME using scRNA-seq data. We constructed a UPP1^{high} tumor cells related co-occurrence network based on the methods introduced in a recent publication(1). This analysis revealed that UPP1^{high} tumor cells was closely related to four groups of immunosuppressive and tumor progression related cells within the TME, including CD4+FOXP3+ Tregs, CD8+PDCD1+LAG3+ exhausted T cells, SPP1+PDL1+ M2 macrophages, and MMP11+ CAFs. In addition, we validated this association using bulk RNA sequencing datasets (Figure S7). Furthermore, to provide deeper insights into the intercellular communications within the TME, we investigated the ligand-receptor networks and the cytokine regulation relationships among these cells (Figure S8). These findings suggested that UPP1^{high} tumor cells could be interacted with these cells in the tumor microenvironment and the interactions among these cells contributed to the establishment of an immunosuppressive tumor microenvironment.

To support these observations, we further conducted multiplex immunofluorescence staining and used the Visiopharm software (<https://visiopharm.com>) for the unbiased spatial relationship analysis (Line 798-814). We found that UPP1^{high} tumor cells were primarily located at the invasive margins of the tumor and had closer spatial proximity with these cell groups (Figure 2 and Figure S9). These results provided substantial evidence supporting the association between UPP1 and the immunosuppressive microenvironment.

The cell-cell communication network analysis revealed broad engagement of cytokine regulations in the interactions between UPP1^{high} tumor cells and other cells. The cytokine array results further (Figure 3A) indicated that UPP1 upregulation in tumor cells could alter the expression of various cytokines, including TGF- β , a potent molecule linked to immune suppression and the regulation of the immunosuppressive tumor microenvironment. We further confirmed that TGF- β played a significant role in UPP1-mediated immune suppression (Figure 3). Additionally, we also performed in vivo experiments and CyTOF analysis to substantiate the relationship between UPP1 and immune suppression (Figure 5 and Figure S12-S15).

In regard to the previous prognostic analysis involving UPP1 in T-cell enriched patient samples, we've taken into account your advice, as well as the comments from other reviewers. We acknowledge that this analysis may not have been appropriate, and, as such, we've decided not to include these results in the revised manuscript. Instead, the main focus of our revised manuscript is to experimentally substantiate our claims. We hope these additional experiments, conducted under your guidance, would satisfactorily address your concerns.

4. - The heading of the third section (“UPP1 contributed to the immune suppression of CD8+ T cells by regulating PD-L1 expression”) is insufficiently substantiated. The point that UPP1 regulates PD-L1 expression is convincing but whether this is responsible for immunosuppression of CD8+ T cells is not proven. In fact in Fig. 4B the added effect of PD-L1 inhibition is minimal and most of the

therapeutic effect comes from inhibition of UPP1 alone. This is in line with the data in Fig. 2G-H which suggests that there is a tumor-intrinsic effect of UPP1 inhibition which is unrelated to immune suppression. Would the effect of UPP1 knockdown on tumour growth be similar in immunodeficient mice or in mice in which T cells are depleted?

Response:

Thank you very much for this valuable comment and guidance. We recognized that the title of this section in our initial manuscript may have overstated the role of UPP1. We have restructured and revised our manuscript accordingly.

Regarding the influence of UPP1 on tumor proliferation. In the previous version of the manuscript, we may not have fully considered this aspect. In the revised paper, we found that inhibiting UPP1 indeed exerts a certain suppressive effect on tumor proliferation (Figure S6), which is consistent with findings from the recent published literature(33). However, to substantiate a connection between UPP1 and tumor immune regulation, beyond just a tumor-intrinsic effect, we have performed additional animal experiments to verify this (Figure S12).

Tumor cells with UPP1 over-expression and the corresponding control cells were separately subcutaneously injected into both immunocompromised (Nude) and immunocompetent (C57) mice. Under the same treatment conditions, we observed that when UPP1 was not upregulated, there was a partial suppression of tumor proliferation in the immunocompetent mice compared to the immunocompromised mice. However, when UPP1 was upregulated, the tumor in the immunocompetent mice were able to maintain a growth pattern similar to that in the nude mice, suggesting that even in the presence of immunity, the tumors can still maintain their original growth dynamics after UPP1 upregulation, indicating a potential scenario of immune evasion (Line 304-316).

On the other hand, when UPP1 expression was inhibited, we noted that while the downregulation of UPP1 led to a certain degree of tumor growth suppression in the nude mice, this inhibitory effect was significantly more pronounced in the immunocompetent mice. This observation further indicated the potential relationship between UPP1 and tumor immunity. Moreover, to further examine the relationship between UPP1 and the immunosuppressive microenvironment, we carried out a CyTOF analysis. This additional analysis reinforced our findings that the upregulation of UPP1 can facilitate the establishment of an immunosuppressive microenvironment (Figure 5 and Figure S12-S15).

As for the relationship between UPP1 and the suppression of CD8 T cell functionality. We found the use of anti-PD-L1 antibodies could partially restore the cytotoxic function of CD8 T cells in the co-culture of CD8 T with UPP1 up-regulated tumor cells (Figure 4I and Figure S11B-G), suggesting that UPP1 could influence the cytotoxic ability of CD8 T cells via PD-L1. Furthermore, we also transplanted LLC-OVA cells into OT-1 mice (Figure S16B-D). We observed that the inhibition of UPP1 significantly affected tumor growth capability in the OT1 mice.

We hope these revisions could provide a more comprehensive view of the role of UPP1 in tumor growth and immune regulation and addressed your concerns. As for the specific regulatory mechanism between UPP1 and PD-L1, a detailed explanation is provided in response to the next question.

5. - The mechanistic link between UPP1 and PD-L1 is insufficiently clear.

Response:

Thank you very much. In the revised manuscript, we have conducted additional experiments to further elucidate the regulatory mechanisms between UPP1 and PD-L1 (Line 276-287). As shown in Figure 4, after upregulating UPP1, we examined the changes in PI3K/AKT/mTOR using Western blotting. Subsequently, upon suppression of UPP1, we treated the samples with SC79 (AKT activator) and again detected the changes in AKT/mTOR via Western blotting. These results suggested that UPP1 could influence PD-L1 expression through the PI3K/AKT/mTOR pathway.

6. - The relationship between UPP1^{high} tumor cells and CXCL13^{low} T cells is poorly supported. The data is mostly correlative (e.g. Fig. S8E), based on small and difficult to interpret differences between CXCL13^{high} and -low cells (Fig. 5G-H), without a comparison to UPP1^{low} cells (Fig. 5G-H) and based on inappropriate gating (Fig. 5K-L).

Response:

Thank you for very much. After carefully considering your comments as well as those from other reviewers, we have conducted a comprehensive revision of our data analysis process. In the revised manuscript, the primary focus has been shifted to comprehensively discuss and validate the role and mechanisms of UPP1. Given this change in focus, the previously discussed relationship between UPP1^{high} tumor cells and CXCL13 T cells, which was less conclusive, was no longer central to our study. As such, we have decided not to include this content in the modified version of our manuscript.

7. - Throughout the paper, the flow cytometry results are of low quality. Gates regularly cut through populations and appropriate controls to show the rationale of the placing of gates are missing. This significantly limits the interpretation of these data.

Response:

Thank you very much for this constructive comment and pointing out the problems in our flow cytometry data. To address these concerns, we repeated all our flow cytometry experiments and performed FMO to guide our gating (Figure S10C-D, Figure S11H, Figure S15, and Figure S16A). We hope these changes would enhance the quality and interpretability of our flow cytometry data. We are grateful for your valuable comments which has guided these improvements.

8. - There are major gaps in the fibroblast story. The authors identify a fibroblast subpopulation associated with poor prognosis, but its relationship to UPP1^{high} (as set apart from UPP1^{low}) tumor cells is unclear. It is unclear whether the populations enriched after co-culture with UPP1-oe tumor cells bear resemblance to the poorly prognostic population identified above. The relationship to immunosuppression is based on correlative data (Fig. 6L) or data of poor quality (Fig. 6M).

Response:

We sincerely appreciate this invaluable comment pointing out the problems in our initial manuscript, and we acknowledge that our previous version did not provide a clear and coherent story on this part. To address this, we have re-analyzed the relationship between UPP1 and the tumor microenvironment and found a significant correlation between UPP1^{high} tumor cells and MMP11+ CAFs (Figure S7A). To further substantiate these findings, we conducted multiplex immunofluorescence staining on patient tissues, which demonstrated a closer spatial distance between UPP1^{high} tumor cells and MMP11+ fibroblasts as compared to UPP1^{low} tumor cells (Figure 2 and Figure S9). Following this, we also re-analyzed the single-cell data from the co-culture experiment and revealed that the upregulation of UPP1 in tumor cells can promote the transition of

fibroblasts towards a CAF phenotype (Figure 3J-P). In the subsequent in vivo analysis, we also validated that the upregulation of UPP1 can increase the infiltration of CAFs (Figure 5L-N). In the revised manuscript, we have integrated the fibroblast-related findings throughout the paper, rather than presenting them in a separate figure. We hope these modifications could more effectively illustrate the findings and address your concerns.

Minor comments

9. - Fig. 1C: what do 'mutated' and 'wild type' refer to?

Response:

Thank you very much for this question. Some of the scRNA-seq datasets we included in this study only provided information on whether there was a mutation or not, without specifying the exact mutation (such as EGFR mutation, TP53 mutation, etc.). To maintain consistency in data presentation, we previously categorized all patients with any type of mutation, such as EGFR or TP53 mutations, under the 'mutated' label. However, to prevent any potential misunderstandings, we have made adjustments in the revised manuscript to present specific types of mutations (Figure S1A and S1B). And the aim of this part of data visualization was to present the characteristics of the patient population in the study.

10. - Fig. 1E: the NK cell cluster also expresses CD3 markers that are shared with the T cell cluster. Has this cluster been correctly assigned?

Response:

Thank you very much for this question. The single-cell data analysis in this study is based on the integration of previously published single-cell data. According to the original publications, the identified NK cells also expressed CD3 (Figure R9-R10, see below)(2, 7). While the NK cells we identified appear to express CD3, this expression is relatively low. Importantly, these NK cells prominently express markers of NK cells, such as KLRF1, underscoring their distinct NK cell features (Figure S1C-D). Therefore, we believe that the delineation of cell identities in our study is accurate and valid.

CANCER

Decoding the multicellular ecosystem of lung adenocarcinoma manifested as pulmonary subsolid nodules by single-cell RNA sequencing

Xudong Xing^{1,2*}, Fan Yang^{3*}, Qi Huang^{3,4†}, Haifa Guo³, Jiawei Li³, Mantang Qiu^{3‡}, Fan Bai^{5,6‡}, Jun Wang^{3‡}

Lung adenocarcinomas (LUAD) that radiologically display as subsolid nodules (SSNs) exhibit more indolent biological behavior than solid LUAD. The transcriptomic features and tumor microenvironment (TME) of SSN remain poorly understood. Here, we performed single-cell RNA sequencing analyses of 16 SSN samples, 6 adjacent normal lung tissues (nLung), and 9 primary LUAD with lymph node metastasis (mLUAD). Approximately 0.6 billion unique transcripts were obtained from 118,293 cells. We found that cytotoxic natural killer/T cells were dominant in the TME of SSN, and malignant cells in SSN undergo a strong metabolic reprogram and immune stress. In SSN, the subtype composition of endothelial cells was similar to that in mLUAD, while the subtype distribution of fibroblasts was more like that in nLung. Our study provides single-cell transcriptomic profiling of SSN and their TME. This resource provides deeper insight into the indolent nature of SSN and will be helpful in advancing lung cancer immunotherapy.

Figure R9. Expression of marker genes in HRA000154.

Single-cell RNA sequencing demonstrates the molecular and cellular reprogramming of metastatic lung adenocarcinoma

Nayoung Kim^{1,2,3,13}, Hong Kwan Kim^{4,13}, Kyungjong Lee^{5,13}, Yourae Hong^{1,6}, Jong Ho Cho⁴, Jung Won Choi⁷, Jung-Il Lee⁷, Yeon-Lim Suh⁸, Bo Mi Ku⁹, Hye Hyeon Eum^{1,2,3}, Soyeon Choi¹, Yoon-La Choi^{6,10,11}, Je-Gun Joung¹, Woong-Yang Park^{1,2,6}, Hyun Ae Jung¹², Jong-Mu Sun¹², Se-Hoon Lee¹², Jin Seok Ahn¹², Keunchil Park¹², Myung-Ju Ahn¹² & Hae-Ock Lee^{1,2,3,6}

Figure R10. Expression of marker genes in GSE131907.

11. - Fig. S1C/D: Can the authors explain why virtually no T cells and NK cells are represented from the Wu et al. cohort?

Response:

Thank you very much. The total number of cells in the Wu et al. cohort was relatively small, with only around 17,000 cells. Besides, the identified number of NK and T cells in this cohort was even smaller, as reported in the original study(6). In our manuscript, we were able to identify only a few hundred NK and T cells in this cohort. In Figure S1F, NK and T cell populations actually are present in this proportion diagram (Figure R11, See below). However, due to their relatively low proportions, they are not prominently visible in the image. To better illustrate this point, we have added Figure S1G and Figure S3 to display the cell population distributions and the precise cell count for each cell population. This additional information should provide a clearer understanding of the representation of different cell types, including NK and T cells, within the Wu et al. cohort.

Figure R11. NK and T cells in Wu et al. cohort.

12. - Table S2 only contains marker genes for the CD8+ T cell cluster

Response:

Thank you very much. we apologize for any confusion caused by the initial presentation of the data. Marker genes for different cells are located in different sheets within the table. To improve clarity and ease of navigation for readers, we have updated the table's format. Information about the sheets has been added to the first page of the table. The updated information is included in Table S2.

13. - Line 186: “expression of UPP1 is higher in metastatic brain and lymph node” In Fig. S3A it does not look like the expression in brain is higher than in the primary tumor. No significance testing has been performed.

Response:

Thank you very much for pointing this out. We have reanalyzed and revised the display of marker genes of prognosis-related tumor clusters. Furthermore, we have conducted statistical analyses to confirm these findings (Figure S5A-B).

14. - Fig S3B: on what basis were these genes chosen? It seems a random subset of immune escape genes. Has this analysis been performed in an unbiased fashion, or were only significant immune escape genes reported?

Response:

Thank you very much for this question. The previous Fig S3B was no longer the main focus of the revised manuscript and thus not included. As for the gene sets, the immune escape gene sets were collected from the MSigDB database (LIN_TUMOR_ESCAPE_FROM_IMMUNE_ATTACK) (<https://www.gsea-msigdb.org/gsea/msigdb>). Regarding the use of gene sets, we have included a new Table S3 to specifically display which gene sets we selected, along with their references and where they were obtained.

15. - Fig. 2C: are the genes defining these pathways defined in the methods? How was this analysis carried out? AUC seems an unusual metric for this. Even though the results are statistically significant, the effect size is minimal. Can the median score be reported to allow a better interpretation of the effect size? Also: how were these pathways selected? Were these part of a larger collection of gene sets, and has any multiple hypothesis correction been performed?

Response:

Thank you very much. we utilized the AUCCell R package to compute the AUC score(30), which is a key analysis step in the widely-used single-cell analysis method, SCENIC(32). As outlined in the official documentation for this analysis method, calculating the mean or median expression of a gene signature might introduce bias, especially for single-cell data. This is largely due to the possibility of cells with higher read counts (attributable to either technical or biological reasons) presenting higher average values for most signatures. To mitigate this effect, library-size normalization is typically required. The AUCCell method, by individually evaluating the detection of signatures on each cell, automatically compensates for library-size variations. This makes its usage more straightforward and independent of the units of the data and other applied normalizations (like raw counts, TPM, UMI counts). Considering the complexities of single-cell data, including the large volume and the significant inter-cell variations, it would be better to use analysis tools specifically developed for single-cell data.

As to previous Figure 2C, as stated in response to major comment 2, we have concentrated our research focus on studying the role of UPP1 and its impact on the tumor microenvironment. The PMRT and related analysis have been removed from the revised manuscript.

In the revised manuscript, **Figure 3O**, **Figure 4F**, **Figure S7E**, and **Figure S8C** were conducted using GSVA method. For the selection of gene sets in these figures, because our research aim was to compare specific biological processes, therefore, well-acknowledged and widely accepted gene sets were used.

For **Figure S5C**, we employed the AUCCell method to calculate the AUC scores of all gene sets described in **Table S3 (Line 629-647)**. Subsequently, we performed differential analysis based on these AUC scores, following the principles of the *FindAllmarker* function of Seurat. Here, multiple hypothesis testing was performed using Bonferroni correction, and terms with adjusted p-values less than 0.05 as significant. Moreover, through differential analysis, we obtained Log2AUC values (which we defined this as the enrichment scores). Based on these Log2AUC values, we ranked the gene sets. After ranking, we selected and displayed the top gene sets that were not only high-ranking but also relevant to tumor progression and in line with the objectives of our study (**Table S4**).

We deeply appreciate your comments and we hope these explanations could address your concerns.

AUCCell documentation:

<https://www.bioconductor.org/packages/devel/bioc/vignettes/AUCCell/inst/doc/AUCCell.html>

16. - Figure S3: the level of siRNA mediated UPP1 knockdown is minimal (Fig S3G), yet the effects on invasion and colony formation are quite strong. Are the authors confident that this is not due to a possible off target effect? Showing this with two independent siRNAs would significantly strengthen this observation

Response:

Thank you very much for this constructive comment. To address your concerns, we have adjusted the conditions for siRNA transfection and have verified the knockdown experiments using two sets of siRNA sequences to avoid potential off-target effects. This approach should significantly strengthen our observations and provide more confidence in the effects we reported on invasion and colony formation following UPP1 knockdown (Figure S6 and Table S8). We hope these changes have improved the reliability of our work. We appreciate your suggestion and hope this adequately addresses your concerns.

17. - Fig. 3B: why was this analysis performed on the bulk RNAseq datasets? The correlation is significant but weak. Is a similar trend seen in the single cell RNAseq datasets? Also: what is the definition of the 'T cell inhibitory score'?

Response:

Thank you very much. After carefully considering your comments and those from other reviewers, we revised our data analysis approach. Most of the results were now based on single-cell analysis and then bulk data were mainly used for verification. Consequently, the "T cell inhibitory score" section has been removed from the revised manuscript.

18. - Fig. 3C: is the association with OS only seen for hot tumors? What about cold tumors?

19. - Fig. 3D: the figure is presented in a confusing way. Why not make two separate panels for CTLtop and CTLbottom patients, and then show the effect of UPP1 levels instead? That would make it more clear that the effect is most pronounced in the CTLhigh patients.

Response:

Thank you very much. For these two comments, in regard to the previous prognostic analysis involving UPP1 in T-cell enriched patient samples, we've taken into account your advice, as well as the comments from other reviewers. We acknowledge that this analysis may not have been appropriate, and, as such, we've decided not to include these results in the revised manuscript.

20. - Line 230: "UPP1 was associated with the immune suppressive microenvironment": this claim is overstated based on the presented data. There is no real characterization of the microenvironment aside from PD1 levels on T cells. Also see the point above.

Response:

Thank you very much for your valuable comments on the association between "UPP1 and the immune suppressive microenvironment" in our manuscript. We recognized that our previous statement might have been overstated and we did not conduct enough experiments to support our claims. To further substantiate that UPP1 was associated with the immune suppressive microenvironment, additional experiments were conducted in the revised manuscript.

1. The relationship between UPP1^{high} tumor cells and the Tumor Microenvironment (TME) cell populations was re-analyzed using single-cell RNA-sequencing (scRNA-seq) data.
2. A co-occurrence network was constructed, revealing that UPP1^{high} tumor cells were closely related to four groups of immunosuppressive and tumor progression-related cells within the TME (Figure S7).
3. Multiplex immunofluorescence staining was conducted for spatial relationship analysis and Visiopharm software was used for unbiased analysis. The analysis revealed that UPP1^{high} tumor

cells were primarily located at the invasive margins of the tumor and had closer spatial proximity with other cell groups (Figure 2 and Figure S9).

4. A cell-cell communication network analysis revealed broad engagement of ligand-receptor and cytokine regulations in the interactions between UPP1^{high} tumor cells and TME cells (Figure S8A).
5. It was confirmed that TGF- β , a key molecule linked to immune suppression, played a significant role in UPP1-mediated immune suppression (Figure 3).
6. In vivo experiments and CyTOF analysis were performed to substantiate the relationship between UPP1 and immune suppression. More markers were used (Figure 5).

We hope these additional experiments could address the concerns raised and to provide a more robust foundation for the association between UPP1 and the immune suppressive microenvironment. We appreciate your valuable comments, which has significantly contributed to the enhancement of our manuscript.

21. - Fig. S5F: this is anecdotal evidence – can the authors, with the support of a pathologist blinded to UPP1 status, score immune exclusion phenotypes and then perform this on their TMA cohort more broadly?

Response:

Thank you very much for this comment. We realized phenomenon of immune exclusion cannot be conclusively demonstrated solely through immunohistochemistry (IHC) as pointed out by reviewer 1. We recognized that our previous manuscript may have overstated these results. Therefore, we have decided to remove the description of immune exclusion from our paper. For TMA and IHC interpretation, to avoid potential errors and subjectivity, we have employed a more objective approach in the revised paper. We used Aipathwell, an AI-based IHC scoring software developed by Servicebio (Wuhan, China) (20, 21). This software is trained on vast amounts of data and enables us to analyze each section and obtain specific H-scores (Line 713-716), which we then used for subsequent analysis (Table S5). We used this software to analyze the all the IHC images in the revised version of the manuscript.

22. - Without a quantification of the proximity of UPP1+PD-L1+ tumor cells and PDCD1+CD8+ T cells the data in Fig. 3I is anecdotal and does not add to the story.

Response:

Thank you very much for this comment. After carefully considering your comments and those from other reviewers. We thought this part was no longer the key focus of our study. Therefore, we have decided to remove this section from our paper. As for the phenotypic identification, quantification, and spatial analysis, we used the Visiopharm software in the revised manuscript.

23. - Fig. S5G-H: the shift in MFI (particularly for the overexpression experiment) is limited – quantification on replicate experiments and significance testing is required. The absence of strong effect may be due to the low level of overexpression and knockdown as evident from the Western blots.

Response:

Thank you deeply for this comment. For the establishment of cell lines, we've refined our protocols and carried out additional rounds of screening with puromycin. In the revised manuscript, we

confirmed the efficacy of overexpression and knockdown using both RT-qPCR and Western blotting (Figure S10A, Figure S11C, Figure S12A, Figure S16B, and Figure 4). For flow cytometry analysis, we repeated the experiments and also added isotype controls to ensure the reproducibility and reliability of our findings. Your insightful comment has greatly helped us improve the rigor and depth of our study. We hope these modifications address your concerns.

24. - Fig. S5K: the decrease in p-Akt signalling as a consequence of UPP1 knockdown is convincing but this does not prove that PD-L1 is regulated through an UPP1-PI3K-Akt axis as it is correlative data.

Response:

Thank you very much for this comment. We have conducted additional experiments to further elucidate the regulatory mechanisms between UPP1 and PD-L1. As shown in Figure 4, after upregulating UPP1, we examined the changes in PI3K/AKT/mTOR/PD-L1 using Western blotting. Subsequently, upon suppression of UPP1, we treated the samples with SC79 (AKT activator) and again detected the changes in AKT/mTOR/PD-L1 via Western blotting. These results suggested that UPP1 could influence PD-L1 expression through the PI3K/AKT/mTOR pathway.

25. - Fig. 4F: the gating strategy for these plots is unclear. For most populations the gates do not clearly identify separated populations but rather cut through the main population. Was there a fluorescence minus one (FMO) control to set the gates?

Response:

Thank you very much. To address these concerns, we repeated all our flow cytometry experiments and performed FMO to guide our gating (Figure S10C-D, Figure S11H, Figure S15, and Figure S16A). We hope these changes would enhance the quality of our flow cytometry data.

26. - In Fig. S6C-D claims are made about the cytotoxicity of T cells but this is an indirect measure. Did the authors perform killing assays to directly measure whether these T cells are more potent in killing target cells? The same gating issues apply as in the comment above.

Response:

Thank you very much. After carefully considering your suggestions, we have re-conducted this part of the experiments, focusing on whether the upregulation of UPP1 in tumor cells would directly impact the cytotoxic ability of T cells. In the revised version of our manuscript, we performed co-culture experiments with UPP1-upregulated LLC-OVA tumor cells and OT1 cells (Figure S11E-G). We used colony experiments and flow cytometry counts to determine the proportion of tumor cells surviving after co-culture. Additionally, we detected the markers of T cell cytotoxicity through flow cytometry (Figure 4I), and FMO was performed to guide the gating (Figure S11H). In subsequent in vivo experiments, LLC-OVA tumor cells with different UPP1 expression levels were transplanted into OT1 mice and the tumor proliferation rates were calculated and compared (Figure S16B-C). We hope that these modifications could adequately address your concerns.

27. - Fig. 4J-K: quantification of the NucView signal is complicated by the fact that there is no clearly defined positive and negative population – the gates cut through the main body of the population. An optimized or alternative method should be used to quantify apoptosis, ideally combined with a quantification of the number of live tumour cells left after co-culture.

Response:

Thank you very much for this comment. We deeply appreciate your suggestion to optimize our methods. In response to your suggestions, we have re-conducted this part of the experiments. We labeled LLC-OVA tumor cells with mCherry, and then co-cultured the UPP1-upregulated LLC-OVA tumor cells with OT1 cells. Following the co-culture, we used flow cytometry to record the proportion of surviving tumor cells as well as absolute counts (Figure S11G). We hope that these modifications would adequately address your concerns.

28. - Fig. S7A-C: number of replicates not reported. UPP1 knockdown already has an effect on tumour cell survival in the absence of T cells – these data should be shown as well in the bar graphs.

Response:

Thank you very much for this comment. As stated in the responses to the previous two comments, we have shifted our focus to investigate whether the upregulation of UPP1 in tumor cells would impact the cytotoxic ability of T cells. We have removed previous Fig. S7A-C in the revised manuscript. We apologize for the missing the number of replicates in the previous version of the manuscript. In the revised manuscript, we have added more specific detailed information about the analyses that were performed. This includes clear definitions, thorough explanations of the analytical methods, and the number of replicates in the figure legends.

29. - Fig. S7F: as for the other flow plots, the gating is challenging to interpret without a proper negative control.

Response:

Thank you very much. To address this concern, we have conducted FMO and included negative controls in our flow cytometry experiments in the revised manuscript. We hope these additions would improve the interpretability of the data.

30. - Fig. 5F: how is the T cell activation score defined?

Response:

Thank you very much. After careful consideration of comments, we considered this aspect is no longer central to our study. Therefore, we have decided to remove this section from our paper. These changes were intended to better make this manuscript in line with the central research findings, focusing more on experimental validation of our claims rather than extensive bioinformatic analysis.

31. - Fig. S8A: the Kaplan Meier curve is not clearly defined. When is a patient classified as CXCL13^{high} and when as CXCL13^{low}?

Response:

Thank you very much. Upon careful consideration, and given the shift in focus of our study, we've determined that CXCL13 T cells was not important in the revised paper. And we've decided not to include this content in the revised version of our manuscript.

32. - Fig. S8D: the UPP1^{high} group should be included as well as reference. Moreover, a KM curve will aid the interpretation of this data.

Response:

Thank you very much for this comment. Upon careful review of our manuscript, we considered this aspect is no longer central to our study. Therefore, we have removed this part from our paper. In the revised version of our paper, we have focused our efforts on emphasizing our core research findings. We have prioritized experimental validation of our claims over extensive bioinformatic analysis, which we believe would provide a clearer and more focused narrative.

33. - Sig. S9A/D: these data are difficult to interpret without a more detailed figure legend.

Response:

Thank you very much for this comment. In the revised manuscript, the primary focus has been shifted to comprehensively discuss and validate the role and mechanisms of UPP1. Given this change in focus, this part regarding the prognosis of immune cells was no longer central to our study. As such, we have decided not to include this content in the modified version of our manuscript.

34. - Fig. S9H: the heat map is incompletely labeled

Response:

Thank you very much for this comment. We have deleted this part in the revised manuscript.

35. - Line 379 refers to Fig. S3J which should be Fig. S9J. The authors highlight IL-10 but other ligand/receptor interactions such as TGF β / TGFBR signalling are equally strong.

Response:

Thank you very much for this comment. Your valuable suggestion regarding the potential role of TGF- β signaling in the context of UPP1^{high} tumor cells was particularly enlightening. Following your comment, we conducted a re-analysis of the relationship between UPP1^{high} tumor cells and the TME. The ligand-receptor networks and the cytokine regulation relationships indicated the involvement of TGF- β signaling in the cell-cell regulatory network between UPP1^{high} tumor cells and the TME cells (Figure S8A). Following this discovery, we carried out experimental validation which confirmed the upregulation of UPP1 could significantly improve the secretion of TGF- β in tumor cells and the secreted TGF- β played a crucial role of in the UPP1-mediated immunosuppressive mechanisms (Figure 3). We truly appreciate this constructive comment, which has enriched our study.

36. - The data on myeloid cells (Fig. S9) feels somewhat misplaced in the context of this story. The focus of this manuscript is on UPP1+ tumor cells. The authors describe myeloid cluster 8 but finish the section by stating that myeloid cluster 6 is actually more relevant in the interaction with UPP1+ tumor cells (although this is based on rather modest differences in interaction number and strength between groups).

Response:

We sincerely appreciate your comment and the time you have dedicated to evaluating our work. In response to your comments, we agreed that the data on previous Fig. S9 did not align well with the main narrative of our manuscript. Therefore, in the revised manuscript, we have decided to remove this section. We hope that this revision would help streamline our manuscript and maintain focus on our primary research findings.

37. - Fig. 6C: can the authors explain why the pseudotime trajectory seems to indicate an early \rightarrow late

direction that is opposite to the arrow and opposite to the differentiation trajectory proposed by the authors?

Response:

Thank you very much for this comment. In the revised manuscript, the fibroblast-related findings were not presented in a separate figure. Given that the pseudotime analysis is not central to our study, we have also chosen to remove this part from our manuscript.

38. - Fig. S10: in this and similar analyses, the authors only evaluate the interaction with UPP1^{high} but not UPP1^{low} tumor cells. These data therefore do not allow differentiating between an interaction with fibroblasts and tumor cells in general, or with UPP1⁺ tumor cells specifically.

Response:

Thank you very much for this comment. We have deleted this part in the revised manuscript.

39. - Fig. 6K: confidence intervals should be shown, as well as data on the other clusters (those not enriched in the UPP1^{high} group) to show that this is specific to the clusters of interest.

Response:

Thank you very much for this comment. Regarding the single-cell RNA sequencing analysis conducted after co-culturing UPP1 over-expressing tumor cells and control cells with fibroblasts, we refined our analysis steps to specifically focus on the proportional changes of the MMP11+CAF-like fibroblasts after co-culturing (Line 995-1002, Figure 3K-P). On the other hand, for all survival analyses in the revised manuscript, we have provided the confidence intervals.

40. - Fig. S11C: it is unclear whether these fibroblasts were co-cultured with UPP1-oe or control cells, and in any case data on both experiments should be reported.

Response:

Thank you very much for this comment. We have deleted this part in the revised manuscript.

41. - Line 409: “using the CAFs-marker genes from the aforementioned CAFs” ◊ does this refer to the markers of the cluster 0 that was reported on in Fig. 6A-B?

Response:

Thank you very much for this comment. We apologize for any confusion caused due to lack of specific explanations for the gene-sets used. In the revised manuscript, this part of analysis was moved to Figure 3K, we compared the overall characteristics of fibroblasts after co-culturing with UPP1 overexpression tumor cells or control cells. For this purpose, we used the gene markers of general CAF characteristics as reported in previous publications (Table S3).

42. - Fig. 7G: while statistically significant, the treatment effect size is limited and unlikely to having clinical significance.

Response:

Thank you very much for this comment. We understand and appreciate your concern about the clinical significance of the observed effect. Our primary intent in this portion of the study was to investigate whether LUAD patients with a higher baseline expression of UPP1 might exhibit increased sensitivity to certain existing drugs. This could potentially provide alternative treatment

strategies in cases where first-line treatments could not yield satisfactory results, particularly for patients expressing high levels of UPP1.

Indeed, this part of the analysis does not simply imply that only the drugs identified in our study could be used. Instead, it might offer a new approach for targeting UPP1 in treatment strategies and potentially create new pathways for future drug development. Furthermore, we acknowledge that the specific clinical significance of these findings must be further validated through rigorous clinical trials.

In the revised manuscript, we have made adjustments to our experimental part, we firstly conducted drug sensitivity experiments using cells with over-expressed UPP1 and their corresponding control cells. We then also repeated the organoid experiments to verify if tumor cells with high UPP1 baseline expression were more sensitive to the drugs that our analyses identified. In addition to this, we added animal experiments to further validate our hypothesis (Figure 7 and Figure S17). We hope these modifications could address your concerns.

Reviewer #3, expertise in lung cancer models, TME and immunotherapy (Remarks to the Author):

In their manuscript ‘Characterization of the immunosuppressive roles of UPP1^{high} tumor cells in the tumor microenvironment of lung adenocarcinoma and its prognostic associations’, Li et al integrate multiple datasets, including multiple lung single cell data sets, bulk RNA-seq data sets, ICI-treated datasets, and experimental data, to identify UPP1 as a possible regulator of the tumor immune interaction and mediator of outcomes. The authors assemble an impressive amount of analyses to implicate UPP1 in lung cancer biology; however, some of the conclusions seem overstated or not fully supported by the data, possible confounding features are not adequately addressed, and the presentation of the data and the associated analyses could be more clear.

Response:

We sincerely thank you for taking the time to review our manuscript and offering valuable insights into our research. Based on your constructive comments, as well as those from other reviewers, we have tried our best to address each of the points raised. We have also added more experiments to validate our claims. We hope these changes could adequately address your concerns and further strengthen our study.

Key modifications include:

1. Heeding your suggestions, we focused on elucidating the role of UPP1 with greater detail. In the first part of our manuscript, we re-analyzed the single-cell data using adjusted parameters to prevent over-clustering. We then highlighted the prognostic relevance of UPP1^{high} tumor cells. Using patient samples, we validated the link between UPP1 expression and patient prognosis. And we performed additional experiments to validate the role of UPP1 in tumor proliferation.
2. In the second part of our manuscript, the relationship between UPP1^{high} tumor cells and the TME was re-analyzed using scRNA-seq data. Through cell-cell communication analyses and multiplex immunofluorescence validation on patient samples, the tight connection between UPP1 and immunosuppressive TME was revealed.
3. In the third part of our manuscript. We expanded our experiments to further investigate how UPP1 expression variations influence the functions of tumor cells and their interplay with other cells. Key findings included the identification of TGF- β as a central molecule in UPP1-mediated immune suppression. Additionally, we discovered that UPP1 can modulate the expression of PD-L1 through the PI3K/AKT/mTOR pathway.
4. In vivo experiments and CyTOF analysis further substantiate the immunosuppressive role of UPP1.
5. We then investigated the feasibility of targeting UPP1 to enhance the efficacy of immunotherapies and the potential of UPP1 as a new target for tumor treatment.

We hope that this enhanced focus and additional experiments could address your previous concerns and add more depth to our research narrative. Once again, your comments have been instrumental in shaping our manuscript, and we're grateful for the insights you've provided.

1. In general, the cell subtyping within each compartment seems overclustered. Utilization of other metrics to assess for cluster stability and heterogeneity would help. The high degree of subclustering also doesn't help the analysis/data presentation, because it is hard follow and to know which subclusters are biologically significant.

Response:

We truly appreciate this comment regarding the potential over-clustering in our cell subtyping. We have reconsidered our analysis process and results in light of your valuable suggestions. We realized that there might indeed be some instances of over-clustering. To address this, we revisited the data and reanalyzed it.

1. Clustering Parameters: After consulting relevant literature sources, we realized that in the Seurat analysis, the resolution parameter isn't universally fixed. For instance, opting for lower resolution values, such as 0.1 or 0.4, tends to yield fewer clusters, potentially amalgamating distinct cell subpopulations. Conversely, higher resolutions like 1.5 or 2.0 might lead to a greater number of clusters, risking the over-segmentation of genuine cell subpopulations. In practical applications, a resolution range of 0.4 to 1.2 is often deemed reasonable, striking a balance between granularity and accuracy. In the revised manuscript, we tested multiple resolution parameters for cell clustering. For each resolution parameter, we compared the number of cells and UMAP distribution of cells in each cluster, as well as the differentially expressed genes of each cell cluster. We then selected a relatively stable resolution parameter, balancing the need to minimize over-clustering while preserving the biological significance of identified cells, for further analysis (Line 570-594).
2. Using tumor cells as an example: we tried resolution parameters from 0.4 to 1.5. At a parameter value of 0.5, the UPP1^{high} tumor cell group emerged distinctly. However, with a slight increase to 0.7, we noticed potential over-clustering, especially in cluster 2, which began fragmenting without evident boundaries (as illustrated below in Figure R12). With careful consideration, we felt that a resolution parameter of 0.6 offered a more accurate representation for tumor cells in the revised analysis (Figure 1D). Furthermore, we investigated whether there were significant differences in the differentially expressed genes of UPP1^{high} tumor cells under different parameters. This analysis was conducted to verify whether our previous results, obtained at a higher resolution, may have been biased. However, the results consistently showed that high UPP1 expression was a defining characteristic of this group of cells across different parameters (Figure R13). These findings suggest that although there may be differences in other cell populations in our previous version of the paper, the characteristics of the UPP1^{high} tumor cell group remained stable under different parameters. This supports the validity of our decision to focus on the UPP1 for further research.
3. Other Cell Populations (Figure S2):
 - For T cells and NK cells, selecting a parameter of 0.9 (Figure R14).
 - For myeloid cells, we gravitated towards a parameter of 0.5 (Figure R15).
 - B cells were best represented at a parameter of 0.4 (Figure R16).
 - For stromal cells, we leaned towards a parameter of 0.8 (Figure R17).

To offer greater clarity on our analytical approaches, we have provided detailed descriptions of these modifications in the revised manuscript (Line 570-594). Your invaluable guidance has been instrumental in these refinements, and we sincerely hope that our revisions could address your concerns.

Figure R12. Clustering of tumor cells under different parameters.

Figure R13. Top differential genes in UPP1 tumor cell group under different parameters.

Figure R14. Clustering of T and NK cells under different parameters.

Figure R15. Clustering of myeloid cells under different parameters.

Figure R16. Clustering of B cells under different parameters.

Figure R17. Clustering of stromal cells under different parameters.

2. More detail regarding the clustering-outcome association is needed in the methods. The authors write that ssGSEA was used to quantify the population abundance of cell types; what gene lists were used as the input genes? What was the outcome metric used in the bulk cohorts and are these uniformly calculated in these pooled datasets? Are the outcomes across cohorts relatively stable, or do they differ based on enrichment with EGFR mutant lung cancer, early stage, etc? How was the outcome analysis actually performed – Is the ssGSEA score used as a continuous variable?

Response:

Thank you very much for this comment. We sincerely apologize for not providing these crucial details in our initial manuscript. In the revised manuscript, we have added more details in methods section.

Regarding the use of gene sets for ssGSEA analysis, taking guidance from the previously published literature(27) and in line with the suggestions of Reviewer #1, the differential marker genes that displayed a LogFC > 1 and an adjusted p-value less than 0.05 were employed for ssGSEA analysis (Line 610-628). After calculating the abundance of each tumor cell group based on ssGSEA, we used the survminer R package to divide the ssGSEA score into high and low groups,

and then performed survival analysis on these groups. The use of the survminer R package followed the methods in previously published literature(1, 12, 22-24).

For the bulk cohorts, the primary outcome metric was overall survival (OS), defined as the duration from diagnosis to the event (death or last follow-up). These were uniformly calculated in these pooled datasets. Our approach to handling these data is based on methodologies described in our previously published literature(22). Furthermore, we have provided a detailed description of this process in the methods section (Line 596) and figure legends (Line 1435-1449) of the revised manuscript for added clarity.

To validate that our observed association between UPP1^{high} tumor cells and patient prognosis is stable and independent, we conducted a multivariate Cox regression analysis in response to your question. The results demonstrated that the UPP1^{high} tumor cell group is an independent prognostic factor, unaffected by gender, mutations, and disease stage (Figure R18).

Figure R18. Multivariate Cox regression.

3. How is purity accounted for in the bulk cluster analyses?

Response:

Thank you for raising this important point. We overlooked the issue of tumor purity in our initial submission. In the revised manuscript, we have now adjusted the bulk data using tumor purity.

We used ESTIMATE method to calculate the tumor purity of bulk RNA-seq data, then, we adjusted the gene expression data for tumor purity using linear regression(34). For each gene, a linear model was fitted with gene expression as the dependent variable and tumor purity as the independent variable. The residuals of the linear models, which represent the differences between the observed expression values and the fitted values predicted by the models, were calculated for each gene. This process adjusts the gene expression data for the confounding effects of tumor purity, allowing for more accurate downstream analyses (Line 612-619).

The corresponding code for this process is as follows:

```
dat=exp #expression matrix
library(estimate)
estimate <- function(dat,pro){
  input.f=paste0(pro,'_estimate_input.txt')
  output.f=paste0(pro,'_estimate_gene.gct')
  output.ds=paste0(pro,'_estimate_score.gct')
  write.table(dat,file = input.f,sep = "\t",quote = F)
  library(estimate)
  filterCommonGenes(input.f=input.f,
                    output.f=output.f ,
                    id="GeneSymbol")
  estimateScore(input.ds = output.f,
               output.ds=output.ds,
               platform = 'illumina')
  scores=read.table(output.ds,skip = 2,header = T)
  rownames(scores)=scores[,1]
  scores=t(scores[,3:ncol(scores)])
  return(scores)
}
pro='cohort'
scores=estimate(dat,pro)
if (!identical(colnames(dat), rownames(scores))) {
  stop("Sample names do not match!")
}
tumor_purity <- as.data.frame(cos(0.6049872018+0.0001467884*scores[,3]))
tumor_purity <- as.data.frame(scores[,4])
colnames(tumor_purity) <- 'Purity'
# Perform linear regression for each gene, including tumor purity as a covariate
adjusted_gene_expression_data <- sapply(1:nrow(dat), function(gene_idx) {
  gene <- rownames(dat)[gene_idx]
  finaldf_five2 <- as.data.frame(dat[gene_idx, , drop = FALSE])
```

```

finaldf_five2 <- as.data.frame(t(finaldf_five2))
expr_data <- cbind(finaldf_five2, tumor_purity)
colnames(expr_data) <- c("expression", "tumor_purity")
lm_formula <- as.formula("expression ~ tumor_purity")
lm_result <- lm(lm_formula, data = expr_data)
residuals(lm_result)
})
# Transpose the matrix to have genes as rows and samples as columns
adjusted_gene_expression_data <- t(adjusted_gene_expression_data)
rownames(adjusted_gene_expression_data) <- rownames(dat)
saveRDS(adjusted_gene_expression_data,'adjust_NG.RData')

```

We hope this amendment would address your concerns, and we're immensely grateful for your guidance on this matter.

4. The malignant cell subclustering analysis needs much more explication. First, the malignant cell UMAP should be included in Figure 2 for ease of visualization along with top expressed genes (dot plot or heatmap) to help justify the assignment of identities such as AT2-like etc. Second, the mixing of malignant cell subclusters across samples needs to be confirmed. Are these subclusters driven by specific patients or do they represent broader biology? Third, as above, how stable/distinct are these subclusters? Fourth, are any of the outcome associations actually statistically significant? Only hazard ratios are reported.

Response:

Thank you very much for your detailed comments and suggestions. We value your feedback and have made efforts to address each of the concerns you raised.

Regarding the inclusion of the malignant cell UMAP for ease of visualization along with top expressed genes, we have added UMAP and a heatmap in **Figures 1D-1E** in the revised manuscript to display the clustering of tumor cell populations and the top marker genes for each tumor cell population. Additionally, UMAP figures and top genes for other cell populations are presented in **Figure S2**.

Concerning the integration of single-cell cell data across samples, we have added **Figure S3** to illustrate this. The UMAP plots in **Figure S3A-E** color cells by Cohort, Patient sample, Stage, and Type to evaluate the data integration results. The UMAP results showed that cells from different cohorts were evenly distributed, indicating successful integration of our data from multiple cohorts. Additionally, in **Figure S3F-J**, we have provided the number of cells in each identified cell group, and the number of samples each cell group appears in (**detailed further in TableS2-Sheet6**). These results indicate these cell populations are not exclusive to individual samples.

As to the stability of the subclusters, as described in response to your major comment #1, we have elucidated that across various resolution parameters, the UPP1 cell group can consistently be identified, with UPP1 being a defining gene. Additionally, in the updated version of our manuscript, we can still distinguish cell groups we've identified before, such as SFTPC+ cells, TK1+ cells, and PTTG1+ cells (**Figure 1D**). They exhibit consistent marker gene expressions, demonstrating that the classification of cell groups is robust.

Regarding the statistical significance of the outcome associations, we have detailed the statistically significant cell groups and their corresponding survival curves in Figure S4. In Figure 1F, for ease of visualization, we only displayed the hazard ratios. However, we have explained in the figure legend that we highlighted the cell groups with statistical significance. Red represents risk, and blue represents $HR < 1$.

We hope these modifications address your concerns adequately.

5. The stage and disease site associations with malignant subclusters may be confounded by sample or disease specificity (i.e. if a subcluster predominantly occurs in an early-stage sample, it will appear associated with early stage disease). It is also important to indicate whether these clusters associate with specific genotypes or other clinical features that may drive prognosis.

Response:

Thank you very much for the comment. As we mentioned in the response to the previous comment, after data integration using Harmony – one of the most efficient methods currently available for single-cell data integration – the cell populations we identified were not restricted to just one sample or one cohort (35, 36). Additionally, even if a subcluster primarily appears in early-stage samples, it should not be simplified as a confounding factor. There is also a possibility that its presence truly represents a crucial characteristic of the early stages of the disease, and the underlying biological significance needs further discussion.

After careful consideration of your comments as well as those from other reviewers and the overall content of our manuscript, we shifted our focus mainly on experimental validation of role of UPP1 in the revised manuscript, rather than heavily relying on bioinformatics analysis to make our point. The purpose of the first part of the revised manuscript was to use single-cell analysis to discover the association of UPP1 with patient prognosis. Subsequently, we primarily conducted further experimental verification on the mechanisms related to tumor progression.

As to the clinical relevance of UPP1, in Figure S5E-5F, the associations between UPP1 and metastasis/recurrence/grade/degree of differentiation were added based on our TMA data and the proteomic source from Xu et al. (37).

6. Related to the previous two points, the association between UPP1 and outcome needs to be analyzed for possible confounding by clinical and genomic features. For example, the fact that UPP1 associates with erlotinib sensitivity suggests it may simply associate with EGFR mutations, which would confound all of the subsequent analyses.

Response:

Thank you very much. As stated in response to your major comment #2, the UPP1 tumor cell serves an independent prognostic factor. Furthermore, we have added the association between UPP1 and clinical features in Figure S5E-5F. Here, to further address your concerns, we also analyzed the association between UPP1 expression and EGFR mutation. However, the results showed no significant correlation (Figure R19), which means our study was not significantly confounded by these factors.

Figure R19. Expression of UPP1 in different mutation groups in scRNA-seq data.

Through the integrative single-cell analysis, we identified prognostic relevance of the UPP1^{high} tumor cell population. Given that UPP1 acts as a marker gene for this group of cells, it is reasonable for us to focus on its role in tumor. In the revised manuscript, we primarily validated our findings through experiments rather than relying heavily on bioinformatics analysis. No matter how extensive the analysis is, direct experimental evidence remains crucial.

From the perspective of bioinformatics analysis for drug screening, the three potential drugs we identified—Bosutinib, Dasatinib, and Erlotinib—are all tyrosine kinase inhibitors (TKIs). This is consistent with our enrichment analysis for the UPP1 tumor cell, which suggested an association between UPP1 and tyrosine kinase-related pathways (Figure S5E). This implies that UPP1's role in tumor progression could be linked to the tyrosine kinase pathway. It is worth noting that UPP1 is indeed associated with the PI3K/AKT/mTOR signaling pathway as proven in our study. Moreover, a recent publication in Nature identified a relationship between UPP1 and the MAPK signaling pathway(38). Both of these signaling pathways are associated with the activation of tyrosine kinase receptors. As such, the gene set associated with UPP1 might be overlapped with the gene sets related to TKI drugs involved in drug sensitivity. This could be one reason we obtained Erlotinib in this analysis. However, it is important to note that the bioinformatics analysis can only serve as a guide providing clues for our next steps in research.

In this section, we adjusted the cell-line level experiments using cells with over-expressed UPP1 and their corresponding control cells (Previously, we used UPP1-over-expressed vs UPP1-sh, but Reviewer #1 believed that the use of an UPP1 overexpression cell line and its corresponding control would be more appropriate). We then also repeated the organoid experiments to verify if tumor cells with high UPP1 baseline expression were more sensitive to the drugs that our analyses

identified. In addition to this, we added animal experiments to further validate our hypothesis (Figure 7 and Figure S17).

As to this drug screening part of our study, our intent was to explore whether LUAD patients with a higher baseline expression of UPP1 might exhibit increased sensitivity to certain existing drugs. This might potentially provide alternative treatment strategies in cases where first-line treatments could not yield satisfactory results, particularly for patients expressing high levels of UPP1. Indeed, this part of the analysis is preliminary and does not simply imply that only the drugs identified in our study could be used. Instead, it might offer a new approach for targeting UPP1 in treatment strategies and potentially create new pathways for future drug development.

7. The text claims, based on figure S3D, that UPP1 associates with disease progression. These gene sets alone cannot support this claim, and furthermore the association is fairly weak. Would need to see the full range of gene sets that associate, and stronger clinical association. In general, the rationale for selecting UPP1 for further study is not totally clear.

Response:

Thank you very much for this comment. After carefully considering your comments and those from other reviewers, we recognize that previous Figure S3D from our initial submission was not central to the primary focus of our study. Therefore, we have decided to remove this section from our paper.

Regarding the rationale for selecting UPP1 for further research, there are several reasons. In Figure 1F, we identified cell populations that correlated with patient prognosis. These populations expressed various marker genes, many of which have been extensively studied in prior literature(39, 40). However, the role of UPP1 in lung cancer remains relatively underexplored. This gap in existing knowledge spurred our interest in investigating UPP1 further. Moreover, one recent publication in Nature has highlighted the significance of UPP1 in pancreatic cancer(38). This underscores the potential importance of UPP1 in various tumor types. Given this emerging evidence, we believe our choice to focus on UPP1 is both justified and understandable.

8. Additionally, the claim that UPP1 associates with T cell inhibition seems poorly supported by the bulk RNA-seq data (3B) – these are very low correlation coefficients even if they are statistically significant.

Response:

Thank you very much. The previous Fig S3B from our initial submission was based on bulk data analysis. In our revised manuscript, we have taken a different approach and primarily focused on validating our core findings, particularly those related to UPP1 and its associated mechanisms. Given this emphasis, certain sections that were not central to this focus have been removed accordingly.

9. The move to look at UPP1 expression within T cell enriched patients is also not supported – why would the expectation be that UPP1 outcome association would be context specific? How is T cell enriched defined? How is UPP1 high vs low defined? How does this differ from the overall outcome association shown in Fig 2? Did UPP1 associate with outcome in T cell low patients?

Response:

Thank you very much. In regard to the previous prognostic analysis involving UPP1 in T-cell enriched patient samples, we have taken into account your advice, as well as the comments from

other reviewers. We acknowledge that this analysis may not have been appropriate, and, as such, we have decided not to include these results in the revised manuscript.

10. The ICI outcome cohorts are small, derived from distinct cancer types, not consistently presented, and do not represent the best published cohorts. Regarding presentation, why is the data from the TIDE cohort differently stratified than Fig 3C (CTL high vs low in UPP1 high vs low rather than looking at UPP1 high vs low in CTL high)? Similarly, why is PFS/OS for the Zhongshan IO cohort not presented, and why isn't response stratified by UPP1 low vs high to be consistent? If non-NSCLC cohorts are going to be used, would use other larger published cohorts, which are enumerated in several meta analyses including Litchfield et al, Cell 2020. The selected datasets are not the best IO validation datasets due to cancer type and small size, and even so not all of the included datasets are used.

Response:

Thank you very much. After careful consideration of your comments, we recognize that our analysis in this section may indeed lack robustness. Relying on immunotherapy data from public cohorts (some of which were not be related to lung cancer) indeed does not provide a compelling case for a direct association between UPP1 and immunotherapy in lung adenocarcinoma. Moreover, our cohort of only 47 patients may not be sufficiently convincing. Regarding the lack of OS and PFS data, the patients in our study were enrolled between 2019-2021. Due to the short follow-up period since their enrollment, we do not yet have survival data. At present, we could not find more supportive lung cancer immunotherapy datasets to support our claims. As for the datasets mentioned in Litchfield et al., Cell 2020, we have checked the publication. Many of the datasets are stored in the EGA database and cannot be downloaded. We have submitted requests for access, but to date, have received no response.

Considering the issues with our data in this section, in the revised manuscript, we no longer emphasize the relevance of UPP1 to immunotherapy. Instead, we focused more on the mechanistic relationship between UPP1 and tumor progression and the tumor microenvironment.

11. The claim that UPP1 is associated with immunosuppression and IO resistance is not conclusively shown by the data in 3A-F. The previous figure demonstrated that UPP1 was potentially prognostic; there is little to suggest that the data is predictive to ICI over and above its prognostic role (related to the previous reviewer point re IO outcome cohorts). If the contention is that UPP1 associates with a more immunosuppressive immune microenvironment, why wasn't the single cell data used to explore this claim?

Response:

Thank you very much. Concerning the relationship between UPP1 and the immune-suppressive microenvironment, we have made significant revisions in the updated manuscript, relying mainly on the scRNA-seq data. The primary amendments are as follows:

1. The relationship between UPP1^{high} tumor cells and the Tumor Microenvironment (TME) cell populations was re-analyzed using single-cell RNA-sequencing (scRNA-seq) data (Figure S7).
2. A co-occurrence network was constructed, revealing that UPP1^{high} tumor cells were closely related to four groups of immunosuppressive and tumor progression-related cells within the TME (Figure S8).

3. Multiplex immunofluorescence staining was conducted for spatial relationship analysis and Visiopharm software was used for unbiased analysis. The analysis revealed that UPP1^{high} tumor cells were primarily located at the invasive margins of the tumor and had closer spatial proximity with other cell groups (Figure 2 and Figure S9).
4. A cell-cell communication network analysis revealed broad engagement of ligand-receptor and cytokine regulations in the interactions between UPP1^{high} tumor cells and TME cells.
5. Furthermore, our analysis highlighted TGF- β , an important molecule associated with immune suppression, played a key role in UPP1-mediated immune suppression (Figure 3).
6. In vivo experiments and CyTOF analysis were performed to substantiate the relationship between UPP1 and immune suppression (Figure 5).

We hope these additional experiments could address the concerns raised and to provide a more robust foundation for the association between UPP1 and the immune suppressive microenvironment.

12. PD-L1 expression can be a positive predictive marker for ICI response in NSCLC. If UPP1 associates with PD-L1 expression, it could have positive or negative implications for ICI response. The mechanism behind UPP1 association with immune response is not explicated.

Response:

Thank you very much for this comment. Regarding the regulatory relationship between UPP1 and PD-L1, we elucidated specifically in the revised article. As shown in Figure 4, after upregulating UPP1, we examined the changes in PI3K/AKT/mTOR using Western blotting. Subsequently, upon suppression of UPP1, we treated the samples with SC79 (AKT activator) and again detected the changes in AKT/mTOR/PD-L1 via Western blotting. These results suggested that UPP1 could influence PD-L1 expression through the PI3K/AKT/mTOR/PD-L1 axis.

On the basis of that, we found the use of anti-PD-L1 antibodies could partially restore the cytotoxic function of CD8 T cells in the co-culture of CD8 T with UPP1 up-regulated tumor cells (Figure 4I and Figure S11C-G), suggesting that UPP1 could influence the cytotoxic ability of CD8 T cells via PD-L1.

Further, by establishing a subcutaneous tumor injection model, we observed that, compared to the shNC-UPP1 group, treatment with PD-L1 monotherapy yielded better therapeutic effects in shUPP1 group, suggesting that UPP1 knockdown could enhance the sensitivity to PD-L1 checkpoint monotherapy (Figure 6).

13. In Fig 3H-I and S5F, could UPP1 just be a tumor cell marker? Do UPP1 negative cells have a different T cell/PD-L1 configuration? Can this be demonstrated across more than a handful of samples?

Response:

Thank you very much for this constructive comment. UPP1 is not a tumor cell marker, so to refine our approach, we re-performed multiplex immunofluorescence on the samples from 15 patients (Figure 2, Figure S8E and Figure S9). We added pan-cytokeratin (panCK) to define the tumor cells and then determined the high or low expression of UPP1 based on its expression on panCK⁺ tumor cells. For phenotypic identification and quantification, we used the Visiopharm software (<https://visiopharm.com>), a commercial provider of quantitative digital pathology solutions, for data analysis. The cells were identified using the incorporated AI-based recognition algorithm. We then utilized the phenotype module provided by Visiopharm software to distinguish UPP1^{high} and

UPP1^{low} tumor cells. We also conducted spatial distance analyses to verify the spatial distribution of UPP1^{high} tumor cells and their relationships with other cells in the tumor microenvironment (Line 798-814). We found that UPP1^{high} tumor cells were primarily located at the invasive margins of the tumor and had closer spatial proximity with these cell groups (Figure 2 and Figure S9). These results provided substantial evidence supporting the association between UPP1 and the immunosuppressive microenvironment. We believe that these modifications present a more accurate and comprehensive view of the role of UPP1 in the tumor microenvironment.

14. The UPP1 knockdown and overexpression does not appear very effective in multiple cell lines.

Response:

Thank you deeply for this comment. For the establishment of cell lines, we have refined our protocols and carried out additional rounds of screening with puromycin. In the revised manuscript, we confirmed the efficacy of overexpression and knockdown using both RT-qPCR and Western blotting (Figure S10A, Figure S11C, Figure S12A, Figure S16B, and Figure 4).

15. Much of the T cell specific analyses identifies features that have previously been described, and are not convincingly associated with UPP1 status – most of this rests on Figs 5G-H which is very difficult to interpret. The co-culture data are interesting but there is still a marginal effect on the proportion of CXCL13^{high} CD8 T cell expression.

Response:

Thank you very much. After carefully considering your comments as well as those from other reviewers, we have conducted a comprehensive revision of our data analysis process. In the revised manuscript, the primary focus has been shifted to comprehensively discuss and validate the role and mechanisms of UPP1. Given this change in focus, the previously discussed CXCL13 T cells, which was less conclusive, was no longer central to our study. As such, we have decided not to include this content in the modified version of our manuscript.

16. The analysis of fibroblasts is interesting but adds a lot of data to an already dense manuscript that would benefit from more validation of the initial, central claims.

Response:

We sincerely appreciate this invaluable comment pointing out the problems in our initial manuscript, and we acknowledge that our previous version did not provide a clear and coherent story on this part. To address this, we have re-analyzed the relationship between UPP1 and the tumor microenvironment and found a significant correlation between UPP1^{high} tumor cells and MMP11+ CAFs (Figure S7). To further substantiate these findings, we conducted multiplex immunofluorescence staining on patient tissues, which demonstrated a closer spatial distance between UPP1^{high} tumor cells and MMP11+ fibroblasts as compared to UPP1^{low} tumor cells (Figure 2 and Figure S9). Following this, we also re-analyzed the single-cell data from the co-culture experiment and revealed that the upregulation of UPP1 in tumor cells can promote the transition of fibroblasts towards a CAF phenotype (Figure 3J-P). In the subsequent in vivo analysis, we also validated that the upregulation of UPP1 can increase the infiltration of CAFs (Figure 5L-N). In the revised manuscript, we have integrated the fibroblast-related findings throughout the paper, rather than presenting them in a separate figure. We hope these modifications could more effectively illustrate the findings and address your concerns.

Minor comments:

17. - The paper is well written but would benefit from further English language editing for ease of reading (e.g. wildly used instead widely)

Response:

Thank you very much for pointing out this problem. We acknowledge that there were some language errors in our initial manuscript that might have hindered readability. We have therefore consulted with native English speakers to refine the language in the revised manuscript. We appreciate your patience and understanding.

18. - There are multiple cohorts used at various points. It would be helpful to define upfront in the results the different cohorts, and to have them used consistently throughout, to avoid the appearance of data cherry-picking.

Response:

Thank you very much for your suggestion. In the revised manuscript, we added **Table S1** and **Table S5** to introduce the cohorts we used in this article and their specific uses. Additionally, we have simplified the number of cohorts we used to ensure the consistency of the article. For all analyses that need to be performed in multiple cohorts, we have presented the data for each cohort to avoid the appearance of cherry-picking.

19. - Fig 3A derives from which cohort? Why only one?

Response:

Thank you very much. In the initial analysis, we performed analysis in all cohorts and the results were meaningful in all of them, but due to space constraints, we only showed one. In the revised manuscript, for all analyses that need to be performed in multiple cohorts, we have presented the data for each cohort.

20. - What does 'mutation' status refer to in the early figures? Better to be specific?

Response:

Thank you very much for this question. Some of the scRNA-seq datasets we included in this study only provided information on whether there was a mutation or not, without specifying the exact mutation (such as EGFR mutation, TP53 mutation, etc.). To maintain consistency in data presentation, we previously categorized all patients with any type of mutation, such as EGFR or TP53 mutations, under the 'mutated' label. However, to prevent any potential misunderstandings, we have made adjustments in the revised manuscript to present specific types of mutations (**Figure S1A and S1B**). And the aim of this part of data visualization was to present the characteristics of the patient population in the study.

21. - Related to the comments above, would focus first on exploring UPP1 and the TME in the single cell datasets, then moving to the therapeutic implications. The initial rich, datasource is not fully taken advantage of.

Response:

Thank you very much for this recommendation. Following your suggestion, in the revised manuscript, we have restructured our manuscript to more comprehensively focus on UPP1 and its

relationship with TME using single cell datasets. We tried our best to address each of the points raised. We have also added more experiments to validate our claims. We hope these changes could adequately address your concerns and further strengthen our study.

We prioritized our single-cell data analysis to lay a robust foundation for the subsequent sections. By re-analyzing this data using more refined parameters to prevent over-clustering, we were able to provide a clearer understanding of the role of UPP1 within the TME. We then used patient samples to validate the link between UPP1 expression and patient prognosis. And we performed additional experiments to validate the role of UPP1 in tumor proliferation.

In the second part of our manuscript, the relationship between UPP1^{high} tumor cells and the TME was re-analyzed using scRNA-seq data. Through cell-cell communication analyses and multiplex immunofluorescence validation on patient samples, the tight connection between UPP1 and immunosuppressive TME was revealed.

We further extended our exploration by investigating how UPP1 expression variations impact the functions of tumor cells and their interactions with surrounding cells in the TME. Our analysis brought to light the central role of TGF- β in the UPP1-driven immune suppression mechanism. Moreover, we provided evidence for UPP1's potential modulation of PD-L1 expression via the PI3K/AKT/mTOR pathway.

The inclusion of in vivo experiments and CyTOF analyses was aimed at reinforcing our findings regarding UPP1's immunosuppressive role.

We then investigated the feasibility of targeting UPP1 to enhance the efficacy of immunotherapies and the potential of UPP1 as a new target for tumor treatment.

We hope that this revised structure could add more depth to our research narrative. Once again, your comments have been instrumental in shaping our manuscript, and we are really grateful for the insights you have provided. If there are any more areas of concern, we would greatly appreciate your guidance and the chance to refine our work further.

References

1. Xue R, Zhang Q, Cao Q, Kong R, Xiang X, Liu H, et al. Liver tumour immune microenvironment subtypes and neutrophil heterogeneity. *Nature*. 2022;612(7938):141-7.
2. Kim N, Kim HK, Lee K, Hong Y, Cho JH, Choi JW, et al. Single-cell RNA sequencing demonstrates the molecular and cellular reprogramming of metastatic lung adenocarcinoma. *Nat Commun*. 2020;11(1):2285.
3. Lin W, Noel P, Borazanci EH, Lee J, Amini A, Han IW, et al. Single-cell transcriptome analysis of tumor and stromal compartments of pancreatic ductal adenocarcinoma primary tumors and metastatic lesions. *Genome Med*. 2020;12(1):80.
4. He D, Wang D, Lu P, Yang N, Xue Z, Zhu X, et al. Single-cell RNA sequencing reveals heterogeneous tumor and immune cell populations in early-stage lung adenocarcinomas harboring EGFR mutations. *Oncogene*. 2021;40(2):355-68.
5. Laughney AM, Hu J, Campbell NR, Bakhoun SF, Setty M, Lavalley VP, et al. Regenerative lineages and immune-mediated pruning in lung cancer metastasis. *Nat Med*. 2020;26(2):259-69.
6. Wu F, Fan J, He Y, Xiong A, Yu J, Li Y, et al. Single-cell profiling of tumor heterogeneity and the microenvironment in advanced non-small cell lung cancer. *Nat*

Commun. 2021;12(1):2540.

7. Xing X, Yang F, Huang Q, Guo H, Li J, Qiu M, et al. Decoding the multicellular ecosystem of lung adenocarcinoma manifested as pulmonary subsolid nodules by single-cell RNA sequencing. *Sci Adv.* 2021;7(5).
8. Luo H, Xia X, Huang L-B, An H, Cao M, Kim GD, et al. Pan-cancer single-cell analysis reveals the heterogeneity and plasticity of cancer-associated fibroblasts in the tumor microenvironment. *Nature Communications.* 2022;13(1):6619.
9. Zheng L, Qin S, Si W, Wang A, Xing B, Gao R, et al. Pan-cancer single-cell landscape of tumor-infiltrating T cells. *Science.* 2021;374(6574):abe6474.
10. Cheng S, Li Z, Gao R, Xing B, Gao Y, Yang Y, et al. A pan-cancer single-cell transcriptional atlas of tumor infiltrating myeloid cells. *Cell.* 2021;184(3):792-809.e23.
11. Eisenhauer EA, Therasse P, Bogaerts J, Schwartz LH, Sargent D, Ford R, et al. New response evaluation criteria in solid tumours: revised RECIST guideline (version 1.1). *Eur J Cancer.* 2009;45(2):228-47.
12. Li Y, Xu F, Chen F, Chen Y, Ge D, Zhang S, et al. Transcriptomics based multi-dimensional characterization and drug screen in esophageal squamous cell carcinoma. *EBioMedicine.* 2021;70:103510.
13. Chen Y, McAndrews KM, Kalluri R. Clinical and therapeutic relevance of cancer-associated fibroblasts. *Nat Rev Clin Oncol.* 2021;18(12):792-804.
14. Peruzzi D, Mori F, Conforti A, Lazzaro D, De Rinaldis E, Ciliberto G, et al. MMP11: a novel target antigen for cancer immunotherapy. *Clin Cancer Res.* 2009;15(12):4104-13.
15. Öhlund D, Handly-Santana A, Biffi G, Elyada E, Almeida AS, Ponz-Sarvise M, et al. Distinct populations of inflammatory fibroblasts and myofibroblasts in pancreatic cancer. *J Exp Med.* 2017;214(3):579-96.
16. Nurmik M, Ullmann P, Rodriguez F, Haan S, Letellier E. In search of definitions: Cancer-associated fibroblasts and their markers. *Int J Cancer.* 2020;146(4):895-905.
17. Geeleher P, Cox N, Huang RS. pRRophetic: an R package for prediction of clinical chemotherapeutic response from tumor gene expression levels. *PLoS One.* 2014;9(9):e107468.
18. Pushpakom S, Iorio F, Eyers PA, Escott KJ, Hopper S, Wells A, et al. Drug repurposing: progress, challenges and recommendations. *Nat Rev Drug Discov.* 2019;18(1):41-58.
19. Yang C, Chen J, Li Y, Huang X, Liu Z, Wang J, et al. Exploring subclass-specific therapeutic agents for hepatocellular carcinoma by informatics-guided drug screen. *Brief Bioinform.* 2021;22(4).
20. Hu JF, Song X, Zhong K, Zhao XK, Zhou FY, Xu RH, et al. Increases prognostic value of clinical-pathological nomogram in patients with esophageal squamous cell carcinoma. *Front Oncol.* 2023;13:997776.
21. Lu T, Zhang Z, Zhang J, Pan X, Zhu X, Wang X, et al. CD73 in small extracellular vesicles derived from HNSCC defines tumour-associated immunosuppression mediated by macrophages in the microenvironment. *J Extracell Vesicles.* 2022;11(5):e12218.
22. Li Y, Gu J, Xu F, Zhu Q, Chen Y, Ge D, et al. Molecular characterization,

biological function, tumor microenvironment association and clinical significance of m6A regulators in lung adenocarcinoma. *Brief Bioinform.* 2021;22(4).

23. Wang S, Su W, Zhong C, Yang T, Chen W, Chen G, et al. An Eight-CircRNA Assessment Model for Predicting Biochemical Recurrence in Prostate Cancer. *Front Cell Dev Biol.* 2020;8:599494.

24. Zeng D, Li M, Zhou R, Zhang J, Sun H, Shi M, et al. Tumor Microenvironment Characterization in Gastric Cancer Identifies Prognostic and Immunotherapeutically Relevant Gene Signatures. *Cancer Immunol Res.* 2019;7(5):737-50.

25. Zhang Y, Chen H, Mo H, Hu X, Gao R, Zhao Y, et al. Single-cell analyses reveal key immune cell subsets associated with response to PD-L1 blockade in triple-negative breast cancer. *Cancer Cell.* 2021;39(12):1578-93.e8.

26. Li X, Sun Z, Peng G, Xiao Y, Guo J, Wu B, et al. Single-cell RNA sequencing reveals a pro-invasive cancer-associated fibroblast subgroup associated with poor clinical outcomes in patients with gastric cancer. *Theranostics.* 2022;12(2):620-38.

27. Chen Z, Zhou L, Liu L, Hou Y, Xiong M, Yang Y, et al. Single-cell RNA sequencing highlights the role of inflammatory cancer-associated fibroblasts in bladder urothelial carcinoma. *Nature Communications.* 2020;11(1):5077.

28. Xie X, Shi Q, Wu P, Zhang X, Kambara H, Su J, et al. Single-cell transcriptome profiling reveals neutrophil heterogeneity in homeostasis and infection. *Nat Immunol.* 2020;21(9):1119-33.

29. Bagaev A, Kotlov N, Nomie K, Svekolkina V, Gafurov A, Isaeva O, et al. Conserved pan-cancer microenvironment subtypes predict response to immunotherapy. *Cancer Cell.* 2021;39(6):845-65.e7.

30. Clarke ZA, Andrews TS, Atif J, Pouyababar D, Innes BT, MacParland SA, et al. Tutorial: guidelines for annotating single-cell transcriptomic maps using automated and manual methods. *Nature Protocols.* 2021;16(6):2749-64.

31. Han Y, Liu D, Li L. PD-1/PD-L1 pathway: current researches in cancer. *Am J Cancer Res.* 2020;10(3):727-42.

32. Aibar S, González-Blas CB, Moerman T, Huynh-Thu VA, Imrichova H, Hulselmans G, et al. SCENIC: single-cell regulatory network inference and clustering. *Nat Methods.* 2017;14(11):1083-6.

33. Wang X, Wang Z, Huang R, Lu Z, Chen X, Huang D. UPP1 Promotes Lung Adenocarcinoma Progression through Epigenetic Regulation of Glycolysis. *Aging Dis.* 2022;13(5):1488-503.

34. Yoshihara K, Shahmoradgoli M, Martínez E, Vegesna R, Kim H, Torres-Garcia W, et al. Inferring tumour purity and stromal and immune cell admixture from expression data. *Nat Commun.* 2013;4:2612.

35. Korsunsky I, Millard N, Fan J, Slowikowski K, Zhang F, Wei K, et al. Fast, sensitive and accurate integration of single-cell data with Harmony. *Nat Methods.* 2019;16(12):1289-96.

36. Tran HTN, Ang KS, Chevrier M, Zhang X, Lee NYS, Goh M, et al. A benchmark of batch-effect correction methods for single-cell RNA sequencing data. *Genome Biol.* 2020;21(1):12.

37. Xu JY, Zhang C, Wang X, Zhai L, Ma Y, Mao Y, et al. Integrative Proteomic

- Characterization of Human Lung Adenocarcinoma. *Cell*. 2020;182(1):245-61. e17.
38. Nwosu ZC, Ward MH, Sajjakulnukit P, Poudel P, Ragulan C, Kasperek S, et al. Uridine-derived ribose fuels glucose-restricted pancreatic cancer. *Nature*. 2023;618(7963):151-8.
39. Wei YT, Luo YZ, Feng ZQ, Huang QX, Mo AS, Mo SX. TK1 overexpression is associated with the poor outcomes of lung cancer patients: a systematic review and meta-analysis. *Biomark Med*. 2018;12(4):403-13.
40. Li H, Yin C, Zhang B, Sun Y, Shi L, Liu N, et al. PTTG1 promotes migration and invasion of human non-small cell lung cancer cells and is modulated by miR-186. *Carcinogenesis*. 2013;34(9):2145-55.

Reviewers' Comments:

Reviewer #1:

Remarks to the Author:

The authors have made significant changes to their approach in this revision and prepared a more cohesive story. The concerns raised by me in the initial version have been satisfactorily addressed. Overall, I believe the authors have shown a good body of work and some interesting results and their claims are largely restricted to the evidence provided. I believe the work presented here fills a literary gap and has the potential to lead to further interesting research in this direction.

Some of my observations are presented below:

1. Based on figure S4A, largest differences in survival between high and low groups for each correlative cluster seem to be in cohort 2. Can you discuss possible reason for this?
2. Line 201- I appreciate the authors performing spatial analysis to confirm that the ligand-receptor interaction analysis aligns with actual spatial distribution. What is the level of infiltration of lymphocytes in these tumors? If they are highly infiltrated, what does the spatial distribution and possible interaction of the TILs with the UPFlow cells inside the tumor reveal?
3. Line 249- could these co-cultured fibroblasts be clustered using the scRNAseq data? Do you see any distinct clusters with potentially highly differentially expressed genes serving to identify the clusters? It would be interesting to see this in addition to showing differences in expression of FAP and MMP11

Reviewer #2:

Remarks to the Author:

The authors have performed an impressive job in turning this paper around. The revised manuscript has addressed all but a few of my initial concerns and has improved greatly. Only a few minor comments remain, which I am confident the authors will be able to address.

- In Figure S1A, the mutation variable is still unclear and does not add to the story. I would suggest removing this variable.
- Line 144: typo "Inflammatory_Response"
- Figure S6: what is the relevance of this data following the conclusion in line 162? It feels disconnected from the storyline.
- Figure S11F: A control of OENC + T cells + aPDL1 is missing. Without this control, it is impossible to evaluate might be a similar bottleneck in OENC cells compared to UPP1-OE cells.
- Figure 6A-D) Since UPP1 knockdown results in loss of PD-L1 expression (Figure 4E), it is surprising that UPP1-sh tumors show sensitivity to aPDL1 in vivo (but not their UPP1-high counterparts). It is not strictly essential that the authors solve this discrepancy, but it would be important to point this out in the discussion. One potential explanation is that in UPP1-high cells alternative checkpoints (Figure 4A) contribute to immune evasion and that therefore blocking PD-L1 is not sufficient.
- Figure 7F: axis labeled as percentage (0-100) but should be proportion (0-1)?
- Figure 7G: it should be confirmed that these are tumor organoids (e.g. based on genomic alterations or IHC) given the potential for outgrowth of normal airway organoids (<https://pubmed.ncbi.nlm.nih.gov/32375033/>)

Reviewer #3:

Remarks to the Author:

The revised manuscript is much improved. The main claims are more clearly supported, the methods are clearer, and it is more focused. I commend the authors on the considerable work involved.

Response to Reviewers

Reviewer #1 (Remarks to the Author):

The authors have made significant changes to their approach in this revision and prepared a more cohesive story. The concerns raised by me in the initial version have been satisfactorily addressed. Overall, I believe the authors have shown a good body of work and some interesting results and their claims are largely restricted to the evidence provided. I believe the work presented here fills a literary gap and has the potential to lead to further interesting research in this direction. Some of my observations are presented below:

Response:

We would like to express our sincere gratitude for your valuable and constructive comments on our manuscript. The insights and suggestions have been instrumental in shaping our research and improving the quality of our work. We are pleased to hear that the revisions we made in response to your previous comments to be satisfactory.

1. Based on figure S4A, largest differences in survival between high and low groups for each correlative cluster seem to be in cohort 2. Can you discuss possible reason for this?

Response:

Thank you very much for this comment. Our primary goal in this analysis was to conduct independent survival prognosis studies across different cohorts, aiming to identify cell clusters consistently linked to patient prognosis across these groups. While we did not initially set out to compare prognostic differences among cohorts.

To explore potentially possible variations in survival analysis among different cohorts, we here compared the risk coefficients calculated within each cohort. This comparison revealed that the risk coefficients across the three cohorts did actually not exhibit significant discrepancies.

From the observations of Figure S4, we think that the possible reasons for the variations seen across different cohorts could be attributed to differences in clinical characteristics and follow-up durations among patients. To explore this possibility, we reviewed the clinical information of these cohorts. We noticed that there were minimal differences in age, gender, and survival status composition among the patients in the three cohorts. However, it's noteworthy that Cohort 2 had a relatively shorter average follow-up duration, which might be one of reasons explaining the disparities seen in the survival analysis.

	Female	Male			Alive	Dead
Cohort1	249	270		Cohort1	335	224
Cohort2	222	176		Cohort2	285	113
Cohort3	276	237		Cohort3	326	187
Age (year)	Min	Median	Mean	Max		
Cohort1	30	62.63	63	85.91		
Cohort2	38	69.36	70	89		
Cohort3	33	66	65.28	88		
Overall-survival follow-up (month)	Min	Median	Mean	Max		
Cohort1	0.67	58.92	59.5	221		
Cohort2	0.1	27.47	26.4	69.23		
Cohort3	0	21.77	30.1	241.6		

It is also notable that epidemiological and genetic differences among the cohorts might play a role, but the limited information of the provided original data restricts us from drawing definitive conclusions in this regard. However, our main focus here is to identify cell clusters associated with patient prognosis across different cohorts.

2. Line 201- I appreciate the authors performing spatial analysis to confirm that the ligand-receptor interaction analysis aligns with actual spatial distribution. What is the level of infiltration of lymphocytes in these tumors? If they are highly infiltrated, what does the spatial distribution and possible interaction of the TILs with the UPP1low cells inside the tumor reveal?

Response:

Thank you very much for this comment, this is a very good question. When we analyzed the spatial distances, we conducted an overall analysis across all ROIs (regions of interest). Overall, tumor cells with high expression of UPP1 are closer in spatial distance to lymphocytes compared to tumor cells with low expression of UPP1.

However, understanding the level of infiltration and spatial distribution of immune cells is also very important. Based on your question, we conducted a re-review analysis of the images.

Overall, the distribution of the level of infiltration of CD4 T cells in these 15 patients is as follows:

For CD4 T cell infiltration, the minimum infiltration level is 0.2728, the median is 0.3644, and the maximum value is 0.4922.

and the distribution of the level of infiltration of CD8 T cells in these 15 patients is as follows:

The proportion of CD8 T cell infiltration, the minimum level is 0.3595, the median stands at 0.4296, and the highest value observed is 0.5130.

Here, we classified tumor samples into high infiltration and low infiltration based on the median level of infiltration and combined with a review of the images. We then conducted separate comparisons for each group. The results revealed that regardless of whether the patients had high or low infiltration, UPP1high tumor cells are always closer in spatial distance to lymphocytes compared to UPP1low tumor cells.

From a spatial distribution perspective, we reviewed the images, in tumors with a higher infiltration ratio of CD4 T cells, we can still observe that CD4 T cells tend to be predominantly located in areas rich in UPP1high tumor cells, being closer in overall spatial distance (yellow frame). In contrast, for UPP1low tumor cells, the overall spatial distance is relatively farther. Within the tumor parenchyma, in areas primarily populated by UPP1low tumor cells, we observed that CD4 T cells are surrounded by scattered UPP1high tumor cells, and they tend to be closer to UPP1high tumor cells (yellow arrow). This phenomenon suggests that the interaction between UPP1high tumor cells and CD4 T cells is more active, and their association with the tumor microenvironment is more closely linked.

Representative images of CD4 T cells

For CD8 T cells, a similar trend is observed. A large number of CD8 T cells are primarily concentrated around UPP1^{high} tumor cells (purple frame). In areas predominantly occupied by UPP1^{low} tumor cells, scattered UPP1^{high} tumor cells also tend to be closer to CD8 T cells (purple arrow). Nevertheless, in the images, it is also observed that some UPP1^{low} tumor cells are in proximity to CD8 T cells, but this correlation is relatively less frequent.

Representative images of CD8 T cells

Overall, in tumors with higher CD4 and CD8 T cell infiltration, these immune cells are predominantly located in regions rich in UPP1^{high} tumor cells. In areas where UPP1^{low} tumor cells are more common, there are still scattered UPP1^{high} tumor cells, and both CD4 and CD8 T cells tend to be closer to these UPP1^{high} cells. The observed spatial closeness of lymphocytes (both CD4 and CD8) to UPP1^{high} tumor cells indicates an active interaction. This suggests that UPP1^{high} cells could play a crucial role in modulating the tumor microenvironment.

3. Line 249- could these co-cultured fibroblasts be clustered using the scRNAseq data? Do you see any distinct clusters with potentially highly differentially expressed genes serving to identify the clusters? It would be interesting to see this in addition to showing differences in expression of FAP and MMP11.

Response:

Thank you sincerely for raising this question. Initially, we had considered this aspect during our analysis. However, upon careful consideration of the comments from other reviewers, we realized it would be better to maintain

consistency with the cell types identified in our single-cell analysis and patient sample validation. Therefore, in this step of our analysis, we used a gene set scoring strategy to analyze MMP11-like fibroblasts.

In the initial phase of our analysis, we did indeed use a clustering approach for fibroblasts. Through this process, we identified 11 distinct cell populations, among which cell populations 2, 5, 6, and 9 were primarily distributed in groups with high UPP1 expression. Specifically, cell population 2 predominantly expressed NEAT1, population 5 mainly expressed BAG3, population 6 primarily expressed IFI6/27, and population 9 mainly expressed IGFBP3. These differential genes previously found to be associated with CAFs(1-3).

Furthermore, functional enrichment analysis of these cell populations revealed that population 5 was mainly associated with tumor metastasis, population 6 with tumor inflammation, and populations 2 and 9 with the TGF-beta pathway and collagen formation, all indicative of their tumor-related fibroblastic characteristics. This suggests that UPP1 tumor cells promoted the transformation of fibroblasts into CAF-like cells.

We think it would be better to maintain consistency with the cell populations analyzed previously, so we employed the gene set approach described above for this analysis. We hope to further explore the characteristics of these cell populations in further work. We deeply appreciate your understanding and value your insights on this matter.

Reviewer #2 (Remarks to the Author):

The authors have performed an impressive job in turning this paper around. The revised manuscript has addressed all but a few of my initial concerns and has improved greatly. Only a few minor comments remain, which I am confident the authors will be able to address.

Response:

Thank you very much for your encouraging comment on our revised manuscript. We are delighted to know that our revisions have addressed most of your initial concerns and that you find the manuscript significantly improved. Your constructive guidance and support throughout this revision process have been immensely valuable to us, and we are grateful for your positive and detailed insights. Thank you once again for your time and thoughtful review.

1. -In Figure S1A, the mutation variable is still unclear and does not add to the story. I would suggest removing this variable.

Response:

Thank you very much for this comment. We agree that this variable does not significantly contribute to the overall narrative of our research. As such, we have decided to remove the mutation variable from Figure S1A to enhance the clarity and focus of the figure in relation to our study's main story. We appreciate your guidance in helping us refine our presentation for better clarity and relevance.

2. -Line 144: typo “Inflammatory_Response”

Response:

Thank you very much for pointing out the typo error. We have corrected this error in the manuscript. We appreciate your attention to detail and thank you for helping us improve the quality of our paper.

3. -Figure S6: what is the relevance of this data following the conclusion in line 162? It feels disconnected from the storyline.

Response:

Thank you for this comment regarding Figure S6 and its connection to the main narrative of our manuscript. In our initial draft, we explored the hypothesis that since UPP1 is associated with patient prognosis, inhibiting this gene might suppress tumor proliferation and invasion. Originally, we employed a single siRNA approach. However, based on your valuable suggestions, we revised our approach to include two independent siRNAs for a more robust validation.

Upon re-evaluation, we agree that this section appears somewhat disconnected from the overall storyline of the manuscript. We have now removed this part from the manuscript. We hope this adjustment would enhance the clarity of our findings. We are grateful for the opportunity to improve our work with your guidance.

4. -Figure S11F: A control of OENC + T cells + aPDL1 is missing. Without this control, it is impossible to evaluate might be a similar bottleneck in OENC cells compared to UPP1-OE cells.

Response:

Thank you very much for your insightful comment. In response to your suggestion, we conducted additional experiments to include the control of OENC+T cells+aPDL1. Our findings revealed that there is no significant statistical difference between the OENC+T cells+aPDL1 group and the OENC+T cells group (Figure S10e-g). This outcome suggests that PDL1 blockade does not substantially improve the T cells' ability to eliminate OENC cells. On the other hand, for OE cells, PDL1 blockade offers partial relief, implying that UPP1 upregulation contributed to some extent of immune suppression via the PD-L1 pathway.

5. -Figure 6A-D) Since UPP1 knockdown results in loss of PD-L1 expression (Figure 4E), it is surprising that UPP1-sh tumors show sensitivity to aPDL1 in vivo (but not their UPP1-high counterparts). It is not strictly essential that the authors solve this discrepancy, but it would be important to point this out in the discussion. One potential explanation is that in UPP1-high cells alternative checkpoints (Figure 4A) contribute to immune evasion and that therefore blocking PD-L1 is not sufficient.

Response:

Thank you very much for raising this insightful question, we have contemplated this aspect of our study. Our findings suggest that UPP1 plays a significant role in shaping the immunosuppressive microenvironment. In our mouse models, the suppression of UPP1 led to an increased infiltration of CD8 T cells, whereas the absence of UPP1 suppression resulted in comparatively lower CD8 T cell infiltration within tumor tissues. Previous research has indicated that a reduction in CD8 T cell infiltration is one of the reasons for the poor efficacy of PD-L1 monoclonal antibodies(4, 5). This might explain why inhibiting UPP1 increases the sensitivity to immunotherapy, namely, after the suppression of UPP1 expression, there is a relative increase in CD8 T cell infiltration, thereby exhibiting a phenomenon of increased sensitivity to immunotherapy.

Additionally, it was also found that tumors with higher UPP1 expression showed a significant increase in PD-L1 expression in macrophages. Previous research has indicated that PD-L1 expression in other immune cells such as macrophages is also crucial for the effectiveness of immunotherapy(6, 7). The overall increase in PD-L1 expression due to elevated UPP1 may also be a contributing factor affecting the efficacy of PD-L1-targeted treatments.

We completely agree with your viewpoint on the potential involvement of additional immune checkpoints. While our study primarily focused on the association between UPP1 and PD-L1, it is possible that UPP1 expression may also be related to the activation and upregulation of several immune checkpoints, such as CD276, CD70, CD47, and LGALS9. The upregulation of these checkpoints can affect the efficacy of PD-1/PD-L1 immunotherapy. For instance, previous studies have shown that combined application of CD47 with PD1/PDL1 blockade can result in stronger anti-tumor effects(8). This suggests that single checkpoint pathway inhibition might not be sufficient. These insights indicate that UPP1 may also be implicated in additional critical immune escape mechanisms. Hence, our study only partially explores the mechanisms by which UPP1 participates in immune suppression. Further, more in-depth research is needed to systematically reveal the core and complete mechanisms of UPP1 in immune escape. This part of discussion is added in the manuscript, line 485-507.

6. -Figure 7F: axis labeled as percentage (0-100) but should be proportion (0-1)?

Response:

Thank you very much for highlighting the labeling issue in Figure 7F. The axis should indeed be labeled as a proportion (0-1) rather than a percentage (0-100). We have corrected this in the figure.

7. -Figure 7G: it should be confirmed that these are tumor organoids (e.g. based on genomic alterations or IHC) given the potential for outgrowth of normal airway organoids (<https://pubmed.ncbi.nlm.nih.gov/32375033/>)

Response:

Thank you very much for this comment. We have carefully read the literature you provided and, in combination with previous literature on organoids(9), we have included three markers in our study: TTF-1, CK5, and P63. According to the literature, lung adenocarcinoma organoids are usually TTF1 positive, CK5 negative, and P63 negative, or P63 can be positive but without polarization. In our revised manuscript, we have conducted immunofluorescence and immunohistochemical staining. Firstly, the organoids we constructed are TTF1 positive, confirming their adenocarcinoma origin, and secondly, they are CK5 negative. The immunohistochemical results for P63 revealed that samples O2, O3, O4, and O6 were all P63 negative (Supplementary Figure 17). O1 and O5 exhibited low P63 expression, but did not demonstrate any apparent polarized characteristics. Instead, they displayed a feature of heterogeneous expression. This confirms that our constructed organoids are derived from tumors.

Reviewer #3 (Remarks to the Author):

The revised manuscript is much improved. The main claims are more clearly supported, the methods are clearer, and it is more focused. I commend the authors on the considerable work involved.

Response:

Thank you very much for the kind words about our revised manuscript. We are really grateful for your helpful comments, which guided us to make it clearer and more focused. Your advice has been truly valuable to us. Thanks again for the time you spent reviewing our work.

References

1. Fan JT, Zhou ZY, Luo YL, Luo Q, Chen SB, Zhao JC, et al. Exosomal lncRNA NEAT1 from cancer-associated fibroblasts facilitates endometrial cancer progression via miR-26a/b-5p-mediated STAT3/YKL-40 signaling pathway. *Neoplasia*. 2021;23(7):692-703.
2. De Marco M, Gauttier V, Pengam S, Mary C, Ranieri B, Basile A, et al. Concerted BAG3 and SIRP α blockade impairs pancreatic tumor growth. *Cell Death Discovery*. 2022;8(1):94.
3. Thomas D, Radhakrishnan P. Role of Tumor and Stroma-Derived IGF/IGFBPs in Pancreatic Cancer. *Cancers (Basel)*. 2020;12(5).
4. Soares KC, Rucki AA, Wu AA, Olino K, Xiao Q, Chai Y, et al. PD-1/PD-L1 blockade together with vaccine therapy facilitates effector T-cell infiltration into pancreatic tumors. *J Immunother*. 2015;38(1):1-11.
5. Kumar S, Singh SK, Rana B, Rana A. Tumor-infiltrating CD8(+) T cell antitumor efficacy and exhaustion: molecular insights. *Drug Discov Today*. 2021;26(4):951-67.
6. Fang W, Zhou T, Shi H, Yao M, Zhang D, Qian H, et al. Progranulin induces immune escape in breast cancer via up-regulating PD-L1 expression on tumor-associated macrophages (TAMs) and promoting CD8(+) T cell exclusion. *J Exp Clin Cancer Res*. 2021;40(1):4.
7. Noguchi T, Ward JP, Gubin MM, Arthur CD, Lee SH, Hundal J, et al. Temporally Distinct PD-L1 Expression by Tumor and Host Cells Contributes to Immune Escape. *Cancer Immunol Res*. 2017;5(2):106-17.
8. Chen SH, Dominik PK, Stanfield J, Ding S, Yang W, Kurd N, et al. Dual checkpoint blockade of CD47 and PD-L1 using an affinity-tuned bispecific antibody maximizes antitumor immunity. *J Immunother Cancer*. 2021;9(10).
9. Li Z, Yu L, Chen D, Meng Z, Chen W, Huang W. Protocol for generation of lung adenocarcinoma organoids from clinical samples. *STAR Protoc*. 2021;2(1):100239.